# Learning Chaos In A Linear Way

**Xiaoyuan Cheng**[1,†]    **Yi He**[1,†]    **Yiming Yang**[1,2]    **Xiao Xue**[1]    **Sibo Cheng**[3]
**Daniel Giles**[4]    **Xiaohang Tang**[2]    **Yukun Hu**[1,*]
[1]Dynamic Systems, University College London, United Kingdom
[2]Statistical Science, University College London, United Kingdom
[3]CEREA, ENPC and EDF R&D, Institut Polytechnique de Paris, France
[4]Computer Science, University College London, United Kingdom
† Equal contribution    * Corresponding author (`yukun.hu@ucl.ac.uk`)

## Abstract

Learning long-term behaviors in chaotic dynamical systems, such as turbulent flows and climate modelling, is challenging due to their inherent instability and unpredictability. These systems exhibit positive Lyapunov exponents, which significantly hinder accurate long-term forecasting. As a result, understanding long-term statistical behavior is far more valuable than focusing on short-term accuracy. While autoregressive deep sequence models have been applied to capture long-term behavior, they often lead to exponentially increasing errors in learned dynamics. To address this, we shift the focus from simple prediction errors to preserving an invariant measure in dissipative chaotic systems. These systems have attractors, where trajectories settle, and the invariant measure is the probability distribution on attractors that remains unchanged under dynamics. Existing methods generate long trajectories of dissipative chaotic systems by aligning invariant measures, but it is not always possible to obtain invariant measures for arbitrary datasets. We propose the Poincaré Flow Neural Network (PFNN), a novel operator learning framework designed to capture behaviors of chaotic systems without any explicit knowledge of the invariant measure. PFNN employs an auto-encoder to map the chaotic system to a finite-dimensional feature space, effectively linearizing the chaotic evolution. It then learns the linear evolution operators to match the physical dynamics by addressing two critical properties in dissipative chaotic systems: (1) contraction, the system's convergence toward its attractors, and (2) measure invariance, trajectories on the attractors following a probability distribution invariant to the dynamics. Our experiments on a variety of chaotic systems, including Lorenz systems, Kuramoto-Sivashinsky equation and Navier–Stokes equation, demonstrate that PFNN has more accurate predictions and physical statistics compared to competitive baselines including the Fourier Neural Operator and the Markov Neural Operator.

## 1 Introduction

**A View to Understand a Chaotic System.** Imagine a box filled with $10^{24}$ identical gas molecules, each with specific positions and momenta. Physicists (Szász, 1996; Zund, 2002) posed the question:

*Suppose the system starts in a certain state; will it eventually return to a state arbitrarily close to the initial one?*

The question seems intractable for an enormous number ($\propto 10^{24}$) of coupled Hamiltonian equations governing the system evolution. However, Poincaré discovered that the system's long-term behavior could be understood without explicit solutions (Poincaré, 1928). He showed that in a closed system with a finite measure, almost all initial states would eventually return close to their starting conditions (detailed proof refers to C.1). This is shown to be the case because the probability measure of the phase space is invariant under the dynamics. This led to the development of the *mean ergodic theorem* (Birkhoff, 1927; Koopman & Neumann, 1932; Neumann, 1932a; Birkhoff & Koopman, 1932), which asserts that in ergodic systems, time averages converge to space averages. Standing on the shoulders of these scientific giants, we recognize that while solving stepwise precise solutions for

chaotic dynamics may be intractable, adopting a measure-theoretic view can significantly simplify the problem.

Understanding the behaviour of chaotic systems remains crucial for applications in weather forecasting, climate modelling, and fluid dynamics (Tang et al., 2020). Over the past decade, forecasting the long-term behavior of chaotic systems has posed a substantial challenge to the machine learning community (Yu et al., 2017; Mikhaeil et al., 2022; Wan et al., 2023; Yang et al., 2025). These dynamical systems are characterized by instabilities and sensitivity to initial conditions. A small perturbation to any given state can cause trajectories to diverge exponentially, a phenomenon attributed to positive Lyapunov exponents. Consequently, accurately predicting the long-term trajectory of such systems is non-trivial. To tackle these challenges, two primary streams have emerged in learning dynamical systems: 1) deep sequence models, which leverage sequential neural networks to learn the temporal patterns, and 2) operator learning, learning the integral-differential operators directly without knowing the differential equations.

**Deep Sequence Models.** Previous studies have focused on predicting short-term dynamics, mainly using deep sequence models by minimizing the mean squared error (MSE) of the next-step prediction. Commonly used models include recurrent neural networks (RNNs) (Lipton et al., 2015; Vlachas et al., 2020), long short-term memory networks (LSTMs) (Mikhaeil et al., 2022), reservoir computing (Pathak et al., 2018; Tanaka et al., 2019) and Transformers (Woo et al., 2024). These approaches have been applied to classic examples like the Lorenz 63 system (Lorenz, 1963) and turbulent flows (Pathak et al., 2018). However, due to the instability of trajectories, these methods often suffer from exponential error accumulation (Ribeiro et al., 2020), which hampers their ability to model chaotic systems effectively. To stabilize predictions, various strategies have been proposed, such as constraining the recurrence matrix to be orthogonal (Helfrich et al., 2018; Henaff et al., 2016), skew-symmetric (Chang et al., 2019), or ensuring globally stable fixed point solutions (Kag et al., 2020). However, these methods struggle with simulating chaotic systems since chaotic systems are usually hyperbolic and aperiodic (Hasselblatt & Katok, 2002). To address this issue, specialized models tailored to chaotic systems are necessary to accurately capture their complex patterns and behaviors. Recognizing that *dissipative chaotic systems* with *attractors exhibit ergodic behaviours*, new sequential models have been developed that go beyond simply minimizing MSEs. For instance, Jiang et al. (2024) improve the learning of chaotic systems by incorporating transport distances to the invariant measure, which is assumed to be *known*. This ensures that the model's predictions are aligned with the true distribution of the system. The methods proposed in these two papers (Jiang et al., 2024; Schiff et al., 2024) generate samples directly from the neighborhood of the attractors and estimate the invariant probability distribution from these generated samples. Differently, Schiff et al. (2024) removed the need for knowledge of the invariant measure of underlying systems, enabling direct measurement of the invariant distribution from trajectories. By incorporating the estimated invariant distribution as a regularized transport term, they train deep sequence models by enforcing long trajectories to match the estimated invariant distribution. However, a significant challenge for these methods lies in accurately estimating the invariant measure on attractors from arbitrary datasets, where this invariant distribution is usually unknown.

**Operator Learning.** Another prominent approach to studying the long-term behavior of dynamical systems is through operator theory. Two classic methods in this domain are the transfer operator (Demers & Zhang, 2011) and the Koopman operator (Bevanda et al., 2021). The transfer operator captures the evolution of probability density functions in chaotic dynamical systems, making it a powerful tool for analyzing statistical mechanics (Lagro et al., 2017), chaos (Jiménez, 2023), and fractals (Ikeda et al., 2022). The Koopman operator, often considered the adjoint of the transfer operator, focuses on the evolution of feature functions. Both of these methods primarily aim to capture the long-term behaviors of chaotic dynamical systems with an invariant measure (Adams & Quas, 2023; Das et al., 2021; Valva & Giannakis, 2023). Although the dynamics on invariant sets govern long-term behavior (Mori & Kuramoto, 2013; Ornstein, 1989; Li et al., 2017), focusing solely on ergodic behaviors can significantly distort short-term predictions. Traditionally, the Koopman operator is approximated using kernel methods (Ikeda et al., 2022; Kostic et al., 2022) or dynamic mode decomposition (DMD) (Williams et al., 2015; Takeishi et al., 2017). Recently, Koopman learning has emerged as a prominent approach for modeling the dynamics of differential equations by leveraging autoencoder architectures to approximate a finite-rank linear operator within a learned feature space (Lusch et al., 2018; Nathan Kutz et al., 2018; Brunton & Kutz, 2023). This approach has facilitated the development of various methods for capturing general dynamical behaviors, such

as time series (Wang et al., 2022; Liu et al., 2024) or graph dynamics (Mukherjee et al., 2022). However, no specific Koopman learning framework has been designed to effectively capture the behaviors of dissipative chaos. Integrating deep learning techniques with operator theory has led to the development of various neural operators aimed at solving differential equations. For instance, Deep Operator Networks (DeepONet) (Lu et al., 2019) serve as universal approximators for initial value problems, while Fourier Neural Operators (FNO) (Li et al., 2020; Kovachki et al., 2021) learn deep Fourier features to minimize loss functions in Sobolev space, effectively solving differential equations with high-order terms. Both DeepONet and FNO aim to capture accurate solution operators for initial value problems in infinite-dimensional function space. However, chaotic systems often require analysis beyond traditional initial value problems with unstable solutions, necessitating a specialized architecture to address. To address this, the Markov Neural Operator (MNO) (Li et al., 2022) improves long-term predictions for dissipative chaotic systems by introducing a concept called an "absorbing ball." This "absorbing ball" acts like a boundary that ensures the chaotic system does not go completely off course over time, guiding it back to a stable range. However, choosing the right size for this absorbing ball is tricky. If the ball is too large, it might make inaccurate predictions, as the system could stray too far from the actual behavior.

To address the aforementioned challenges, we propose the Poincaré Flow Neural Network (PFNN) established on Koopman theory, a novel model designed to capture both dynamics and long-term behaviors of dissipative chaotic systems without requiring *prior knowledge or assumptions* of the invariant measure from the raw data. Unlike the previous Koopman deep learning methods for general dynamics (Li et al., 2017; Lusch et al., 2018; Nathan Kutz et al., 2018; Brunton & Kutz, 2023), our method is tailored to learn dissipative chaos by embedding intrinsic physical properties. Specifically, dissipative chaos can be separated into two phases based on the physical properties: a) Contraction Phase: where the system converges toward attractors due to energy dissipation, which governs transient short-term dynamics; b) Measure-Invariant Phase: where the system fully explores the bounded invariant set according to the invariant distribution over time, enabling the derivation of statistical properties from time averages (ergodic behaviours). PFNN learns a linear contraction operator and a unitary operator to accurately model in an infinite-dimensional feature space the corresponding physical behaviors of the contraction phase and measure-invariant phase (as shown Figure 1), respectively. Our approach avoids relying on a known invariant measure for guiding long-term predictions. Instead, we employ simple one-step forward learning, regularized with physical constraints (contraction and unitary) on the linear operators.

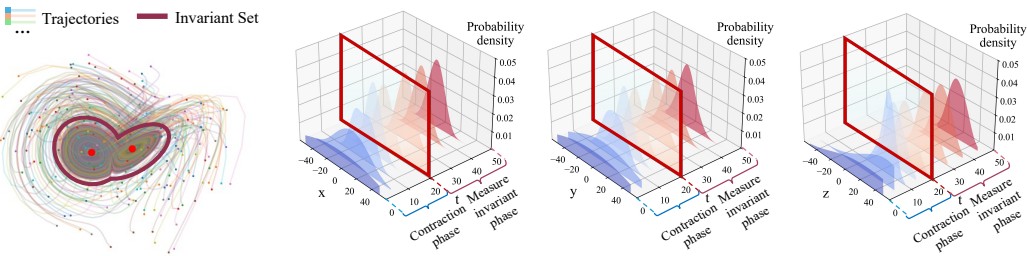

(a) Trajectory samples    (b) The evolution of probability density function of three dimensions

Figure 1: Contraction and measure-invariant phase in chaotic dynamics (Lorenz 63 system): (a) trajectories from random initial states all spiral toward attractors, ultimately moving within the butterfly-shaped invariant set; (b) the probability density of the trajectories samples in each state dimension keeps changing over the earlier evolution, which indicates a contraction phase in the beginning; whilst the probability density becomes invariant over the later evolution, which echos the states finally move consistently within the invariant set.

## 2 BACKGROUND AND PROBLEM FORMULATION

**Notation.** $(X, \mathcal{B}, \mu)$ denotes the measure space, with set $X$, Borel $\sigma-$algebra $\mathcal{B}$ and measure $\mu$. The forward map on the state space is denoted as $T$. The Lebesgue space with $p-$norm is represented as $\mathcal{L}^p(X, \mu)$, which is abbreviated as $\mathcal{L}^p$ in this paper. Specifically, the Lebesgue space $\mathcal{L}^2$ is equipped with an inner product structure as $\langle \cdot, \cdot \rangle$. The functions $\phi, \psi$ are denoted as feature functions in the $\mathcal{L}^2$ space. $\square^T$ denotes the transpose of a real matrix, and $\square^*$ denotes the conjugate transpose.

$\mathcal{O}(\cdot)$ denotes an asymptotically tight upper bound while $o(\cdot)$ indicate that the upper bound is not asymptotically tight. The ceiling function is denoted as $\lceil x \rceil = \min\{n \in \mathbb{Z} \mid x \leq n\}$.

**Problem Setup.** In what follows, we focus on forecasting dissipative chaotic dynamical systems that can be described as

$$z_{k+1} = T(z_k),\ z \in \mathcal{M} \subset \mathbb{R}^m, \tag{1}$$

where $z_k$ denotes the state of the system at time $k \in \mathbb{N}$ and $\mathcal{M}$ is a bounded space. The forward map $T : \mathcal{M} \to \mathcal{M}$ is nonlinear, pushing states from time $k$ to time $k+1$. The above model assumes the states depend only on the current state $z_k$ regardless of information from long historical states.

To forecast future states, one might use a neural network to approximate the forward map $T$. However, the resulting model ignores prior knowledge related to the problem, and it can be challenging to analyze. In contrast, our model is based on learning the evolution of latent features $\phi \in \mathcal{L}^2$ in a linear way. In this framework, the evolution of the system becomes linear in the function space $\mathcal{L}^2$. Such an evolution map is the so-called neural operator. Our model adopts the Koopman operator (Koopman, 1931), the forward map $T$ induces a linear operator $\mathcal{G}$ as

$$\mathcal{G}\phi = \phi \circ T, \tag{2}$$

where $\phi$ is the feature map as $\phi : \mathcal{M} \to \mathbb{R}$, with $\phi \in \mathcal{F} \subset \mathcal{L}^2$ being function space $\mathcal{F}$ on $\mathcal{M}$. Essentially, the operator $\mathcal{G}$ describes the linear evolution of factorized features. Consequently, future states of the features can be obtained by iteratively applying $\mathcal{G}$. More specifically, the feature $\phi(z_k)$ is mapped to $\phi(z_{k+1})$ under the operator $\mathcal{G}$. The evolution of the entire feature space can be expressed as a linear combination of features:

$$\mathcal{G}(\alpha_1 \phi_1 + \alpha_2 \phi_2 + \cdots) = (\alpha_1 \phi_1 + \alpha_2 \phi_2 + \cdots) \circ T, \tag{3}$$

where $\alpha_i \in \mathbb{R}$ is the weight corresponding to the feature functions $\phi_i$.

**Definition 2.1 (Measure Preserving Transformation and Ergodicity (Walters, 2000))** *Let* $(X, \mathcal{B}, \mu, T)$ *be measure preserving transformation (MPT), meaning that for every* $E \in \mathcal{B}$, *the measure satisfies* $\mu(T^{-1}E) = \mu(E)$. *MPT is said to be ergodic if, for any invariant set* $E$, *either* $\mu(E) = 0$ *or* $\mu(X \setminus E) = 0$. *In this case,* $\mu$ *is referred to as an ergodic measure.*

Ergodicity $T : X \to X$ ensures that $\mu(T^{-1}E) = \mu(E) \Leftrightarrow \mu(E) = \{0, 1\}$, meaning the system almost surely follows the behavior described by the invariant measure. Simply speaking, the ergodicity implies that spatial statistics are the same as the temporal statistics.

**Definition 2.2 (Global Attractor (Hasselblatt & Katok, 2002))** *A compact, invariant set* $A$ *is called a global attractor if, for any bounded set* $B \subset \mathcal{M}$ *and existing a time* $k^* \in \mathbb{N}$ *such that* $k > k^*$ *such that* $T^k(B)$ *is contained within the neighbourhood of* $A$.

In this paper, we aim to forecast dissipative chaotic dynamical systems by considering invariant measures on attractors. Given an arbitrary initial state, such systems typically converge to their attractors (an interpretation of this can be found in Appendices B.1 and B.2). Due to the dissipative nature of the dynamics, the phase space contracts, meaning that all trajectories will eventually approach specific regions within the phase space. The contraction reflects energy dissipation from a physical standpoint. Once the system enters the neighbourhood of an attractor, it becomes ergodic, and the system's trajectories distribute across the attractor following an invariant measure that does not vary over time (see 2.1). Many dynamical systems have been proven to exhibit this property, including the Lorenz 63 (Tucker, 1999), Lorenz 96 (Maiocchi et al., 2024) and Kuramoto-Sivashinsky (Temam, 2012). In other words, the trajectory of a chaotic system will span the attractor in a way that accurately reflects the statistical properties of the entire invariant set, as shown in Figure 1.

## 3 METHOD

In the following sections, we introduce the Poincaré Flow Neural Network (PFNN) designed for forecasting dissipative chaotic dynamical systems. PFNN consists of two steps: first, mapping the system into a finite-dimensional feature space using an auto-encoder (AE); second, embedding physical properties in the finite-rank operator to capture behaviors in contraction and measure-invariant phases.

## 3.1 FINITE-RANK OPERATOR APPROXIMATION

A popular approach for operator learning $\mathcal{G} : \mathcal{L}^2 \rightarrow \mathcal{L}^2$ is to construct a finite-dimensional approximation. Specifically, a convergent approximation method for prediction problems can be developed by utilizing the fact that $\mathcal{G}$ is a bounded (and therefore continuous) linear operator, without explicitly considering its spectral properties. Given an arbitrary orthonormal basis $\{\phi_1, \phi_2, \dots\}$ of $\mathcal{L}^2$, we define the operator's orthogonal projection as $\Pi_L : \mathcal{L}^2 \rightarrow \text{span}\{\phi_1, \dots, \phi_L\}$. The finite-rank operator is then given by $\mathcal{G}_L = \Pi_L \mathcal{G} \Pi_L$, where the operator $\mathcal{G}_L$ is characterized by the matrix elements $\mathcal{G}_{ij,L} = \langle \phi_i, \mathcal{G}\phi_j \rangle$ with $1 \leq i,j \leq L$. Due to the continuity of $\mathcal{G}$, the sequential predictions of the operator $\mathcal{G}_L$ converge pointwise to those of $\mathcal{G}$ on $\mathcal{L}^2$.

The finite-dimensional feature space is spanned by learned adaptive features of the neural networks used in this paper. The learned encoder and decoder are denoted as $g_{\theta_1}^{\text{en}}$ and $g_{\theta_2}^{\text{de}}$, with parameters $\theta_1$ and $\theta_2$ respectively. Specifically, the encoder $g_{\theta_1}^{\text{en}}$ functions as the projection operator onto the learned latent feature space, effectively embedding the state space into the function space. The decoder $g_{\theta_2}^{\text{de}}$ then maps the feature space back to the original space. In this setup, the operator $\mathcal{G}$ and $\theta_1, \theta_2$ are learned jointly by minimizing the following loss function:

$$\arg \min_{\hat{\mathcal{G}}, \theta_1, \theta_2} \mathbb{E}_{z \sim \nu} \big[ \| \hat{\mathcal{G}} g_{\theta_1}^{\text{en}}(z_k) - g_{\theta_1}^{\text{en}}(z_{k+1}) \| + \gamma \| g_{\theta_2}^{\text{de}} \circ g_{\theta_1}^{\text{en}}(z_k) - z_k \|_2 \big], \quad (4)$$

where $\hat{\mathcal{G}}$ is the approximated finite-rank operator, and $\nu$ is data distribution. The first term in Equation 4 represents the prediction loss, while the second term accounts for the reconstruction loss with a coefficient $\gamma$ denoted as $\Gamma_{rec}$. To ensure a consistent solution in the feature space, the autoencoder must satisfy the bijective property, meaning that encoder and decoder satisfy the constrained relationship $I_d = g_{\theta_2}^{\text{de}} \circ g_{\theta_1}^{\text{en}}$, where $I_d$ denotes the identity matrix. For the class of finite-rank operators, we prove Theorem 3.1 regarding the PFNN. This result shows that the finite-rank operator can approximate the infinite-dimensional operator with a sufficiently small error.

**Theorem 3.1 (Convergence of Finite-Rank Operator)** *Let $\mathcal{M} \subset \mathbb{R}^m$ be a compact set, the solution operator $\mathcal{G} : \mathcal{L}^2 \rightarrow \mathcal{L}^2$ associated to the dynamics is locally Lipschitz. Then, for a sufficient large dimension $L \in \mathbb{N}$ and a sufficiently small number $\epsilon > 0$, there exists an approximated finite-rank operator $\hat{\mathcal{G}} : \mathcal{L}^2 \rightarrow \mathcal{L}^2$ converging pointwise such that $\sup_{z \in \mathcal{M}} \| \hat{\mathcal{G}} g_{\theta_1}^{en}(z) - \mathcal{G}\phi(z) \|_2 \leq \epsilon$.*

The theorem reflects that the finite-rank operator, derived from the infinite-dimensional operator, operates in a function space spanned by learned latent feature functions. By selecting a sufficiently large $L$ as the dimension of the latent feature space, stepwise prediction can be achieved (see the procedure in Appendix C). While the existence of the finite-rank operator can be proven, relying solely on this approximation for time-stepping can lead to exponential error accumulation. To address this, long-term predictions can be improved by incorporating physical properties. In the following sections, we introduce two loss functions designed to enhance long-term prediction accuracy. To distinguish the different operators, those applied during the contraction phase and the measure-invariant phase are denoted as $\mathcal{G}_c$ and $\mathcal{G}_m$, respectively.

## 3.2 STAGE I: CONTRACTION PHASE

In accordance with the definition of attractors in Section 2.2, a dissipative system induces a contraction operator (Lumer & Phillips, 1961) due to the presence of attractors. In this stage, there exists a volume contraction. It means any bounded set $B$ containing the invariant set $A$ will contract towards $A$. To characterize this contraction, we constrain the spectral properties of the operator $\mathcal{G}_c$, ensuring $\|\mathcal{G}_c\| \leq 1$. Specifically, we focus on $\mathcal{G}_c^T \mathcal{G}_c$ instead of $\mathcal{G}_c$ because $\mathcal{G}_c^T \mathcal{G}_c$ is symmetric (Franklin, 2012) with real, non-negative eigenvalues, and enforce the condition:

$$\langle \mathcal{G}_c \phi, \mathcal{G}_c \phi \rangle \leq \lambda^2 \|\phi\|_2^2, \quad 0 < \lambda \leq 1. \quad (5)$$

Eigenvalues of $\mathcal{G}_c$ with a modulus bounded by 1 reflect the contraction (refer to Lumer–Phillips theorem in Appendix D). To enforce these contraction properties in practice, we introduce a specialized loss function as

$$\arg \min_{\hat{\mathcal{G}}_c, \theta_1, \theta_2} \mathbb{E}_{z \sim \nu} \big[ \| \hat{\mathcal{G}}_c g_{\theta_1}^{\text{en}}(z_k) - g_{\theta_1}^{\text{en}}(z_{k+1}) \| + \gamma_1 \Gamma_{con} + \gamma \Gamma_{rec} \big], \quad (6)$$

where the contraction term is defined as $\Gamma_{con} := \text{ReLU}\big(\sigma(\hat{\mathcal{G}}_c^T \hat{\mathcal{G}}_c - I_d)\big) = \max\big(\sigma(\hat{\mathcal{G}}_c^T \hat{\mathcal{G}}_c - I_d), 0\big)$ with coefficient $\gamma_1 \in (0, 1)$. $\sigma(\cdot)$ is denoted as eigenvalues of operator, and $\text{ReLU}\big(\sigma(\hat{\mathcal{G}}_c^T \hat{\mathcal{G}}_c - I_d)\big)$ [1] constraints the negative semidefinite of $\hat{\mathcal{G}}_c^T \hat{\mathcal{G}}_c - I_d$. The $\Gamma_{con}$ encourages the contraction of the state space by making the eigenvalues of $\hat{\mathcal{G}}_c^T \hat{\mathcal{G}}_c$ no larger than one, based on Equation 5. The data distribution $\nu$ is lying in the bounded set as $\mathcal{M} \setminus A$, which encourages the state space to contract towards the invariant set $A$ from outside.

However, the invariant set of the global attractor is typically not directly observable in the raw dataset. To address this, one can utilize the dissipation property to effectively truncate the dataset, preparing it for training various operators. Dissipative chaotic systems take time to approach their invariant distribution (Žnidarič, 2015), which corresponds to the relaxation time in physics (or mixing time in ergodic theory). The relaxation time is bounded by the log-Sobolev time (Bauerschmidt & Dagallier, 2024; Mori & Shirai, 2020), which provides a stronger constraint on the convergence rate to the invariant distribution. More specifically, the relaxation time estimates the time $k$ required for a system to approach its invariant distribution $\mu_*$ over the invariant set $A$, starting from an arbitrary initial state. The relaxation time can be computed using the log-Sobolev inequality, typically in the form of an exponential rate decay $\mathcal{O}(\frac{1}{c_{LSI}} \log(\frac{1}{\epsilon}))$ (Feng & Iyer, 2019), where $c_{LSI}$ represents the log-Sobolev constant and $\epsilon$ is the tolerance. In this case, the following holds: $d(\mu_k, \mu_*) \leq \epsilon$ after $k \propto \lceil \frac{1}{c_{LSI}} \log(\frac{1}{\epsilon}) \rceil$ steps, where $d(\cdot, \cdot)$ measures the total variation distance between the two distributions. Although determining the log-Sobolev constant can be challenging, it can be inferred that the probability of lying outside $A$ is sufficiently small, such that the timestep $k$ is proportional to $\lceil \frac{1}{c_{LSI}} \log(\frac{\|\phi\|_2}{\epsilon}) \rceil$. Note that $\phi$ is the feature function of state $z$. Thus, the empirical calculation of relaxation time is $\lceil \frac{1}{c_{LSI}} \log(\frac{\|z\|_2}{\epsilon}) \rceil$ and the norm of state $\|z\|_2$ can be understood as *energy-like* quantity in physics. As illustrated in Figure 1, the initial distribution contracts toward the invariant distribution. Thus, it is feasible to use the trajectory data before $k$ for the training of the contraction phase.

## 3.3 Stage II: Measure-Invariant Phase

Once the system reaches the global attractor, it explores the bounded set $A$ based on the invariant measure, with spatial and long-term temporal statistics aligned. Von Neumann (Neumann, 1932b) established that MPT induces a unitary operator on the corresponding Hilbert space $\mathcal{L}^2$. Consequently, a unitary operator $\mathcal{G}$ can be learned to describe the system's evolution with feature space.

**Proposition 3.2 (Unitary Property)** *When the map $T$ is MPT, the operator $\mathcal{G}_m$ is unitary on $\mathcal{L}^2$. That is, for all $\phi, \psi \in \mathcal{L}^2$,*

$$\langle \mathcal{G}_m \phi, \mathcal{G}_m \psi \rangle = \langle \phi, \psi \rangle, \tag{7}$$

*where $\langle \phi, \psi \rangle = \int_X \langle \phi(z), \psi(z) \rangle d\mu$ is the inner product on $\mathcal{L}^2$.*

**Theorem 3.3 (Koopman-von Neumann (KvN) Ergodic Theorem (Neumann, 1932b))** *Let $(X, \mathcal{B}, \mu, T)$ be an MPT. If $\phi \in \mathcal{L}^2$, then $\lim_{N \to \infty} \frac{1}{N} \sum_{k=0}^{N-1} \mathcal{G}_m^k \phi = \lim_{N \to \infty} \frac{1}{N} \sum_{k=0}^{N-1} \phi \circ T^k = \overline{\phi}$ where $\overline{\phi}$ is invariant. If $T$ is ergodic, then $\overline{\phi} = \int \phi d\mu$.*

The proof can be referred to Appendix C.3 and C.4. The KvN ergodic theorem shows that for an ergodic MPT $T$, the associated operator $\mathcal{G}$ is unitary, preserving norms in $\mathcal{L}^2$. This means the time averages of $\mathcal{L}^2$ functions converge to their space averages, ensuring the long-term predictability of chaotic systems. The theorem links chaos with operator theory, providing a framework to study the statistical behavior through $\mathcal{G}_m$. The spectrum of the operator $\mathcal{G}_m$ lies on the complex unit circle $\sigma(\mathcal{G}_m) \subset \{r \in \mathbb{C} \mid |r| = 1\}$, with a positive measure on both discrete and continuous spectrum for chaotic systems (Koopman & Neumann, 1932). The presence of a continuous spectrum indicates chaotic behavior, meaning the learned operator should have its spectrum densely distributed along the unit circle. The validity of the learning results can be assessed by analyzing the spectrum of the operator, as shown in Figure 2.

---

[1] PyTorch (2014) supports the auto-differentiation of eigenvalue function of symmetric matrix or Hermitian matrix, which improves computational stability.

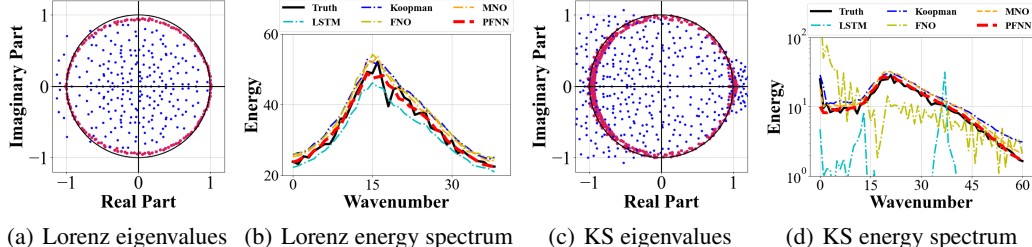

(a) Lorenz eigenvalues  (b) Lorenz energy spectrum  (c) KS eigenvalues  (d) KS energy spectrum

Figure 2: Operator eigenvalues and energy spectrum of the Lorenz 96 and KS equations: The **eigenvalues** in subfigures (a) and (c) display the difference of eigenvalue distribution of learned operator layer from the aspect of the measure-invariant constraint. Red points represent the PFNN's eigenvalues in the measure-invariant phase, densely distributed near the unit circle in the complex plane, indicating invariant measure under evolution. In contrast, blue points represent eigenvalues without the measure-invariant constraint, showing values both inside and outside the unit circle, leading to gradient vanishing and blow-up for long-term predictions. Subfigures (b) and (d) show the corresponding **energy spectrum** for the Lorenz 96 and KS equations, illustrating the system's energy distribution across modes. The energy spectra of PFNN's prediction fit the true energy spectra most by the introduction of the measure-invariant constraint.

To determine the unitary operator in the ergodic state (see Proposition 3.2), we leverage the intrinsic property that $\mathcal{G}_m$ and its conjugate transpose $\mathcal{G}_m^*$ satisfy $\mathcal{G}_m^* \mathcal{G}_m = I_d$. This property ensures that $\mathcal{G}_m$ is unitary, preserving norms and inner products, and establishes a well-defined backward dynamic. Specifically, the backward operator $\mathcal{G}_m^{-1}$ is equal to $\mathcal{G}_m^*$. This conjugate transpose relationship guarantees that $\mathcal{G}_m^* \mathcal{G}_m = \mathcal{G}_m \mathcal{G}_m^* = I_d$, indicating that $\mathcal{G}_m$ is invertible and the inversion is consistent with unitarity (see theoretical aspects in Appendix D.3). To learn $\mathcal{G}_m$ effectively, we use a bi-directional approach that captures both forward and backward dynamics. $\mathcal{G}_m$ predicts the forward process, while $\mathcal{G}_m^*$ governs the reverse. Aligning the learning of $\mathcal{G}_m$ and $\mathcal{G}_m^*$ ensures the operator remains unitary and accurately represents the system's dynamics in both directions.

According to the estimated relaxation time, the trajectories of timestep $k > \lceil \frac{1}{c_{LSI}} \log(\frac{\|z\|_2}{\epsilon}) \rceil$ are used to train the measure-invariant phase. The measure-invariant loss function becomes

$$
\arg \min_{\hat{\mathcal{G}}_m, \theta_1, \theta_2} \mathbb{E}_{z \sim \mu} \Big[ \underbrace{\|\hat{\mathcal{G}}_m g_{\theta_1}^{en}(z_k) - g_{\theta_1}^{en}(z_{k+1})\|_2}_{\text{forward loss}} + \underbrace{\|\hat{\mathcal{G}}_m^* g_{\theta_1}^{en}(z_{k+1}) - g_{\theta_1}^{en}(z_k)\|_2}_{\text{backward loss}} \\
+ \gamma_2 \underbrace{\left( \|\hat{\mathcal{G}}_m^* \hat{\mathcal{G}}_m - I_d\|_F + \|\hat{\mathcal{G}}_m \hat{\mathcal{G}}_m^* - I_d\|_F \right)}_{\text{consistent constraint}} + \gamma \Gamma_{rec} \Big],
$$
(8)

where $\gamma_2 \in (0, 1)$ is the regularized coefficient, $\| \cdot \|_F$ is denoted as the Frobenius norm, and the distribution of $\mu$ follows the distributions with timestep larger than integer $k$. The loss function of the first line is represented as forward and backward prediction, respectively. The last term in Equation 8 is a consistency constraint on the backward and forward process of $\mathcal{G}$. The combination of three terms in Equation 8 implies the unitary evolution in ergodic theorem in Sections 3.2 and 3.3.

Equations 4, 6 and 8 will be used for training the corresponding operators to predict the distinct chaotic behaviors. We provide the pseudocode in Appendix E.

## 4 NUMERICAL EXPERIMENTS AND ABLATION STUDY

In this section, we empirically evaluate the performance of PFNN in chaotic systems by comparing it against four baselines: (a) LSTM (Vlachas et al., 2018), a type of RNN architecture designed to learn long-term dependencies in sequential data; (b) Koopman operator (Pan et al., 2023), a linear operator that represents the evolution of functions in a dynamical system; (c) Fourier neural operator (FNO) (Kovachki et al., 2023), which uses fast Fourier transform (FFT) to efficiently parameterize integral operators and learn the system evolution; and (d) Markov neural operator (MNO) (Li et al., 2022), which enhances FNO by imposing a hard-coded constraint to better capture dissipative chaos. These baselines cover both deep sequence models and operator learning methods.

**Learning tasks.** (1) Lorenz 63 (Lorenz, 1963): A 3-dimensional simplified model of atmospheric convection, known for its chaotic behavior and sensitivity to initial conditions. (2) Lorenz 96 (Lorenz, 1996): A surrogate model for atmospheric circulation, characterized by the coupled differential equations with periodic boundary conditions and varying state dimensions. To test PFNN across different dimensionalities, we generated datasets with state dimensions of 9, 40, and 80, respectively. (3) Kuramoto-Sivashinsky (KS) equation (Papageorgiou & Smyrlis, 1991): A fourth-order nonlinear partial differential equation that models diffusive instabilities and chaotic behavior in systems, such as fluid dynamics, and reaction-diffusion processes. In this case, we uniformly sub-sampled 128 states from a dense spatial discretization during integration as system states. (4) Kolmogorov flow governed by Navier–Stokes (NS) equation Temam (2012): A two-dimensional shear flow commonly used in fluid dynamics to study turbulence, characterized by sinusoidal velocity fields in one direction and external forcing in the perpendicular direction. In our experiments, we utilize a $64 \times 64$ resolution field for studying.

**Training details.** Given the generated trajectories, each trajectory was divided into two phases: contraction and measure-invariant, with the split occurring at an estimated relaxation time. The data from each stage was then segmented into single-step pairs of observations and true states, which were subsequently shuffled to create the training data. All trajectories are sampled from random initial states, with no prior statistical information available to guide the training process.

**Evaluation metrics.** The models are evaluated using specific metrics: (1) Normalized Root Mean Square Error (NRMSE) for short-term performance, measuring prediction accuracy within a 10-step roll-out; (2) Kullback–Leibler divergence (KLD), and (3) Maximum Mean Discrepancy (MMD) for long-term performances, both of which compare the predicted and true distributions estimated from roll-out samples of the models and the dataset; (4) Turbulent Kinetic Energy (TKE) is distribution of turbulent energy districture, which reflects the error of TKE between the learned model and ground truth. Notably, KL divergence can be problematic in high-dimensional spaces, where probability distributions are overly spread out. To address this, we performed kernel principal component analysis (KPCA) to identify a low-dimensional subspace ($m = 3$), and used kernel density estimation (KDE) to estimate the predicted and true distributions. Further details on the learning tasks, training data, evaluation metrics, and baseline models can be found in Appendix G.

Table 1: Performance comparison of models (FNO, LSTM, Koopman, MNO, and PFNN) across various dynamical systems (the Lorenz 63, Lorenz 96, Kuramoto-Sivashinsky and Kolmogorov Flow) with corresponding evaluation metrics: NRMSE, KL divergence, MMD, and TKE. The best performance is highlighted in bold. NRMSE is evaluated 50 times using different initial states, with the results reported as mean±standard deviation. Here, $m$ represents the state dimension.

| | Metrics | LSTM | Koopman | FNO | MNO | PFNN |
|---|---|---|---|---|---|---|
| | NRMSE | 1.83±0.85 | 0.67±0.33 | 0.53±0.24 | **0.44±0.20** | 0.49±0.26 |
| Lorenz 63 ($m = 3$) | KLD | $+\infty^2$ | 0.66 | 0.61 | 0.37 | **0.29** |
| | MMD | 1.05 | 0.46 | 0.51 | 0.48 | **0.39** |
| | NRMSE | 1.57±0.44 | 0.61±0.23 | 0.42±0.36 | 0.28±**0.09** | **0.21**±0.14 |
| Lorenz 96 ($m = 9$) | KLD | 3.49 | 2.61 | 2.12 | 2.01 | **1.87** |
| | MMD | 0.21 | 0.13 | 0.11 | **0.10** | **0.10** |
| | NRMSE | 4.77±2.15 | 0.74±0.12 | **0.09**±0.07 | 0.15±**0.04** | 0.11±0.06 |
| Lorenz 96 ($m = 40$) | KLD | $+\infty$ | 0.16 | 0.12 | **0.11** | **0.11** |
| | MMD | 0.854 | 0.0239 | 0.015 | **0.011** | **0.011** |
| | NRMSE | 2.28±0.73 | 1.64±0.49 | 0.51±0.22 | 0.37±0.16 | **0.32±0.09** |
| Lorenz 96 ($m = 80$) | KLD | 1.32 | 0.24 | 0.15 | 0.18 | **0.10** |
| | MMD | 0.056 | 0.0088 | 0.0094 | 0.0084 | **0.0083** |
| | NRMSE | 13.87±4.69 | 1.71±0.35 | 0.53±0.29 | 0.31±0.07 | **0.25±0.05** |
| KS ($m = 128$) | KLD | $+\infty$ | 52.51 | 142.73 | 24.53 | **9.37** |
| | MMD | 0.996 | 0.204e | 0.915 | 0.031 | **0.011** |
| NS ($m = 64 \times 64$) | NRMSE | 10.65 ± 4.09 | 10.06 ± 4.23 | 4.22±4.23 | 3.81±4.48 | **3.42±3.79** |
| | TKE | 2.2965 | 1.8092 | 2.0973 | 1.5828 | **1.3638** |

---

[2]The LSTM yields constant predictions, resulting in a degenerate (Dirac delta) distribution at that value, which has different support from the true distribution, leading to an infinite KL divergence.

## 4.1 RESULTS AND COMPARISON

Table 1 presents the performance of all baseline methods and PFNN, evaluated using four metrics across the selected learning tasks. PFNN consistently outperforms the baselines, particularly excelling in KLD and MMD, which highlights its capability to capture the long-term dynamics of complex, chaotic systems. For short-term prediction, both PFNN and MNO show better accuracy, as reflected by the NRMSE metric. This is likely due to their ability to account for the initial phase of energy dissipation, which characterizes transient behaviors. PFNN (contraction operator) and MNO incorporate physical properties during this phase, resulting in predictions that more accurately reflect the system's dissipative nature. However, MNO enforces a fixed dissipation rate and pre-defined absorbing ball, which can reduce its effectiveness in short-term forecasting. Models like LSTM and Koopman, are hard to capture transient behaviors for states lying outside attractors, exhibit large deviations from the ground truth. Notably, while FNO uses FFT to extract features and provides relatively accurate short-term predictions, its reliance on integer Fourier modes tends to produce periodic behavior. This periodicity conflicts with the non-periodic nature of dissipative chaos.

In terms of long-term statistical properties, PFNN's unitarity directly leads to an invariant distribution, which results in better performance in KLD and MMD. In contrast, the periodic tendencies of integer Fourier modes in FNO and MNO disrupt ergodicity, preventing a full exploration of the invariant sets. This limitation is reflected in the higher KLD and MMD values for FNO and MNO. A key observation that further validates this distinction is evident in the eigenvalue distribution of the learned unitary operator $\hat{\mathcal{G}}_m$ for the Lorenz 96 and KS systems. Specifically, the eigenvalues are densely distributed along the unit circle in Figure 2. This distribution aligns with the presence of a continuous spectrum in chaotic systems, as predicted by the KvN theorem, indicating a positive measure on the continuous spectrum. Additionally, Figure 2 demonstrates the effectiveness of our consistent loss function in maintaining the unitarity of the operator and accurately learning key invariant distributions, which ensures stable long-term predictions in chaotic systems. In contrast, when the model is not constrained by regularized terms in Equation 8, it results in unstable/stable modes. Empirically, this instability causes rapid forecast divergence/convergence and failure to learn the invariant distribution, as shown in the eigenvalue distributions (see blue points in Figure 2(a) and 2(c)). Furthermore, the unitarity of the operator reflects underlying conservation laws, resulting in a more accurate energy spectrum, as demonstrated in Figure 2(b), 2(d) and Figure 3. Due to the periodic behaviors in MNO and FNO, the distribution of TKE is inconsistent with ground truth. More experimental details showcasing the short- and long-term prediction and physical statistics are provdied in Appendix F.

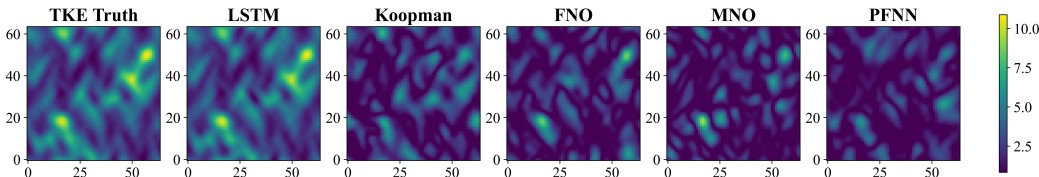

Figure 3: Visualization of the prediction error for Kolmogorov flow trajectories in terms of TKE, demonstrating the model's accuracy in capturing the kinetic energy distribution. We compare the model-predicted trajectories with the ground truth and plot the absolute error in TKE.

## 4.2 ABLATION STUDY

In this section, we revisited the KS system and conducted ablation studies to evaluate the effects of key hyperparameters in PFNN, including the relaxation time $k$, the unitary regularized coefficient $\gamma_2$ (with $\gamma_1$ fixed[3]), and the finite feature dimension $L$. Table 2 includes the metrics evaluated for each configuration, providing insight into how these hyperparameters affect short- and long-term performance.

**Regularized coefficient.** We retrained each model, systematically varying $\gamma_2$ and found that $\gamma_2 \in [0.1, 0.5]$ consistently performs well and achieves a good balance between short-term precision

---

[3]Since the $\gamma_1$ does not affect long-term performance and is trained with a smaller dataset, we performed a grid search over $(0, 1]$ to determine its value.

and long-term consistency. Outside this range, lower $\gamma_2$ reduces long-term consistency, while higher values overly constrain the model, degrading short-term accuracy.

**Feature dimensions.** We explored the impact of latent feature dimension $L$ in the PFNN by varying $L$ while keeping other hyperparameters fixed. From Table 2, we found that $L = 256, 512$ consistently perform well, while smaller values reduce the model's expressiveness to capture complex patterns, while larger values require more data for training and risk overfitting. This ablation study result reflects a classic trade-off in deep learning between model complexity and data volume.

**Log-Sobolev constant.** Systems exhibit varying dissipation rates, making it important to test different relaxation times. In most dissipative chaotic systems (Garbaczewski & Olkiewicz, 2002), the approach to the invariant distribution follows an exponential contraction rate. In this experiment, we varied the relaxation time $k$ and constructed two corresponding datasets for training the contraction and unitary operators in PFNN. Due to the logarithmic relationship, the $\epsilon$ has a weak effect on relaxation time $k$. Therefore, we adjust the relaxation time by fixing $\epsilon = 0.01$ and varying the constant term $\frac{1}{c_{LSI}}$. For the KS equation, dissipation predominantly occurs in the higher Fourier modes, typically for wave numbers exceeding 1 (Papageorgiou & Smyrlis, 1991), leading to slow dissipation characterized by a log-Sobolev constant bounded as $\frac{1}{c_{LSI}} \geq 1$. In such a situation, we test the constant as $\frac{1}{c_{LSI}} = \{1, 3, 5, 7, 9\}$, the model consistently performs well once the $k$ exceeds a certain threshold $\frac{1}{c_{LSI}} = 5$ in KS system. Stable performance was observed when $\frac{1}{c_{LSI}} \geq 5$, highlighting that an appropriate relaxation time is crucial: if too short, it introduces bias, while if too long, many samples lie on the attractor, leading to data contamination from transient behaviors. Furthermore, continuously increasing the parameter beyond this range does not have an obvious effect on the long-term statistics. If the hyperparameter $\frac{1}{c_{LSI}}$ is set too high, the data volume available for the measure-invariant phase in Equation 8 becomes insufficient for training. In other experiments, we set the constant term $\frac{1}{c_{LSI}} = 5$ obtaining a similar result.

Table 2: Ablation experiments showing the effects of varying PFNN hyperparameters. **NRMSE (100 steps)** measures the normalized root mean square error over 100 steps. **KLD** is estimated using a sample size of 110,000 for long-term statistical analysis. In the left column, we fix $\gamma_1 = 0.3$, set the feature dimension $L = 512$, and assign the constant term $\frac{1}{c_{LSI}} = 5$. In the middle column, we fix $\gamma_1 = 0.3$, $\gamma_2 = 0.5$, and $\frac{1}{c_{LSI}} = 5$. In the right column, we fix $\gamma_1 = 0.3$, $\gamma_2 = 0.5$, and set the feature dimension $L = 512$.

| Parameters Metrics | Regularized coefficient, $\gamma_2$ | | | | | Feature dimension, $L$ | | | | Log-Sobolev constant, $\frac{1}{c_{LSI}}$ | | | | |
|---|---|---|---|---|---|---|---|---|---|---|---|---|---|---|
| | 0 | 0.1 | 0.3 | 0.5 | 1 | 128 | 256 | 512 | 1024 | 1 | 3 | 5 | 7 | 9 |
| **NRMSE 100 steps** | 26.59 | 26.67 | 23.11 | **15.79** | 26.51 | 21.59 | 19.13 | **15.79** | 43.44 | 585k | 46.77 | 15.79 | 15.07 | **14.23** |
| **KLD** | 136.68 | 85.33 | 37.21 | **6.55** | 48.09 | 50.12 | 41.33 | **6.55** | 1637.46 | 186k | 9.17k | 6.55 | **4.91** | 5.78 |

## 5 CONCLUSIONS AND LIMITATIONS

**Conclusions.** Learning chaotic behavior poses a significant challenge due to the inherent instability and unpredictability of these systems. To address this complexity, we adopt a physically informed approach and introduced the Poincaré Flow Neural Network (PFNN), a novel operator learning framework designed to capture both the contraction and measure-invariant phases of dissipative chaotic systems. Our method outperforms traditional deep sequence models and neural operators, providing superior short- and long-term predictions while ensuring more consistent statistical outcomes across a variety of dissipative chaotic dynamical system experiments.

**Limitations.** We evaluate the limitations from both theoretical and practical perspectives. Theoretically, the assumption of ergodicity in the second stage does not apply to all chaotic systems, which can lead to a breakdown in operator unitarity for non-ergodic cases. A promising direction for future work is to extend the method to handle non-measure-preserving chaotic systems. Practically, the bi-directional training results in slow convergence due to the high computational complexity of the Frobenius norm. To alleviate this issue, future research could leverage the Hermitian properties of unitary operators or employ Hutchinson's trace estimation to optimize trace calculations and reduce computational load. Lastly, a promising direction is to extend the discretized-grid input in PFNN to a discretization-agnostic input.

## 6 ACKNOWLEDGEMENT

The authors thank the reviewers and the program committee of ICLR 2025 for their insightful feedback and constructive suggestions that have helped improve this work. We also acknowledge the support from the University College London (Dean's Prize, Chadwick Scholarship), the Engineering and Physical Sciences Research Council projects (EP/W007762/1, EP/T517793/1, EP/W524335/1), and the Royal Academy of Engineering (IF-2425-19-AI165).

## 7 REPRODUCIBILITY STATEMENT

The full algorithm of the proposed method can be refered to Appendix E. Code details of numerical experiments are available at `https://github.com/Hy23333/PFNN`.

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

CONTENTS

## A   TABLE OF NOTATIONS

| Notations | Meaning |
| --- | --- |
| $c_{LSI}$ | log-Sobolev constant |
| $g_{en}$ | encoder |
| $g_{de}$ | decoder |
| $k$ | time index |
| $m$ | dimension of state space |
| $z$ | state |
| $A$ | invariant set or attractor |
| $\mathcal{B}$ | Borel $\sigma-$algebra |
| $B$ | bounded set |
| $\mathcal{F}$ | function space |
| $\mathcal{G}$ | forward operator on $\mathcal{L}^2$ space |
| $\mathcal{G}_c$ | contraction operator |
| $\mathcal{G}_m$ | unitary operator |
| $I_d$ | identity matrix |
| $L = D \times m$ | finite rank of approximated operator |
| $\mathcal{M}$ | bounded state space |
| $\mathcal{O}, o$ | describes limiting behavior of a function |
| $T$ | nonlinear forward map |
| $\theta$ | parameters of neural networks |
| $\phi, \psi$ | feature functions in $\mathcal{L}^2$ space |
| $\mu$ | Lebesgue measure |
| $\mu_*$ | invariant distribution |
| $\sigma(\cdot)$ | eigenvalues of operators |
| $d(\cdot, \cdot)$ | total variation distance of probability measures |
| $\mathcal{L}^p(X, \mu)$ | function spaces defined using a natural generalization of the $p$-norm |
| $(X, \mathcal{B}, \mu)$ | measure space |

The Appendix is organized into two main sections: the first three chapters delve into theoretical analysis, while the subsequent chapters address empirical results and algorithm architectures.

We also encourage readers to spend time on the seminal monographs on ergodic theory (Birkhoff, 1927; Koopman & Neumann, 1932; Neumann, 1932a; Birkhoff & Koopman, 1932), as they offer profound insights that enrich the theoretical foundations presented in the Appendix.

## B  DISSIPATION AND CONSERVATION

The reason we would like to discuss Poincaré's claim is to offer a perspective on understanding dissipative chaos from a macro-structural viewpoint. In statistical mechanics, contraction corresponds to dissipative systems, while measure preservation corresponds to conservative systems. Below, we introduce two key concepts.

**Definition B.1 (Wandering Set (Hasselblatt & Katok, 2002))** *Let $T : X \to X$ be a flow map in topological space $X$. A point $x \in X$ is said to be a wandering point if there is a neighbourhood $U$ of $x$ and positive integer $N$ such that for all $k > N$, the flow map is pairwise disjoint as*

$$T^k(U) \cap U = \emptyset. \tag{9}$$

**Remark B.2** *Wandering sets are transient. When a dynamical system has a wandering set of non-zero measures, then the system is regarded as a dissipative system. This is the opposite of the conservative system in Poincare's claim. An intuitive way to understand the wandering set is: if a portion of phase space "wanders away" during system evolution, and never visits again, then the system is dissipative. A simple example is Lorenz 63, it will contract to the global attractor without visiting the outside again. When the global attractor exists, the invariant set can be expressed as $\omega-limit$ set as $\omega(x, T) = \bigcap_{n=1}^{\infty} \overline{\bigcup_{k=n}^{\infty} \{T^k(x)\}}$. The invariant set is thus non-wandering.*

**Theorem B.3 (Liouville's Theorem in Hamiltonian (Agarwal, 2007))** *Consider a Hamiltonian dynamical system with canonical coordinates $q_i = (q_i^1, q_i^2, q_i^3)$ and momenta $p_i = (p_i^1, p_i^2, p_i^3)$ of $i-th$ molecule for $i = 1, \ldots, N$. Then the phase space distribution $\mu(p, q)$ determines the probability $\mu(p, q)d^n q d^n p$ that the system will be found in the infinitesimal phase space volume $d^n q d^n p$. The Liouville equation governs the evolution of $\mu(p, q; t)$ in time $t$ as*

$$\frac{d\mu}{dt} = \frac{\partial \mu}{\partial t} + \sum_{i=1}^{N} \left( \frac{\partial \mu}{\partial q_i} \dot{q}_i + \frac{\partial \mu}{\partial p_i} \dot{p}_i \right) = 0. \tag{10}$$

*Or we can say the distribution function is constant along any trajectory in phase space.*

**Remark B.4** *In Hamiltonian mechanics, the phase space is a smooth manifold that comes naturally equipped with a smooth measure (locally, this measure is the $6N$-dimensional Lebesgue measure). The theorem says this smooth measure is invariant under the Hamiltonian flow due to the preservation properties of the symplectic form. By the language in ergodic theory, we can assert that the $T : X \to X$ is a continuous conservative flow such that $\mu(E) = \mu(T^k E)$ for any $E \subset X$ and $k \in \mathbb{N}$.*

## C  PROOFS OF MAIN THEOREMS

### C.1  PROOF OF POINCARÉ'S CLAIM

**Theorem C.1 (Poincaré's Claim)** *Imagine a box filled with gas made of $N$ identical molecules. Classical mechanics says that if we know the positions $q_i = (q_i^1, q_i^2, q_i^3)$ and momenta $p_i = (p_i^1, p_i^2, p_i^3)$ of $i-th$ molecule for $i = 1, \ldots, N$, the positions and momenta of each particles at time $t$ determines by the Hamiltonian's equations as*

$$\dot{p}_i^j(t) = \frac{\partial H}{\partial q_i^j},$$
$$\dot{q}_i^j(t) = \frac{\partial H}{\partial p_i^j}, \tag{11}$$

*where $H(q_1, \ldots, q_N, p_1, \ldots, p_N)$ is the Hamiltonian, measuring the total energy of the system.*

*The state of entire system is denoted as $(q, p) = (q_1, \ldots, q_N, p_1, \ldots, p_N)$. Let $X$ denote the collection of all possible states. If the Hamiltonian is bounded above, then the transformation of state can be defined by a map as*

$$T_t : (q, p) \rightarrow (q(t), p(t)), \tag{12}$$

*where $T_t$ push the state move towards after $t$ steps. When the system is regular, there exist $6N$ equations as shown in $Equation\,11$ coupled together. The problem can be regarded as an initial value problem, and $T$ satisfies the conservative law for the Hamiltonian system, which means $x(0) \in X \Rightarrow T_t(x(0)) \in X$ for all $t$.*

*When the system with enormous molecules, Poincaré claimed that the system starts at a certain state $(q(0), p(0))$, it will eventually return to a state close to $(q(0), p(0))$ after a long enough time.*

*Proof.* Here is Poincaré's solution. Definite $T := T_1$ and $T^k = T_k$. Fix $\epsilon > 0$ and consider the set of $W$ of all states $x = (q, p)$ such that $d(x, T^k(x)) > \epsilon$ for all $k > 1$, where $d(\cdot, \cdot)$ is the Euclidean distance. Divide $W$ into finitely many disjoint pieces as $W_i$ of diameter less than $\epsilon$.

For each fixed $i$, the sets $T^{-k}(W_i)(k \geq 1)$ are pairwise disjoint, otherwise we can derive that $T^{-n}(W_i) \cap T^{-n-k}(W_i) \neq \emptyset \Rightarrow W_i \cap T^{-k}(W_i) \neq \emptyset$. In such a situation, this leads to a contradiction

- $x \in T^{-k}(W_i)$ implies that $T^k(x) \in W_i$, whence $d(x, T^k x) \leq diam(W_i) < \epsilon$,
- $x \in W_i \subset W$ implies that $d(x, T^k x) > \epsilon$ by the initial settings of $W$.

So $T^{-k}(W_i)$ must be pairwise disjoint.

Since $\{T^{-k}W_i\}_{k \geq 1}$ are pairwise disjoint, $\mu(X) \geq \sum_{k \geq 1} \mu(T^{-k}W_i)$. However, by the Liouville's theorem in statistical mechanics in B.3 Agarwal (2007) [4], all $T^{-k}(W_i)$ have the same measure, and $\mu(X) < \infty$, so a natural result must be $\mu(W_i) = 0$ for all $W_i$. The summation of all $\mu(\bigcup_{i \in \mathbb{N}} W_i) = 0$ means the measure zero of *wandering set* (Feldman, 2019). In summary, we can infer $\mu(X \setminus \bigcup_{i \in \mathbb{N}} W_i) = 1$, and the system will explore sufficiently for all possible set $X \setminus \bigcup_{i \in \mathbb{N}} W_i$. Consequently, we have the property as $d(T^k(x), x) < \epsilon$ for $k \geq 1$.

**Proposition C.2** *Suppose $(X, \mathcal{B}, \mu, T)$ is an MPT on a complete measure space, then the following are equivalent (Cornfeld et al., 2012):*

- *$\mu$ is ergodic;*

- *if $E \in \mathcal{B}$ and $\mu(T^{-1}E \Delta E) = 0$ [5],then $\mu(E) = 0$ or $\mu(X \setminus E) = 0$;*

- *$f : X \rightarrow \mathbb{R}$ is a measurable function and $f \circ T = f$ almost everywhere, then there is a constant $c \in \mathbb{R}$, s.t. $f = c$ almost everywhere.*

**Proposition C.3** *Mixing property implies ergodicity.*

*Proof.* Suppose $E$ is invariant, then the mixing property and measure preserving property imply $\mu(E) = \lim_{k \to \infty} \mu(E \cap T^{-k}E) \rightarrow \mu(E)^2$, whence $\mu(E)^2 = \mu(E)$. It follows that $\mu(E) = 0$ or $\mu(E) = 1 = \mu(X)$. This result reflects the second item in C.2, and thus ergodicity.

**Proposition C.4** *A MPT $(X, \mathcal{B}, \mu, T)$ is strongly mixing iff for every $\phi, \psi \in L^2$, $\lim_{k \to \infty} \int \phi \psi \circ T^k d\mu \rightarrow \int \phi d\mu \int \psi d\mu$, or equivalent to say $\lim_{k \to \infty} Cov(\phi, \psi \circ T^k) \rightarrow 0$.*

*Proof.* The following fact can be easily derived as

- Fact 1. Since $\mu \circ T^{-1} = \mu$, $\|\phi \circ T\|_2 = \|\phi\|_2$, for all $\phi \in \mathcal{L}^2$;
- Fact 2. $|Cov(\phi, \psi)| \leq 4\|\phi - \int \phi\|_2 \|\psi - \int \psi\|_2$;

---

[4]The Liouville equation governs the evolution of probability space is invariant under the flow map as $\mu(T_t E) = \mu(E)$ for $E \subset X$.

[5]$\Delta$ is the symmetric set difference such that $A\Delta B = (A \setminus B) \cup (B \setminus A)$.

- Fact 3. The function $Cov(\phi, \psi)$ is bilinear.

The proof of proposition needs to indicate it as a necessary and sufficient condition.

(Necessary.) The condition that for $\phi, \psi \in \mathcal{L}^2$, the mixing result reflects that $\phi, \psi$ can be regarded as the indicator function as $\phi = 1_E$ and $\psi = 1_F$. By the property, we have $\lim_{k \to \infty} Cov(\phi, \psi \circ T^k) \to 0$ according to the mixing property. Consequently, the necessity is proved.

(Sufficient.) Let $\phi, \psi$ be the linear combination of indicator functions on the invariant set. Generally $\phi, \psi \in \mathcal{L}^2$, giving a small perturbation $\epsilon > 0$, the finite linear combinations of indicators become $\phi_\epsilon$, $\psi_\epsilon$, satisfying $\|\phi - \phi_\epsilon\|_2, \|\psi - \psi_\epsilon\|_2 < \epsilon$. Then, by the analysis tricks as

$$
\begin{aligned}
&\lim_{k \to \infty} |Cov(\phi, \psi \circ T^k)| \\
&\leq \lim_{k \to \infty} |Cov(\phi - \phi_\epsilon, \psi_\epsilon \circ T^k)| + \lim_{k \to \infty} |Cov(\phi_\epsilon, \psi_\epsilon \circ T^k)| + \lim_{k \to \infty} |Cov(\phi_\epsilon, (\psi_\epsilon - \psi) \circ T^k)| \\
&\leq 4\epsilon\|\psi\|_2 + o(1) + 4(\|\phi\|_2 + \epsilon)\epsilon,
\end{aligned}
\tag{13}
$$

where the result tells us $\lim_{k \to \infty} Cov(\phi, \psi \circ T^k) \leq 4\epsilon\|\psi\|_2 + o(1) + 4(\|\phi\|_2 + \epsilon)\epsilon$ according to the Fact 1, 2 and 3. When $\epsilon$ is sufficiently small, the limit becomes zero, the proof is finished.

## C.2 PROOF OF 3.1

Since $\{\phi_1, \phi_2, \dots\}$ is an orthonormal basis of $\mathcal{L}^2$, for $L$ is large enough, we have a projection map as $g_{en} := \Pi_L \phi$ approximates $\phi$ within $\epsilon-$bound as

$$
\|\phi_L - \phi\|_2 \leq \epsilon.
\tag{14}
$$

Since the forward operator $\mathcal{G}$ is locally Lipschitz bounded by a constant $c$, the following the inequality holds

$$
\sup_{z \in \mathcal{M}} \|\mathcal{G}\phi_L - \mathcal{G}\phi\|_2 < \sup_{z \in \mathcal{M}} c\|\phi_L(z) - \phi(z)\| < c\epsilon.
\tag{15}
$$

Now, $\phi_L$ is a function lying the space of $\mathcal{L}^2$ since it is a linear combination of $\{\phi_i\}_{i \in I}$. The encoder map $g_{\theta_1}^{en}$ learns the adaptive feature from the dataset. When the dataset is sufficiently large, there exists a map as

$$
g_{\theta_1}^{en} = \phi_L,
\tag{16}
$$

where $g_{en}$ spans a finite-dimensional linear space, the relation holds due to the universal property of neural networks (Voigtlaender, 2023). In such case the learned operator $\hat{\mathcal{G}}$ can be written as

$$
\hat{\mathcal{G}} = \Pi_L \mathcal{G} \Pi_L,
\tag{17}
$$

where the finite representation of the operator $\hat{\mathcal{G}}$ is consistent with $\mathcal{G}_L = \Pi_L \mathcal{G} \Pi_L$. Consequently, we have the inequality as

$$
\begin{aligned}
&\sup_{z \in \mathcal{M}} \|\hat{\mathcal{G}} \circ g_{\theta_1}^{en}(z) - \mathcal{G} \circ \phi(z)\|_2 \\
&\leq \underbrace{\sup_{z \in \mathcal{M}} \|\hat{\mathcal{G}} \circ g_{\theta_1}^{en}(z) - \hat{\mathcal{G}} \circ \phi(z)\|_2}_{\text{strongly continuous property}} + \underbrace{\sup_{z \in \mathcal{M}} \|\hat{\mathcal{G}} \circ \phi(z) - \mathcal{G} \circ \phi(z)\|_2}_{\text{condition of strong operator topology}} \\
&\leq (c+1)\epsilon.
\end{aligned}
\tag{18}
$$

The second line is from the triangle inequality. The third line holds due to the strongly continuous operator[6] and strong operator topology[7] in Hilbert space (see D.1), then uniform limit of some sequence of finite-rank operators exists. Therefore, we can assert that existing a finite rank $L \in \mathbb{N}$ to approximate the operator $\mathcal{G}$ with arbitrary small error.

---

[6]1. Suppose $\{x_n\}$ a sequence of elements in $\mathcal{L}^2$ that converges to $x$ in the norm of the space, i.e., $\lim_{n \to \infty} \|x_n - x\| \to 0$. The strong continuity of $\mathcal{G}$ implies that the operator $\mathcal{G}$ is continuous with respect to the norm topology when applied to sequences of vectors. Therefore, it must hold that $\lim_{n \to \infty} \|\mathcal{G}x_n - \mathcal{G}x\| \to 0$.

[7]Let $\mathfrak{L}(X)$ be the space of all bounded, linear operators on Hilbert space $X$. A net $\{\mathcal{G}_n\} \subset \mathfrak{L}(X)$ converges pointwise to $\mathcal{G} \in \mathfrak{L}(X)$ on $(X, \|\cdot\|)$ (i.e., $\lim_{n \to \infty} \|\mathcal{G}_n x - \mathcal{G}x\| \to 0, \ \forall x \in X$) if and only if $\{\mathcal{G}_n\}$ converges to $\mathcal{G}$ in the strong operator topology on $\mathfrak{L}(X)$.

## C.3 PROOF OF 3.2

(Unitary property.) The unitary of $\mathcal{G}$ follows from the measure-preserving property of $T$. For $\phi, \psi \in \mathcal{L}^2$, we have

$$
\begin{aligned}
&\langle \mathcal{G}\phi, \mathcal{G}\psi \rangle \\
&= \int_X (\mathcal{G} \circ \phi(x)) \overline{(\mathcal{G} \circ \psi(x))} d\mu \\
&= \int_X (\phi(Tx)) \overline{(\psi(Tx))} d\mu(x)
\end{aligned}
\tag{19}
$$

Using the change of variable $y = Tx$ and $d\mu(T^{-1}(X)) = d\mu(X)$, we have

$$
\int_X \phi(y) \overline{\psi(y)} d\mu(y) = \langle \phi, \psi \rangle.
\tag{20}
$$

A more interesting way to prove the unitarity of $\mathcal{G}$ comes from the perspective of quantum mechanics (Gyamfi, 2020). In measure-conserving systems, the Liouville operator is skew-Hermitian (as seen in the discussion of the Liouville operator and Poisson brackets in (Karasev et al., 2012)), which implies that the operator $\mathcal{G}$ is unitary.

## C.4 PROOF OF 3.3

Fact from 3.2 - the MPT property of $T$ implies the isometric transformation as $\|\phi\|_2 = \|\phi \circ T\|_2$ for all $\phi \in \mathcal{L}^2$.

This convergence of MPT can be proved by the coboundary property [8] such that the function $\phi$ can be set as $\phi = \psi - \psi \circ T$ (both $\phi, \psi \in \mathcal{L}^2$) (such function $\psi$ can be regarded as transfer function from the perspective of cohomology in Tao (2008b)), this is easy to see

$$
\begin{aligned}
&\lim_{N \to \infty} \left\| \frac{1}{N} \sum_{k=0}^{N-1} \phi \circ T^k \right\|_2 \\
&= \lim_{N \to \infty} \left\| \frac{1}{N} \sum_{k=0}^{N-1} (\psi \circ T^{k-1} - \psi \circ T^k) \right\|_2 \\
&= \lim_{N \to \infty} \frac{1}{N} \left\| \psi - \psi \circ T^{N-1} \right\|_2 \\
&\leq \lim_{N \to \infty} \frac{2}{N} \left\| \psi \right\|_2 \to 0.
\end{aligned}
\tag{21}
$$

Consequently,

$$
\lim_{N \to \infty} \frac{1}{N} \sum_{k=0}^{N-1} \mathcal{G}^k \phi \to \overline{\phi}.
\tag{22}
$$

Thus it derived if holds for the element of subspace $\{\psi - \psi \circ T \mid \psi \in \mathcal{L}^2\}$. Then by the analysis tricks, choose a arbitrary function $\chi \in \{\psi - \psi \circ T \mid \psi \in \mathcal{L}^2\}$ satisfying $\|\chi - \phi\|_2 < \epsilon$. For the sufficiently large $N$ we have

$$
\begin{aligned}
\lim_{N \to \infty} \left\| \frac{1}{N} \sum_{k=0}^{N-1} \phi \circ T^k \right\|_2 &\leq \lim_{N \to \infty} \left\| \frac{1}{N} \sum_{k=0}^{N-1} (\phi - \chi) \circ T^k \right\|_2 + \left\| \frac{1}{N} \sum_{k=0}^{N-1} \chi \circ T^k \right\|_2 \\
&\leq \lim_{N \to \infty} \left\| \frac{1}{N} \sum_{k=0}^{N-1} (\phi - \chi) \circ T^k \right\|_2 + \epsilon < 2\epsilon
\end{aligned}
\tag{23}
$$

The proof of the first part is completed.

---

[8] **Coboundary of a dynamical system.** This concept is developed from cohomology theory (an interesting interpretation can be found in (Tao, 2008b)). Two functions $f, g : X \to \mathbb{C}$ are said to be cohomology via a transfer function $h$, if $f = g + h - h \circ T$. A function which is cohomologous to zero is called a coboundary.

In particular, $\bar{\phi}$ is invariant. If $T$ is ergodic, $\bar{\phi}$ must be constant and $\bar{\phi} = \int \bar{\phi} d\mu$ almost surely. Also, since $\lim_{N \to \infty} \frac{1}{N} \sum_{k}^{N-1} \mathcal{G}^k \phi \to \bar{\phi}$, we have

$$\int \phi d\mu = \lim_{N \to \infty} \frac{1}{N} \sum_{k=1}^{N-1} \langle 1, \phi \circ T \rangle = \langle 1, \lim_{N \to \infty} \frac{1}{N} \sum_{k=0}^{N-1} \phi \circ T^k \rangle \to \langle 1, \bar{\phi} \rangle \to \int \bar{\phi} d\mu \qquad (24)$$

The proof is different from the original proof in Neumann (1932b), Von Neumann started the proof by factorizing the spectrum using Fourier analysis and Borel functional calculus and derived the only non-trivial invariant subspaces are those consisting of constant functions.

# D   ANALYSIS ON OPERATOR

Contraction and unitary operators in Hilbert spaces are key tools for predicting and understanding chaotic systems in this paper. To make our work self-contained, we will provide a clear and formal analysis of these two important types of operators. Generally, the analysis on the one-parameter group is on Banach space. Since our paper is specifically defined on $\mathcal{L}^2$ space, we restrict our definitions and analysis to Hilbert space.

**Definition D.1** ($C_0-$**semigroup (Engel et al., 2000)**) *A strongly continuous one-parameter semigroup on a Hilbert space $\mathcal{L}^2$ is a family $\mathcal{G} = \{\mathcal{G}(t) \mid t \in \mathbb{R}_+\}$ of bounded linear operators $\mathcal{G}(t)$ on $\mathcal{L}^2$ satisfying*

- *$\mathcal{G}(0) = I_d$, (the identity operator on $\mathcal{L}^2$)*

- *$\forall t, s \in \mathbb{R}_+ : \mathcal{G}(t + s) = \mathcal{G}(t)\mathcal{G}(s)$,*

- *$\lim_{t \to t_0} \mathcal{G}(t)\phi = \mathcal{G}(t_0)\phi$ for $t_0 \in \mathbb{R}_+$, $\phi \in \mathcal{L}^2$.*

*The first two axioms are algebraic, and state that $\mathcal{G}$ is a representation of the semigroup $(\mathbb{R}_+, +)$; the last is topological, and states that the map $\mathcal{G}$ is continuous in the strong operator topology.*

*If, in addition, $\|\mathcal{G}(t)\| \leq 1$ for all $t \in \mathbb{R}_+$, then $\mathcal{G}$ is called a strongly continuous one-parameter semigroup of contractions (or contraction operator).*

In our paper, our target is to learn continuous linear operators on Hilbert space, which belong to $C_0-$semigroup.

**Definition D.2 (Infinitesimal Generator (Engel et al., 2000))** *The infinitesimal generator of a strongly continuous one-parameter semigroup is the linear operator $P$ on $\mathcal{L}^2$ defined by*

$$P\phi = \lim_{t \to 0} \frac{1}{t}(\mathcal{G}(t) - I_d)\phi, \tag{25}$$

$$\mathcal{D}(P) = \left\{\phi \in \mathcal{L}^2 \mid \lim_{t \to 0} \tfrac{1}{t}(\mathcal{G}(t) - I_d)\phi \text{ exists}\right\}.$$

The strongly continuous semigroup $\mathcal{G}(t)$ with generator $P$ is often denoted by the symbol $e^{tP}$. This notation is compatible with the notation of matrix exponentials. Generally, directly approximating the operator $P$ is more difficult than the operator $\mathcal{G}$, since operator $P$ is usually unbounded and is defined on a dense subspace of $\mathcal{L}^2$, see the following proposition.

**Proposition D.3 ((Schmüdgen, 2012))** *The infinitesimal generator $P$ is a densely defined closed operator on $\mathcal{L}^2$ which determines the strongly continuous one-parameter semigroup $\mathcal{G}$ uniquely. We have*

$$\frac{d}{dt}\mathcal{G}(t)\phi = P\mathcal{G}(t)\phi, \quad \phi \in \mathcal{D}(P) \text{ and } t \in \mathbb{R}_+. \tag{26}$$

## D.1   THEORETICAL ASPECTS OF KOOPMAN OPERATOR

The Koopman operator $\mathcal{G}(t)$ (Koopman, 1931; Das et al., 2021; Ikeda et al., 2022; Cheng et al., 2023) acts on a function space by composing with the forward map $T$ (also known as the flow map), effectively implementing time shifts in the function $\phi$. Various choices exist for the function space, such as $\mathcal{L}^2$ and spaces of continuous functions. Specifically, for a function $\phi \in \mathcal{L}^2$ and time $t \in \mathbb{R}$, the Koopman operator $\mathcal{G}(t) : \mathcal{L}^2 \to \mathcal{L}^2$ is defined as:

$$\mathcal{G}(t)\phi = \phi \circ T. \tag{27}$$

According to Definition D.1, $\mathcal{G}(t)$ forms a semigroup.

In general, if $T$ is a $C^{k}$[9] flow for some $k \geq 0$, then $\mathcal{G}(t)$ maps the space $C^r$ into itself for every $0 \leq r \leq k$. The infinitesimal generator $P$ of the Koopman operator $\mathcal{G}(t)$ is defined by:

$$Pg := \lim_{t \to 0} \frac{1}{t}\left(\mathcal{G}(t)\phi - \phi\right), \quad \phi \in \mathcal{D}(P), \tag{28}$$

---

[9]$C^k$ denotes the function space with $k$-th order differentiability.

where $\mathcal{D}(P) \subseteq C^1 \cap \mathcal{L}^2$ is the domain of $P$, consisting of functions for which this limit exists. Typically, $P : C^1 \cap \mathcal{L}^2 \to C^0 \cap \mathcal{L}^2$ when $T$ is smooth. However, merely considering the semigroup property of the Koopman operator is insufficient for modeling the complex behavior of dissipative chaotic systems. To account for the intrinsic physical properties of such systems, we incorporate this physical knowledge into the learning framework. Specifically, we analyze two intrinsic physical properties from spectral theory—contraction and unitarity—as detailed in Section D.4 and Appendix D.3.

## D.2 Contraction Operator on Hilbert Space

**Definition D.4 ((Schmüdgen, 2012))** *We shall say that the operator $B$ is accretive if $Re(\langle B\phi, \phi \rangle) \geq 0$ for all $\phi \in \mathcal{D}(B)$ that $B$ is $m-$accretive if $B$ is closed, acceretive, and $R(B - \lambda_0 I)$ is dense in $\mathcal{L}^2$ for some $\lambda_0 \in \mathbb{C}$, $Re(\lambda_0) < 0$ [10].*

*$B$ is called dissipative if $-B$ is accretive and $m-$dissipative if $-B$ is $m-$accretive.*

**Theorem D.5 (Lumer–Phillips theorem in Hilbert space)** *A linear operator $P$ on a Hilbert space $\mathcal{L}^2$ is the generator of a strongly continuous one-parameter contraction semigroup if and only if $P$ is $m-$dissipative, or equivalently, $-P$ is m-accretive.*

Proof. If $P$ is the generator of a contraction semigroup $\mathcal{G}$, then in particular $P$ is dissipative, and hence $Re(\langle P\phi, \phi \rangle)$ for $\phi \in \mathcal{L}^2$ by D.4. Thus, the latter fact can be easily derived directly. Indeed, using that $\mathcal{G}(t)$ is a contraction in D.1, we obtain

$$Re(\langle (\mathcal{G}(t) - I_d)\phi, \phi \rangle) = Re(\langle \mathcal{G}(t)\phi, \phi \rangle) - \|\phi\|^2 \leq \|\mathcal{G}(t)\phi\| \|\phi\| - \|\phi\|^2 \leq 0 \qquad (29)$$

Dividing by $t \to 0$ and passing to the limit $t \to 0^+$, we get $\langle P\phi, \phi \rangle \leq 0$.

Combining the D.3 and D.5 states the eigenvalues of infinitesimal generator $P := \lim_{t \to 0^+} \frac{\mathcal{G}_t \phi - \phi}{t}$ of contraction operator $\mathcal{G}$ has non-positive real parts. However, the operator $P$ is usually unbounded and is defined on a dense subspace of $\mathcal{L}^2$ (Engel et al., 2000). Directly learning the infinitesimal generator $P$ can be more difficult than working with operator $\mathcal{G}$. To simplify the problem, we can consider eigenvalues of $\mathcal{G}$ within the unit disk in the complex plane during the contraction stage see Equation 6.

**Spectral Analysis on $\mathcal{G}^T\mathcal{G}$.** We define the operator norm as $\|\mathcal{G}\| := \sup_{\phi \in \mathcal{L}^2} \|\mathcal{G}\phi\|_2$. Given that $\mathcal{G}$ is contraction, we know

$$\|\mathcal{G}\| \leq 1 \quad \Rightarrow \quad \|\mathcal{G}^T\mathcal{G}\| \leq \|\mathcal{G}\|^2 \leq 1. \qquad (30)$$

Since $\mathcal{G}^T\mathcal{G}$ is a positive operator (because $\mathcal{G}^T\mathcal{G}$ is self-adjoint and non-negative), the norm $\|\mathcal{G}^T\mathcal{G}\|$ is equal to the spectral radius of $\mathcal{G}^T\mathcal{G}$, and hence

$$\sigma(\mathcal{G}^T\mathcal{G}) \subset [0, \|\mathcal{G}^T\mathcal{G}\|] \subset [0, 1]. \qquad (31)$$

This show that $\mathcal{G}^T\mathcal{G}$ is also a contraction operator, since $\|\mathcal{G}^T\mathcal{G}\| \leq 1$.

## D.3 Unitary Operator

**Definition D.6 (Unitary Operator (Schmüdgen, 2012))** *A strongly continuous one-parameter unitary group briefly a unitary group, is a family $\mathcal{G} = \{\mathcal{G}(t) \mid t \in \mathbb{R}\}$ of unitaries $\mathcal{G}(t)$ on Hilbert space $\mathcal{L}^2$ such that*

- *$\mathcal{G}(t)\mathcal{G}(s) = \mathcal{G}(t + s)$ for all $t, s \in \mathbb{R}$,*

- *$\lim_{h \to 0} \mathcal{G}(t + h)\phi = \mathcal{G}(t)\phi$ for $\phi \in \mathcal{L}^2$ and $t \in \mathbb{R}$.*

Axiom (i) in D.6 means that $\mathcal{G}$ is group homomorphism of the additive group $\mathbb{R}$ into the group of unitary operators on $\mathcal{L}^2$. In particular, this implies that $\mathcal{G}(0) = I_d$ and

$$\mathcal{G}(-t) = \mathcal{G}^{-1} = \mathcal{G}^* \quad \forall t \in \mathbb{R}. \qquad (32)$$

---

[10] $Re(\cdot)$ is denoted as the real part, and $R(B)$ is the range of operator $B$.

Axiom (ii) in D.6 is a strong continuity of $\mathcal{G}$. It clearly suffices to require (ii) for $t = 0$ and for $\phi$ from a dense subset of $\mathcal{L}^2$. Since the operators $\mathcal{G}(t)$ are unitaries, it is even enough to assume that $\lim_{t \to 0} \langle \mathcal{G}(t)\phi, \phi \rangle = \langle \phi, \phi \rangle$ for $\phi$ from a dense subset of $\mathcal{L}^2$.

The generator $P$ of the one-parameter unitary group $\mathcal{G}(t)$ satisfying the following relationship

$$\mathcal{G}(t) = \exp(itP) \quad \forall t \in \mathbb{R}. \tag{33}$$

The operator $P$ governs the infinitesimal behavior of the group and is defined by the strong limit:

$$P\phi = \lim_{t \to 0} \frac{\mathcal{G}\phi - \phi}{it}, \tag{34}$$

for all $\phi$ in domain $\mathcal{D}(P)$ of $P$, which consists of those functionals for which limit exists.

Since $\mathcal{G}(t) = \exp(itP)$ where $P$ is skew-adjoint $P^* = -P$, the spectrum of $\mathcal{G}(t)$ must lie on the unit circle.

**Spectral Properties.** For all $\mathcal{G}(t), t \in \mathbb{R}$, its spectrum $\sigma(\mathcal{G}(t))$ lies on the complex unit circle $\mathbb{T}$, which is the set of complex numbers with absolute value 1

$$\sigma(\mathcal{G}(t)) \subset \{r \in \mathbb{C} \mid |r| = 1\}.$$

**Physical Interpretation of Two Operators.** During the contraction phase, changes in the probability distribution inherently reflect the asymmetry of entropy with respect to time. This phenomenon is explained by the second law of thermodynamics: the contraction operator, associated with energy dissipation. However, with an invariant measure, the system's evolution is governed by a unitary operator if it satisfies the Liouville equation (in the classical setting) or the Liouville-von Neumann equation (in the quantum setting) (see B.3), which makes the density matrix or the invariant measure time-independent (Tao, 2008a), respectively. Consequently, a unitary operator and its conjugate transpose can describe the forward and backward chaotic dynamics on the invariant set.

# E   ALGORITHM

---

**Algorithm 1** Poincaré Flow Neural Network

---

**PFNN: Training**

**Require:** Training data $\mathcal{D} = \{\{(s_t^i, s_{t+1}^i)\}_{t=0}^{T-1}\}_{i=0}^N$ with number of trajectories $N$ and trajectory length $T$, relaxation time $k$, training epochs $N_{\text{train}}$, learning rate $\alpha$, regularized coefficients $\gamma_1, \gamma_2$ encoder $g_{\theta_1}^{\text{en}} : \mathcal{M} \to \mathbb{R}^L$, decoder $g_{\theta_2}^{\text{de}} : \mathbb{R}^L \to \mathcal{M}$ with latent feature dimension $L$

(contraction phase) operator $\hat{\mathcal{G}}_c \in \mathbb{R}^{L \times L}$

(measure invariant phase) forward operator $\hat{\mathcal{G}}_m \in \mathbb{R}^{L \times L}$, backward operator $\hat{\mathcal{G}}_m^* \in \mathbb{R}^{L \times L}$

1: Separate the dataset $\mathcal{D}$ into contraction phase dataset $\mathcal{D}_c = \{\{(s_t^i, s_{t+1}^i)\}_{t=0}^{k-1}\}_{i=0}^N$ and measure invariant phase dataset $\mathcal{D}_m = \{\{(s_t^i, s_{t+1}^i)\}_{t=k}^{T-1}\}_{i=0}^N$.

2: **for** training epoch $e = 1, ..., N_{\text{train}}$ **do**

3:     Compute contraction loss $\mathcal{L}_{\text{contraction}}$ by Equation 6 with regularized coefficient $\gamma_1$ on $\mathcal{D}_c$.

4:         Update $\hat{\mathcal{G}}_c, \theta_1, \theta_2 -= \alpha \nabla_{\hat{\mathcal{G}}_c, \theta_1, \theta_2} \mathcal{L}_{\text{contraction}}$

5:     Compute measure invariant loss $\mathcal{L}_{\text{unitary}}$ by Equation 8 with regularized coefficient $\gamma_2$ on $\mathcal{D}_m$.

6:         Update $\hat{\mathcal{G}}_m, \hat{\mathcal{G}}_m^*, \theta_1, \theta_2 -= \alpha \nabla_{\hat{\mathcal{G}}_m, \hat{\mathcal{G}}_m^*, \theta_1, \theta_2} \mathcal{L}_{\text{unitary}}$

7: **end for**

8: **return** $g_{\theta_1}^{\text{en}}, g_{\theta_2}^{\text{de}}, \hat{\mathcal{G}}_c, \hat{\mathcal{G}}_m, \hat{\mathcal{G}}_m^*$

9:

**PFNN: Evaluating**

10: Given initial condition $s_0 \in \mathcal{M}$

11: **for** timestep $t = 1, ..., k$ **do**

12:     Predict using $\hat{\mathcal{G}}_c$, $g_{\theta_1}^{\text{en}}(\hat{s}_{t+1}) = \hat{\mathcal{G}}_c g_{\theta_1}^{\text{en}}(\hat{s}_t)$

13: **end for**

14: Collection $\mathcal{D}_{\text{statistics}} = \{\}$

15: **for** timestep $t > k$ **do**

16:     Predict using $\hat{\mathcal{G}}_m$, $g_{\theta_1}^{\text{en}}(\hat{s}_{t+1}) = \hat{\mathcal{G}}_m g_{\theta_1}^{\text{en}}(\hat{s}_t)$ and $\hat{s}_{t+1} = g_{\theta_2}^{\text{de}} \circ g_{\theta_1}^{\text{en}}(\hat{s}_{t+1})$

17:     $\mathcal{D}_{\text{statistics}} \cup \{\hat{s}_{t+1}\}$

18: **end for**

19: Estimate required long-term statistics $\{\mu, \sigma, ...\}$ using $\mathcal{D}_{\text{statistics}}$

20: **return** long-term statistics $\{\mu, \sigma, ...\}$

---

## F  MORE EXPERIMENTAL RESULTS

### F.1  LORENZ 63

The Lorenz 63 model (Lorenz, 1963), which consists of three coupled nonlinear ODEs,

$$\frac{dx}{dt} = \sigma(y - x), \ \frac{dy}{dt} = x(\rho - z) - y, \ \frac{dz}{dt} = xy - \beta z \tag{35}$$

used as a model for describing the motion of a fluid under certain conditions: an incompressible fluid between two plates perpendicular to the direction of the earth's gravitational force. In particular, the equations describe the rate of change of three quantities with respect to time: $x$ is proportional to the rate of convection, $y$ to the horizontal temperature variation, and $z$ to the vertical temperature variation. The constants $\sigma$, $\rho$, and $\beta$ are system parameters proportional to the Prandtl number, Rayleigh number, and coupling strength. In this paper, we take the classic choices $\sigma = 10$, $\rho = 28$, and $\beta = \frac{8}{3}$ which leads to a chaotic behavior with two strange attractors $(\sqrt{\beta(\rho - 1)}, \sqrt{\beta(\rho - 1)}, \rho - 1)$ and $-(\sqrt{\beta(\rho - 1)}, -\sqrt{\beta(\rho - 1)}, \rho - 1)$. Its state is $s = (x, y, z) \in \mathbb{R}^3$ bounded up and below from $\pm 30$.

Table 3: KL Divergence for long-term prediction distributions of Lorenz 63 system

| Lorenz 63 Components | KL Divergence | | | | |
|---|---|---|---|---|---|
| | FNO | LSTM | Koopman | MNO | PFNN |
| x | 0.5045 | $\infty$ | 0.8621 | 0.6263 | **0.2408** |
| y | 0.9339 | $\infty$ | 0.6975 | 0.3132 | **0.5155** |
| z | 0.3913 | $\infty$ | 0.4172 | 0.1793 | **0.1005** |

### F.2  LORENZ 96

This system was introduced in (Lorenz, 1996) as a low-order model of atmospheric circulation at a constant latitude circle. The system consists of $K$ variables $\mathbf{S} = (S_1, ..., S_K) \in \mathcal{S} \subseteq \mathbb{R}^K$, representing the values of atmospheric velocity measured along a circle of $K$ evenly spaced locations on the certain latitude of the earth. The governing equations are given by,

$$\frac{dS_k}{dt} = -S_{k-1}(S_{k-2} - S_{k+1}) - S_k + F, \tag{36}$$

where the parameter $F$ represents the forcing term. Here, the first term models advection, the second term represents linear damping, and F is an external forcing. We choose the set of variables of $K = \{9, 40, 80\}$ and the external forcing $F = 8$, parameters where the system is chaotic with the Lyapunov exponent is approximately $1.67$. Its dynamics exhibit strong energy-conserving nonlinearity, and for a large $F \geq 10$, it can exhibit strong chaotic turbulence and symbolizes the inherent unpredictability of the Earth's climate.

**Data generation**: we generated 1800 trajectories for training and 200 trajectories for testing, where each trajectory contains 2000 timesteps with integration time $0.01$ and a sample rate of 10. Meanwhile, in the best alignment with real scenarios, the initial conditions for trajectories in the training and testing set were drawn from a normal distribution.

**Experiment setup**: for PFNN models, when training the PFNN (consist) model, we discarded the initial 1000 timesteps to avoid the dissipative process; whilst for training the PFNN (contract) model, the initial steps were maintained to specialize the model in learning the dissipativity in the early-stage emulation. For all other models, the initial 100 timesteps were discarded. In the model prediction part, the PFNN (full) model autoregressively predicted the system states with the PFNN (contract) model for the beginning 900 timesteps, and then switched to PFNN (consist) model to

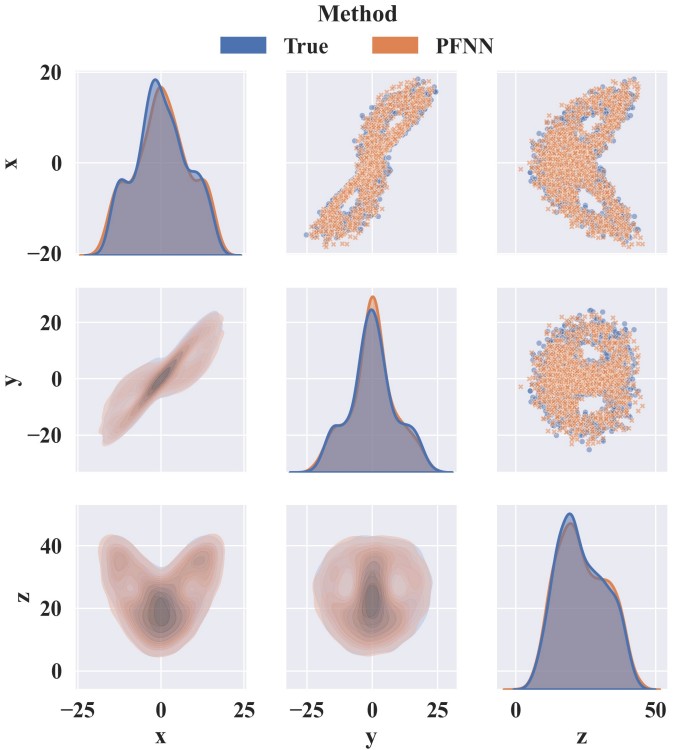

Figure 4: Visualization of long-term statistics of model predictions for Lorenz 63 system of 3 dimensions: we visualize the spatial correlation among 3 components of the velocity, focusing on evaluating the learned spatial correlation from the PFNN model's long-term predictions compared with the ground truth.

predict the states for the rest 1000 timesteps. For all other models, the trajectory for 1900 timesteps was predicted in the same autoregressive way.

### F.2.1 EXPERIMENT ON 9 DIMENSIONAL DATASET

Contents:

- Table of KL divergence (Table 4)
- Short-term prediction plot set (Figure 5)
- Long-term prediction plot set (Figure 6)

Table 4: KL Divergence for long-term prediction distributions by models across 3 principal components of Lorenz 96 system of 9-dimensional states

| Principle Components (PC) | KL Divergence | | | | |
|---|---|---|---|---|---|
| | FNO | LSTM | Koopman | MNO | PFNN |
| PC1 | 2.1491 | 2.2315 | 2.7091 | 1.8112 | **1.5920** |
| PC2 | 2.1152 | 2.6774 | 2.7133 | 2.1292 | **1.9539** |
| PC3 | 2.1090 | 5.5756 | 2.4214 | 2.0916 | **2.0637** |

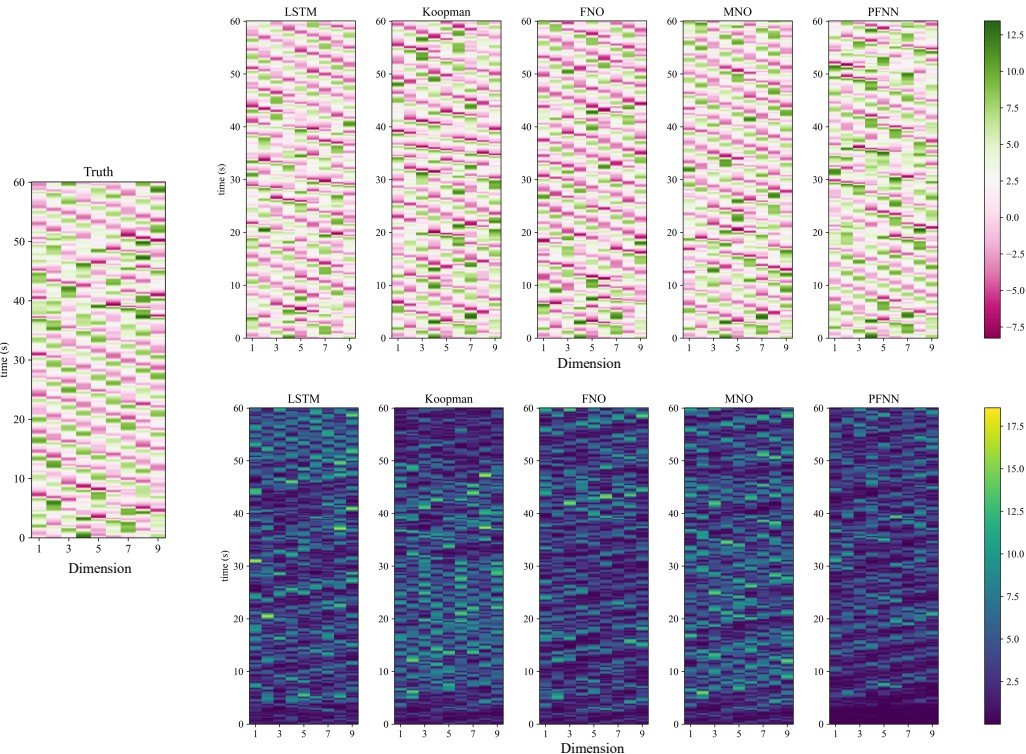

(a) Prediction visualization. The ground truth trajectory is visualized in the middle of the leftmost side. The predicted trajectories by baseline models and PFNN are shown in the first row. The corresponding absolute error trajectories of the predictions against the ground truth are shown in the second row.

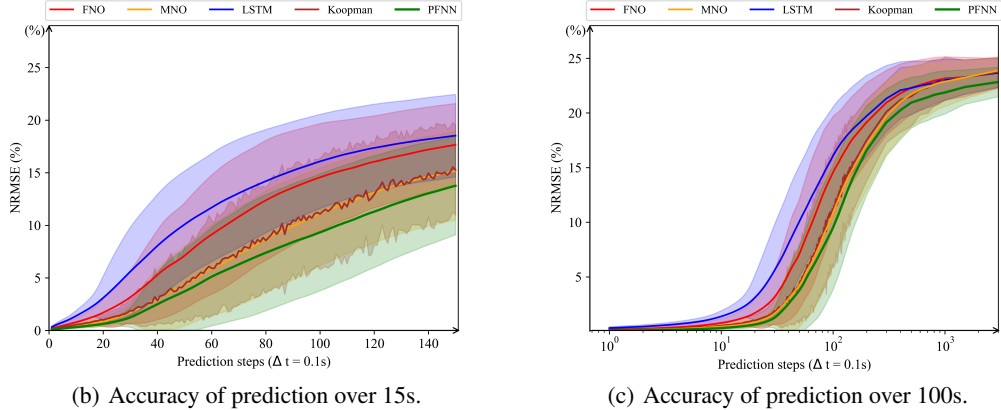

(b) Accuracy of prediction over 15s.        (c) Accuracy of prediction over 100s.

Figure 5: Visualization of prediction error in NRMSE: we visualize the comparison of model predictions of Lorenz 96 dynamics of 9 dimensions over 15 seconds (150 timesteps, short-term) and 100 seconds (1000 timesteps, mid-term).

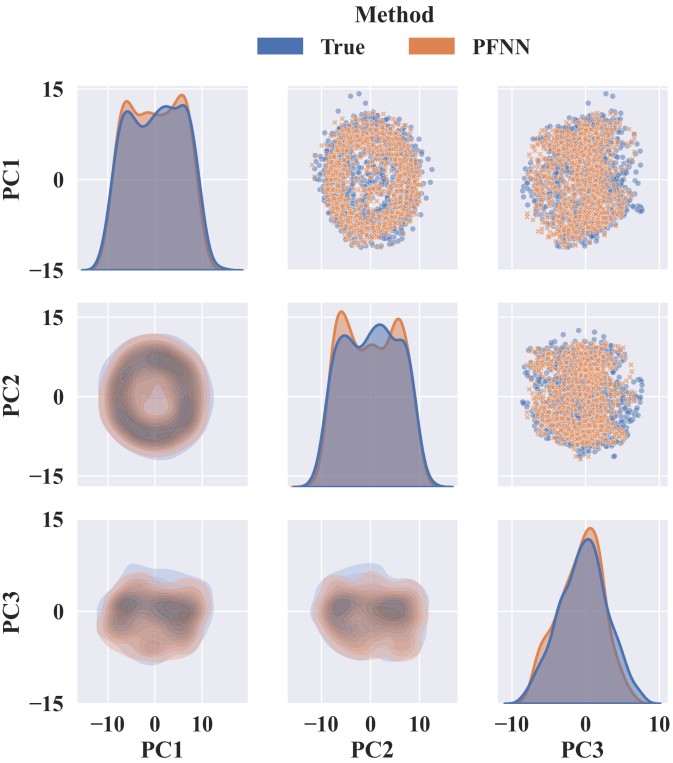

(a) Spatial correlation of PFNN in 3 principle components

Figure 6: Visualization of long-term statistics of model predictions for Lorenz 96 of 9 dimensions: we visualize the spatial correlation among 3 principle components of the velocity, focusing on evaluating the learned spatial correlation from the PFNN model's long-term predictions compared with the ground truth.

### F.2.2 EXPERIMENT ON 40 DIMENSIONAL DATASET

Contents:

- Table of KL divergence (Table 5)
- Short-term prediction plot set (Figure 7)
- Long-term prediction plot set (Figure 8)

| Principle Components (PC) | KL Divergence | | | | |
|---|---|---|---|---|---|
| | FNO | LSTM | Koopman | MNO | PFNN |
| PC1 | 0.1427 | 62659.3087 | 0.1723 | 0.0954 | **0.1097** |
| PC2 | 0.0987 | 62822.3591 | 0.1384 | 0.1009 | **0.0756** |
| PC3 | **0.0767** | 64132.3089 | 0.1475 | 0.1255 | 0.1315 |
| PC4 | 0.1309 | 63519.8891 | 0.1772 | 0.1382 | **0.1297** |
| PC5 | 0.1305 | 62346.3862 | 0.1647 | 0.1106 | **0.1056** |

Table 5: KL Divergence for long-term prediction distributions by models across 5 principal components of Lorenz 96 system of 40-dimensional states

### F.2.3 EXPERIMENT ON 80 DIMENSIONAL DATASET

Contents:

- Table of KL divergence (Table 6)
- Short-term prediction plot set (Figure 9)
- Long-term prediction plot set (Figure 10)

Table 6: KL Divergence for long-term prediction distributions by models across 5 principal components of Lorenz 96 system of 80-dimensional states

| Principle Components (PC) | KL Divergence | | | | |
|---|---|---|---|---|---|
| | FNO | LSTM | Koopman | MNO | PFNN |
| PC1 | 0.0785 | 1.3057 | 0.1650 | 0.2384 | **0.0733** |
| PC2 | 0.1045 | 1.3413 | 0.2477 | 0.2268 | **0.0801** |
| PC3 | 0.1546 | 1.4145 | 0.1613 | 0.0574 | **0.0524** |
| PC4 | 0.1600 | 1.2329 | 0.2324 | **0.0394** | 0.1220 |
| PC5 | 0.2451 | 1.2965 | 0.3733 | 0.3391 | **0.1724** |

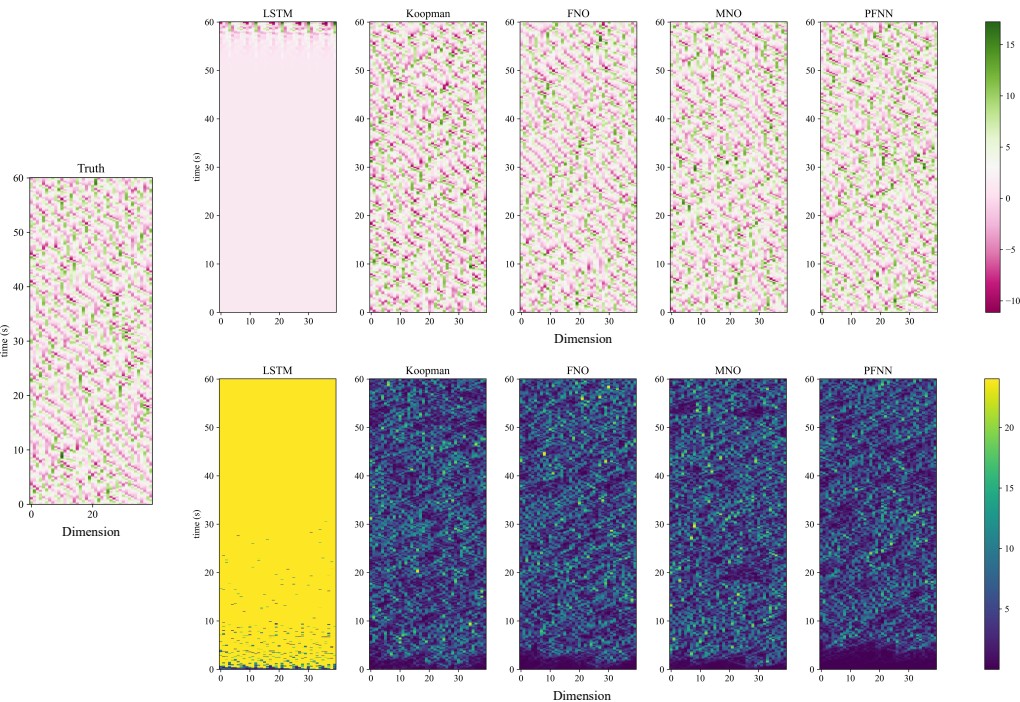

(a) Prediction visualization. The ground truth trajectory is visualized in the middle of the leftmost side. The predicted trajectories by baseline models and PFNN are shown in the first row. The corresponding absolute error trajectories of the predictions against the ground truth are shown in the second row.

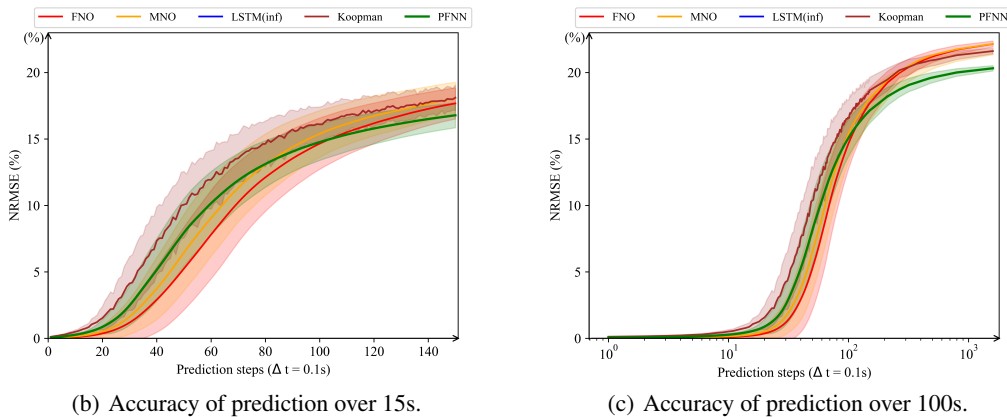

(b) Accuracy of prediction over 15s.

(c) Accuracy of prediction over 100s.

Figure 7: Visualization of prediction error in NRMSE: we visualize the comparison of model predictions of Lorenz 96 dynamics of 40 dimensions over 15 seconds (150 timesteps, short-term) and 100 seconds (1000 timesteps, mid-term).

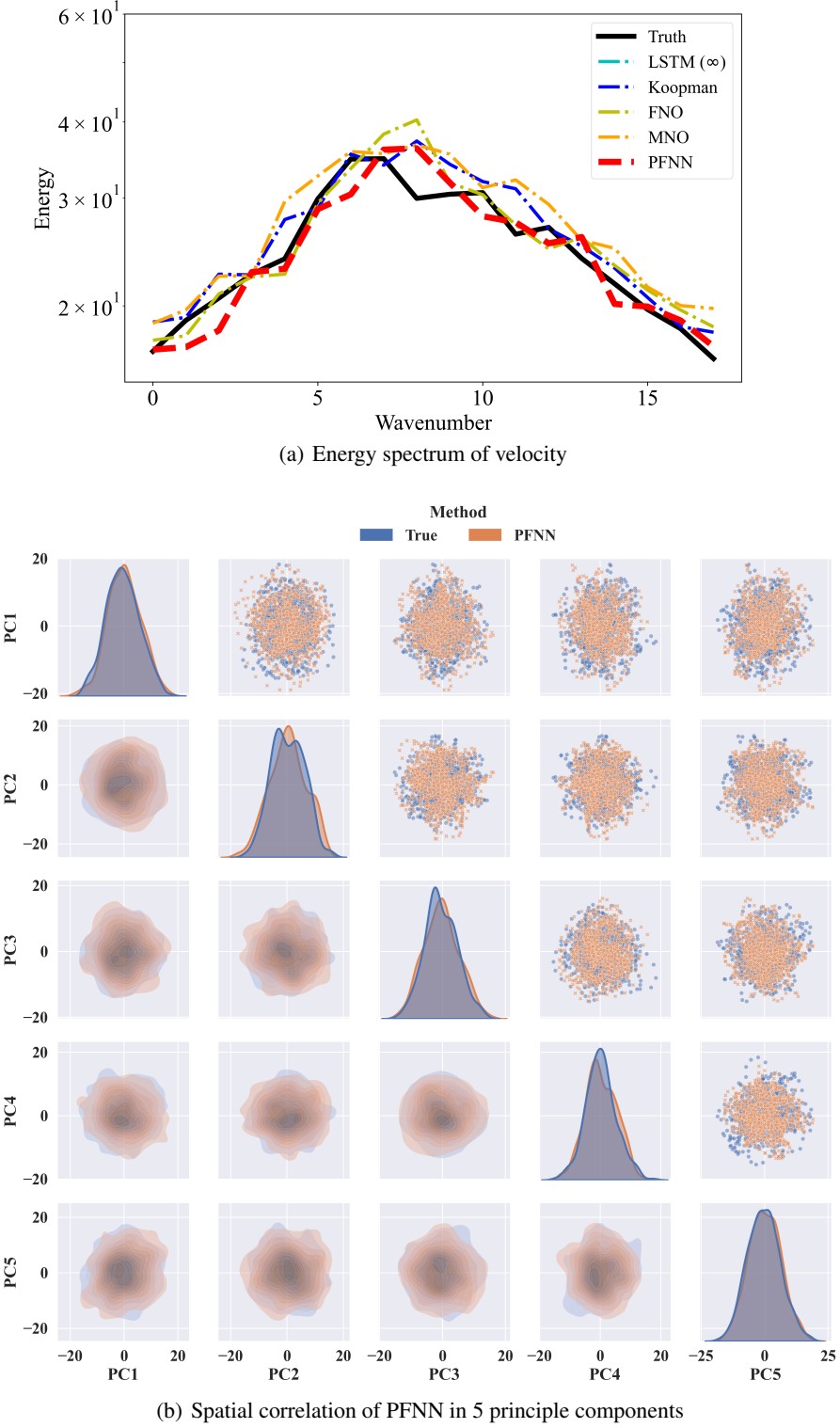

(a) Energy spectrum of velocity

(b) Spatial correlation of PFNN in 5 principle components

Figure 8: Visualization of long-term statistics of model predictions for Lorenz 96 system of 40 dimensions: we visualize density plots of each state dimension of the system velocity predicted by all six models; and then we visualize the spatial correlation among 5 principle components of the velocity, focusing on evaluating the learned spatial correlation from the PFNN model's long-term predictions compared with the ground truth.

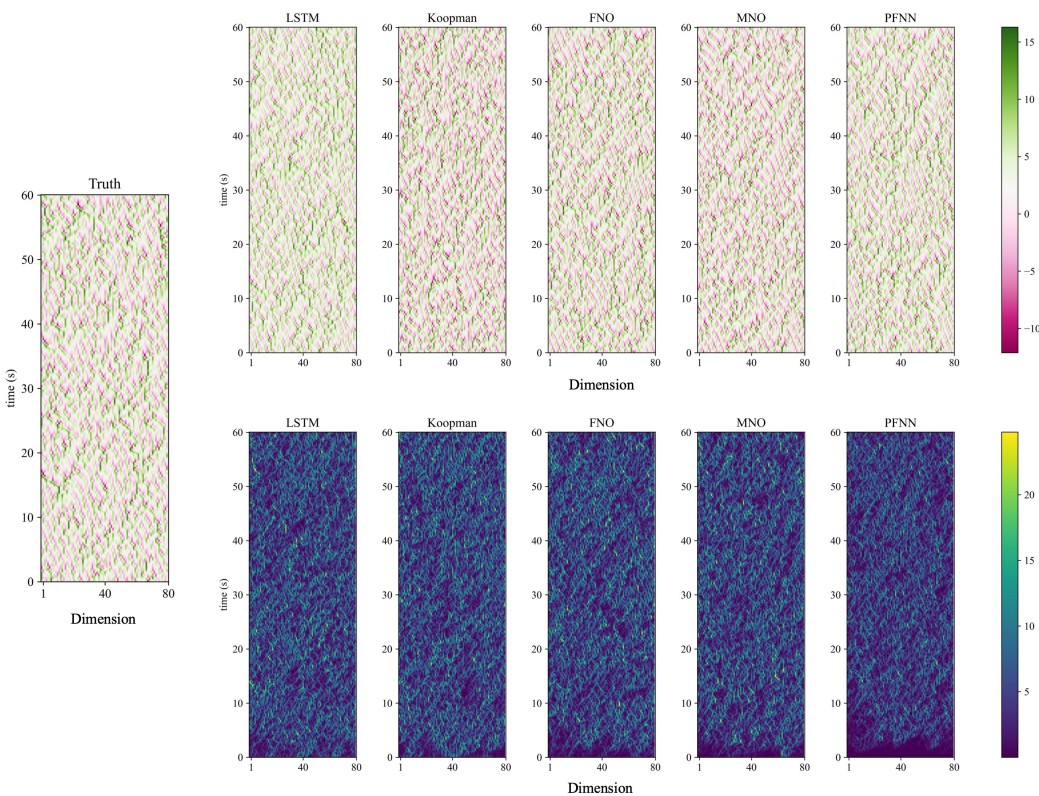

(a) Prediction visualization. The ground truth trajectory is visualized in the middle of the leftmost side. The predicted trajectories by baseline models and PFNN are shown in the first row. The corresponding absolute error trajectories of the predictions against the ground truth are shown in the second row.

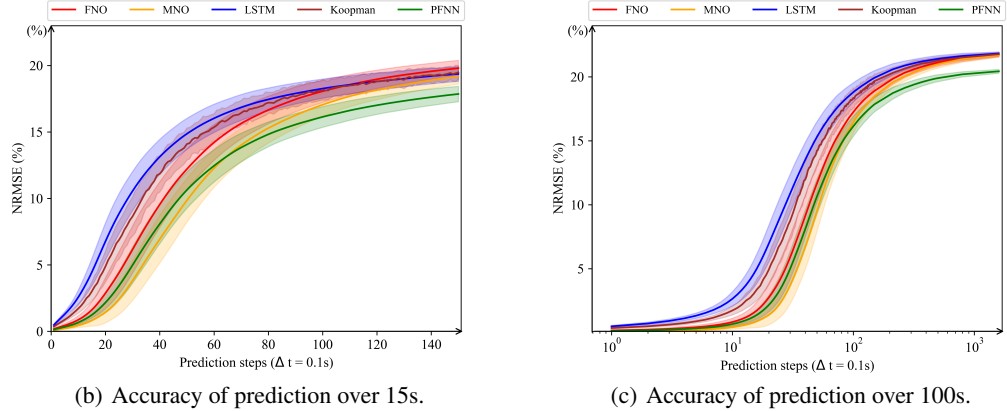

(b) Accuracy of prediction over 15s.

(c) Accuracy of prediction over 100s.

Figure 9: Visualization of prediction error in NRMSE: we visualize the comparison of model predictions of Lorenz 96 dynamics of 80 dimensions over 15 seconds (150 timesteps, short-term) and 100 seconds (1000 timesteps, mid-term).

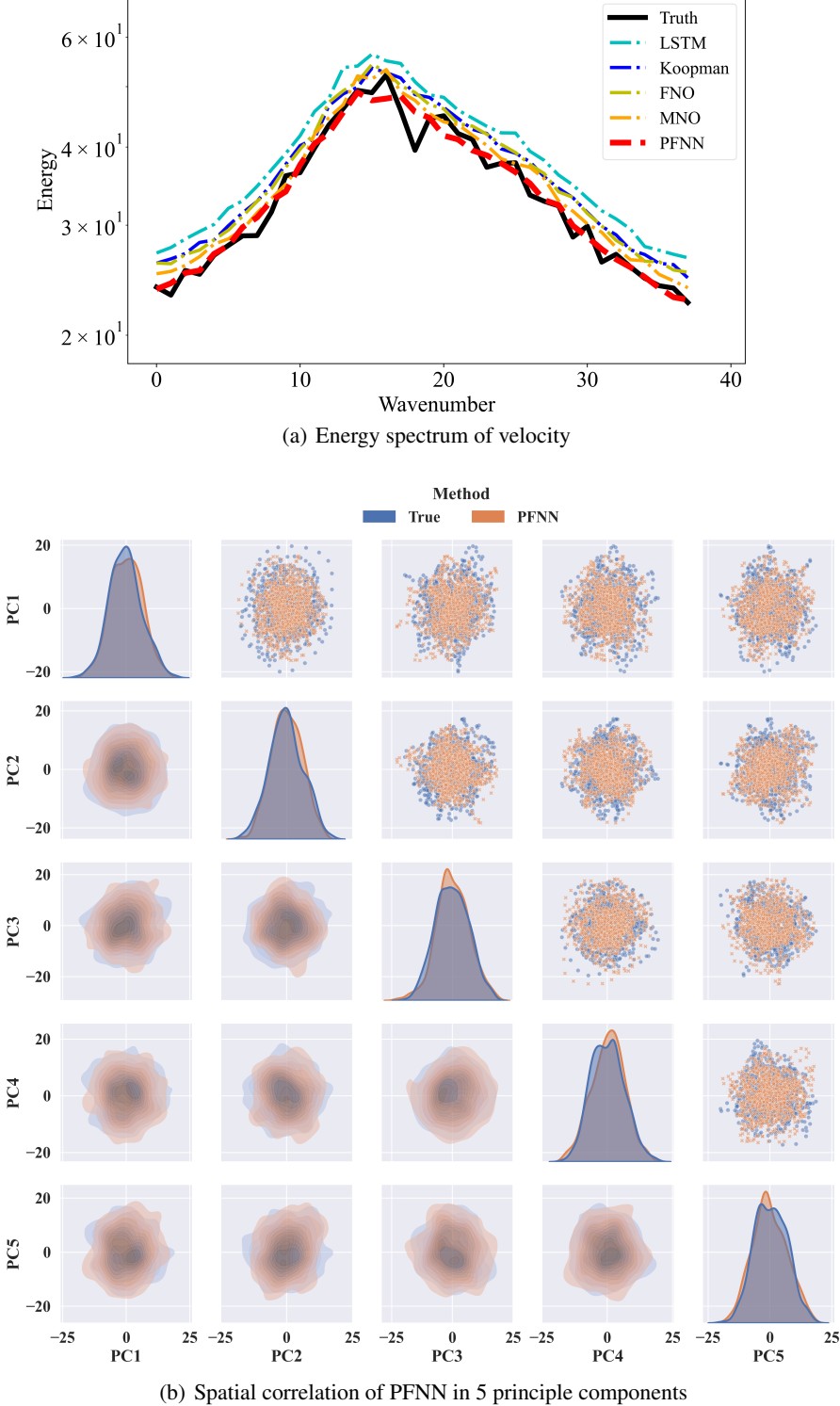

(a) Energy spectrum of velocity

(b) Spatial correlation of PFNN in 5 principle components

Figure 10: Visualization of long-term statistics of model predictions for Lorenz 96 system of 80 dimensions: we visualize density plots of each state dimension of the system velocity predicted by all six models; and then we visualize the spatial correlation among 5 principle components of the velocity, focusing on evaluating the learned spatial correlation from the PFNN model's long-term predictions compared with the ground truth.

F.3 KURAMOTO-SIVASHINSKY

The Kuramoto-Sivashinsky (KS) equation (Papageorgiou & Smyrlis, 1991), well-known for its chaotic behavior, is a nonlinear PDE applied to studying pattern formation and instability in fluid dynamics, combustion, and plasma physics. The dynamics in 1d spatial domain $u(x, t)$ is given by

$$\frac{\partial u}{\partial t} + u\frac{\partial u}{\partial x} + \frac{\partial^2 u}{\partial x^2} + \frac{\partial^4 u}{\partial x^4} = 0, \tag{37}$$

where $x \in [0, L]$ with a periodic boundary condition. The interaction of high-order terms produces complex spatial patterns and temporal chaos when the domain length $L$ is large enough (Cvitanović et al., 2010).

**Dataset processing for training and testing:** The dataset of Kuramoto-Sivashinsky simulations we utilized was obtained from the public source, consisting of 1200 simulated trajectories with 512 spatial dimensions, $u(x, t)$, on the periodic boundary. The simulations employed a pseudo-spectral method with exponential time differencing (ETD) to evolve the dynamics. Each trajectory was generated over 2000 timesteps with integration time $0.005s$ and sample rate of 10. We sampled 128-dimensional $u$ and used 1000 trajectories for training and 200 trajectories for testing, where each trajectory was truncated to 1990 timesteps to preserve the contraction phase before the system reached the ergodic state. For model training, the full trajectory length (1990 timesteps) was used to train baseline models, while the trajectory before the contraction step $k$ was used to train the PFNN contraction operator $\hat{\mathcal{G}}_c$, and the trajectory after $k$ was used to train the PFNN measure-invariant operators $\hat{\mathcal{G}}_c$ and $\hat{\mathcal{G}}_m^*$. The determination of the contraction step $k$ via the ablation study can be found in Table 2. For model evaluation, the full trajectory length in the test set was applied to all models.

**PFNN model architecture:** The PFNN model is designed to incorporate three pairs of encoder-decoder layers: one operator layer to represent the contraction dynamics, and two operator layers to represent unitary characteristics in the measure-invariant forward and backward dynamics. The model is implemented with PyTorch version 2.3.1, and the details of the layers are presented in Table 8.

**Benchmarks for Kuramoto-Sivashinsky:** We compare the PFNN model with classic recurrent neural networks: the Long Short-Term Memory network (LSTM), Koopman Operator network, Fourier Neural Operator (FNO), and Markov Neural Operator (MNO). We use the Adam optimizer to minimize the relative L2 loss with a learning rate of 1e-4, and a step learning rate scheduler that decays by half every 10 epochs, for a total of 100 epochs. Based on the provided code source in Appendix G, (1) for LSTM, we chose 2 layers and a latent feature dimension of $2 \times 128$; (2) for the Koopman Operator, we chose 1 layer and a latent feature dimension of $4 \times 128$; (3) for FNO and MNO, we set the width to 40, considering the down-sampled state dimension, and chose 30 Fourier modes to improve their ability to learn high-frequency dynamics; (4) for PFNN, the latent feature dimension is $8 \times 128$ for both the contraction model and the measure-invariant model.

Table 7: KL Divergence for long-term prediction distributions by models across principal components of Kuramoto-Sivashinsky

| Principle Components (PC) | KL Divergence | | | | |
|---|---|---|---|---|---|
| | FNO | LSTM | Koopman | MNO | PFNN |
| PC1 | 143.6508 | $\infty$ | 40.6045 | 23.8991 | **9.2303** |
| PC2 | 133.1811 | $\infty$ | 43.0569 | 26.6271 | **7.1571** |
| PC3 | 137.7399 | $\infty$ | 40.6242 | 25.9322 | **5.9686** |
| PC4 | 122.4125 | $\infty$ | 29.3022 | 19.4713 | **11.7539** |
| PC5 | 176.6664 | $\infty$ | 108.9509 | 26.7227 | **6.9519** |

**Accuracy results:** we provide more short-term and mid-term accuracy evaluations in 11. We noted Figure Notes: Visualization of model prediction performance from the NRMSE of short-term

prediction to relevant statistics of long-term prediction. We used 200 test samples and plotted the mean and standard deviation of all test results to obtain the NRMSE of the prediction.

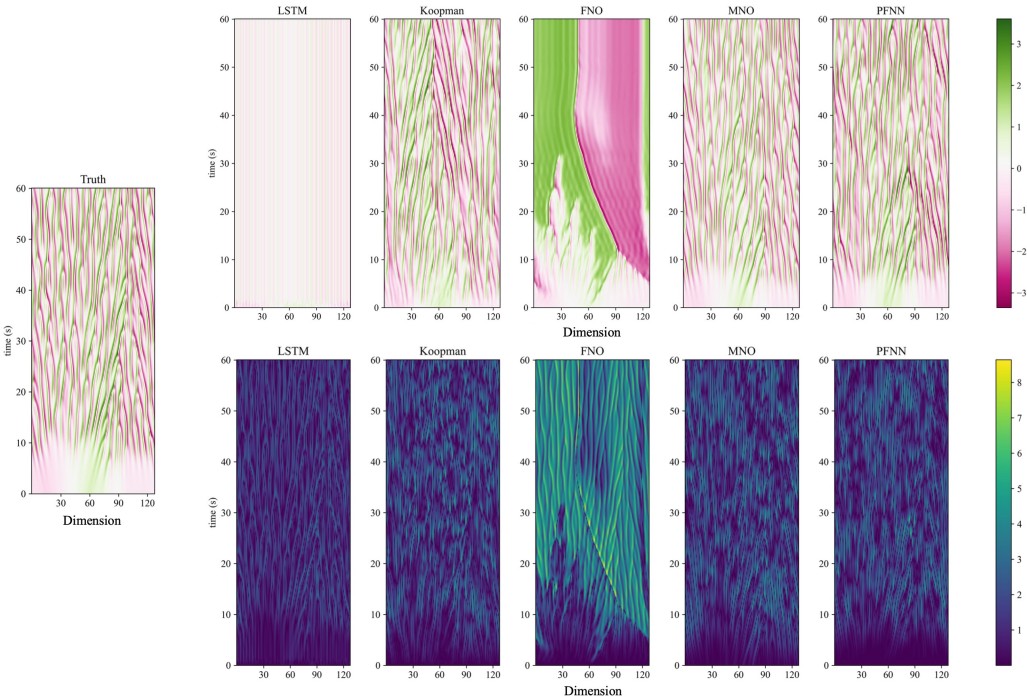

(a) Prediction visualization. The ground truth trajectory is visualized in the middle of the leftmost side. The predicted trajectories by baseline models and PFNN are shown in the first row. The corresponding absolute error trajectories of the predictions against the ground truth are shown in the second row.

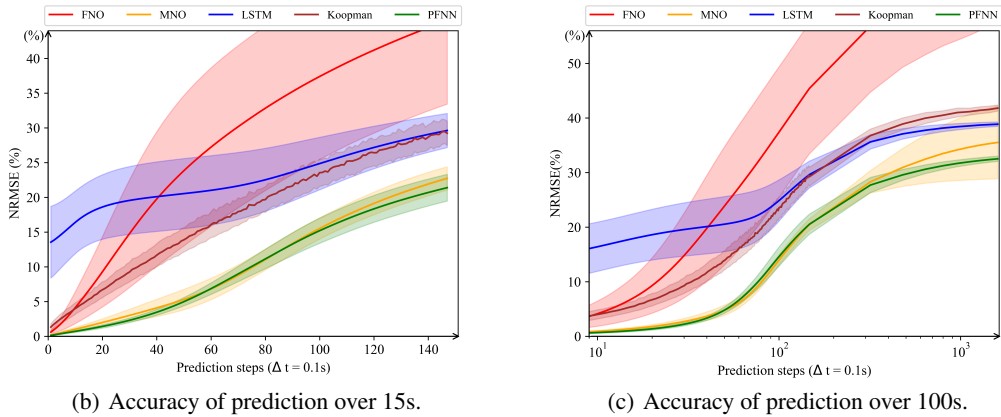

(b) Accuracy of prediction over 15s.      (c) Accuracy of prediction over 100s.

Figure 11: Visualization of prediction error in NRMSE: we visualize the comparison of model predictions of Kuramoto-Sivashinsky dynamics over 15 seconds (150 timesteps, short-term) and 100 seconds (1000 timesteps, mid-term).

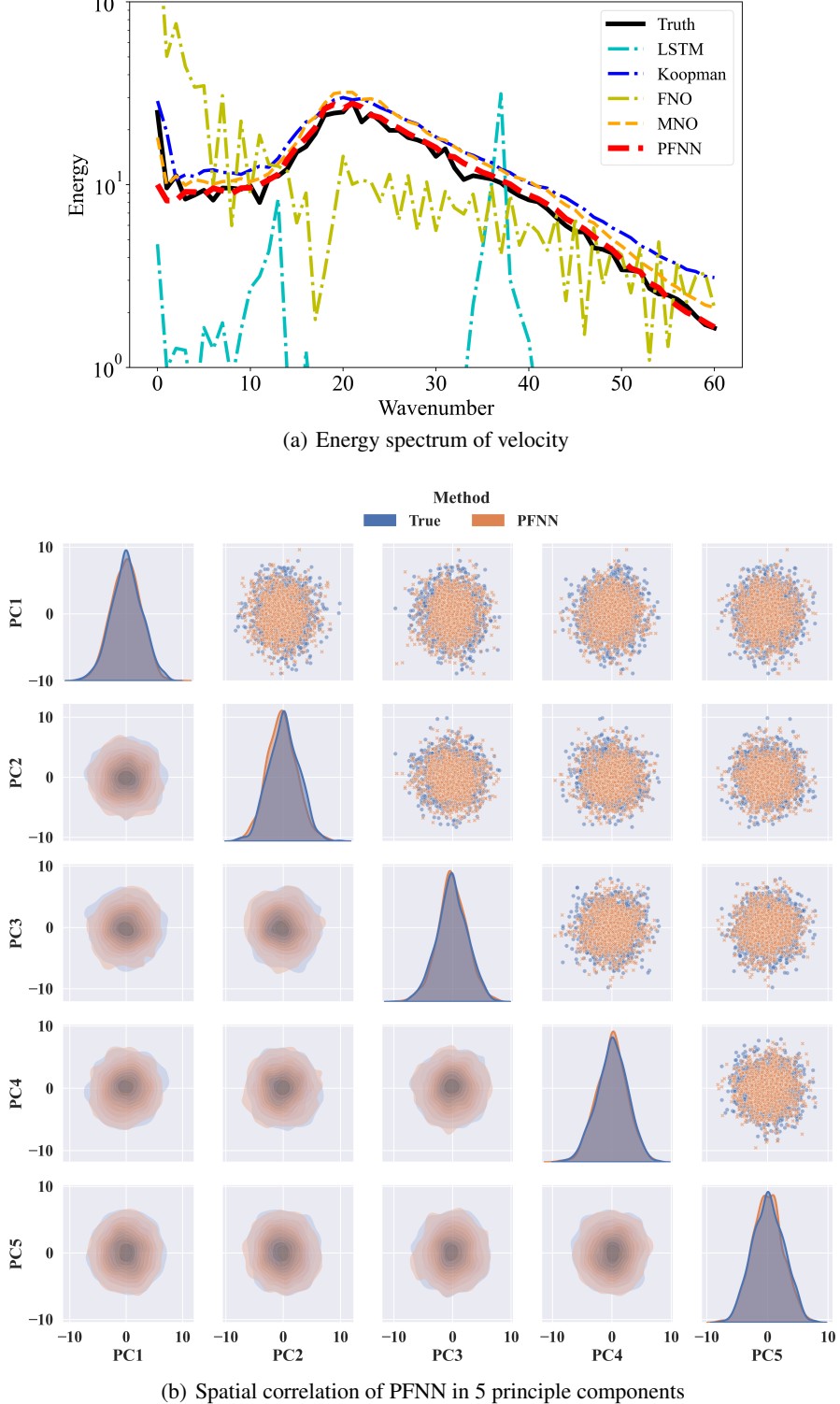

(a) Energy spectrum of velocity

(b) Spatial correlation of PFNN in 5 principle components

Figure 12: Visualization of long-term statistics of model predictions for Kuramoto-Sivashinsky: we visualize the energy spectrum of model predictions over 64 wavenumbers; and then we visualize the spatial correlation among 5 principle components of PFNN long-term predictions over all test states comparing with the spectrum of the ground truth.

### F.4 KOLMOGOROV FLOW

The Kolmogorov flow system, introduced by Arnold Kolmogorov, is a classic model for studying fluid instabilities and turbulence in two-dimensional incompressible flows Temam (2012). It is described by a nonlinear, incompressible Navier-Stokes equation driven by a sinusoidal forcing term.

$$\frac{\partial \mathbf{u}}{\partial t} + (\mathbf{u} \cdot \nabla)\mathbf{u} + \nabla p - \frac{1}{Re}\nabla^2 \mathbf{u} - \sin(kx)\hat{\mathbf{y}} = 0, \tag{38}$$

where $\mathbf{u}(x, y, t)$ is the velocity field, $p$ is the pressure, $\nu$ is the kinematic viscosity, and $\sin(kx)\hat{\mathbf{y}}$ represents the external forcing in the $y$-direction. The experiment setting follows the (Alieva et al., 2021).

**PFNN model architecture:** The PFNN model for the 2-D task begins with Convolution2D layers to do patch embedding of the input state, and employs the attention mechanism within Transformer Blocks Dosovitskiy et al. (2021) to perform feature encoding. Then, one operator layer is used to learn the contraction dynamics; and two operator layers are used to learn the unitary characteristics in the measure-invariant forward and backward dynamics. The model is implemented with PyTorch version 2.3.1, with detailed layer configurations provided in Table 9.

**Short-term prediction accuracy:** We compared the model performance in the short-term forecasting accuracy by evaluating the absolute error of model predictions with the true states at timesteps 2, 4, 8, 16 and 32 in measure-invariant phase in Figure 13. The result showed PFNN exhibits the smallest error across all steps, closely matching the ground truth and maintaining low errors even at Step 32.

**Estimating statistics in invariant-measure:** We further evaluated PFNN on the turbulent kinetic energy (TKE) metric, which is defined as:

$$TKE = \frac{1}{2}\left(\overline{u'^2} + \overline{v'^2}\right), \tag{39}$$

where $u', v'$ are the fluctuating components of velocity compared to velocity mean over time in the $x$ and $y$ directions respectively. The result in Figure 14 showed PFNN predicted trajectory preserved the internal physics statistics in (1) reproducing the TKE distribution with low absolute error (top row); and (2) capturing the time-average state of a system in invariant-measure (middle row); (3) comparing the lowest absolute error (AE) of the predicted mean state against the true mean state, PFNN outperforms the baselines as well (bottom row).

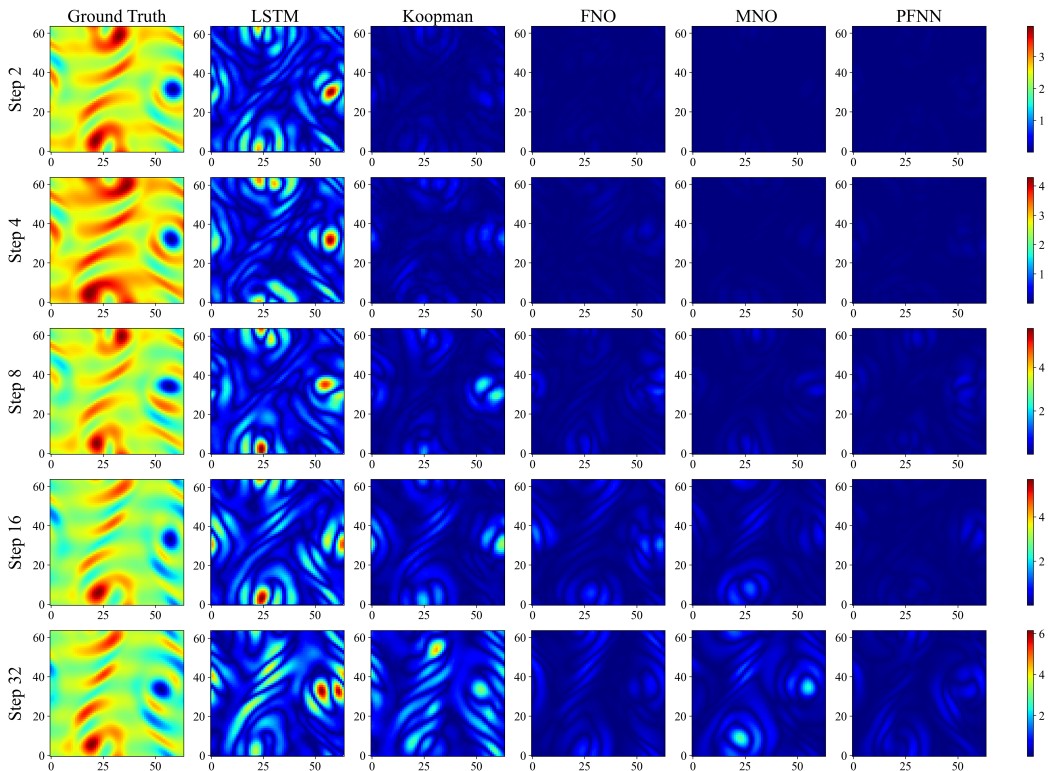

Figure 13: Visualization of the absolute difference of model predictions from ground truth in short-term steps $\{2, 4, 8, 16, 32\}$.

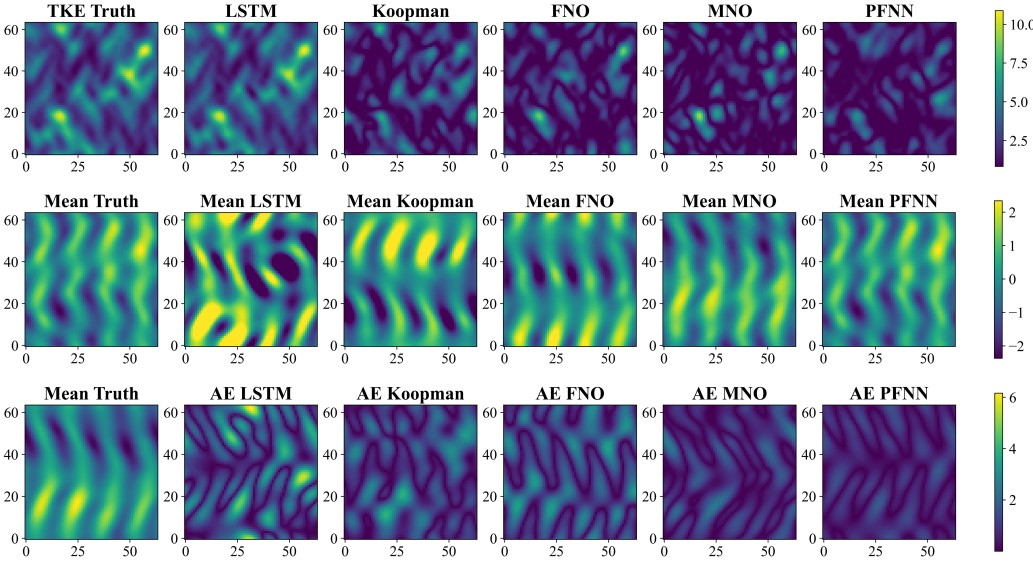

Figure 14: Visualization of the forecast trajectory sample of Kolmogorov Flow system in **top row:** the turbulent kinetic energy, which illustrates the accuracy of model predictions in reproducing the kinetic energy distribution; **middle row**: the mean state that demonstrated each model's ability in capturing the time-average velocity field of a system in equilibrium, where plot **bottom row:** the absolute error (AE) outstands the difference the predicted mean state against the true mean state.

# G    EXPERIMENT SETTINGS

## G.1    NEURAL NETWORK ARCHITECTURE

Table 8: Neural network architecture for the one dimensional chaotic systems. $d$ is the state dimension and $L$ is the latent feature dimension.

| Components | Layer | Weight size | Bias size | Activation |
|---|---|---|---|---|
| Encoder | Fully Connected | $m \times 10m$ | $10m$ | ReLU |
| Encoder | Fully Connected | $10m \times 10m$ | $10m$ | ReLU |
| Encoder | Fully Connected | $10m \times L$ | $L$ | |
| Forward operator $\hat{\mathcal{G}}_c$ | Fully Connected | $L \times L$ | $0$ | |
| Forward operator $\hat{\mathcal{G}}_m$ | Fully Connected | $L \times L$ | $0$ | |
| Backward operator $\hat{\mathcal{G}}_m^*$ | Fully Connected | $L \times L$ | $0$ | |
| Decoder | Fully Connected | $L \times 10m$ | $10m$ | ReLU |
| Decoder | Fully Connected | $10m \times 10m$ | $10m$ | ReLU |
| Decoder | Fully Connected | $10m \times m$ | $m$ | |

Table 9: Model architecture for Kolmogorov Flow

| Components | Layer type | Layer number | Channels, $(H, W)$ | Activation |
|---|---|---|---|---|
| Patch embedding | Convolution2d | 1 | $1 \rightarrow 32, (64, 64)$ | |
| Encoder | Transformer Block | 2 | $32 \rightarrow 64, (32, 32)$ | ReLU |
| Encoder | Transformer Block | 3 | $64 \rightarrow 128, (8, 8)$ | ReLU |
| Encoder | Transformer Block | 3 | $128 \rightarrow 128, (2, 2)$ | ReLU |
| Contraction operator $\hat{\mathcal{G}}_c$ | Fully Connected | 1 | $128 \times 2 \times 2, -$ | |
| Measure-invariant operator $\hat{\mathcal{G}}_m$ | Fully Connected | 1 | $128 \times 2 \times 2, -$ | |
| Backward operator $\hat{\mathcal{G}}_m^*$ | Fully Connected | 1 | $128 \times 2 \times 2, -$ | |
| Decoder | Transformer Block | 3 | $128 \rightarrow 128, (8, 8)$ | ReLU |
| Decoder | Transformer Block | 3 | $128 \rightarrow 64, (32, 32)$ | ReLU |
| Decoder | Transformer Block | 2 | $64 \rightarrow 32, (64, 64)$ | ReLU |
| Decoder | Convolution2d | 1 | $32 \rightarrow 1, (64, 64)$ | |

## G.2    DATASET.

(1) Lorenz 63: A 3-dimensional simplified model of atmospheric convection, known for its chaotic behavior and sensitivity to initial conditions. To learn our models, we generated a dataset consisting of 50 trajectories of 80,000 timesteps from random conditions. (2) Lorenz 96: A surrogate model for atmospheric circulation, characterized by a chain of coupled differential equations. We generated three datasets corresponding to 9, 40, and 80-dimensional states, respectively, each consisting of 2,000 trajectories with 1,500 timesteps. (3) Kuramoto-Sivashinsky equation: A fourth-order nonlinear partial differential equation that models diffusive instabilities and chaotic behavior in systems, such as fluid dynamics, and reaction-diffusion processes. We sampled a 128-dimensional dataset consisting of 1,000 trajectories with 500 timesteps from the dataset [11]. The description of the three dynamical systems is listed in Appendix F).

---

[11]Dataset for Kuramoto-Sivashinsky: https://zenodo.org/records/7495555

### G.3 TRAINING DETAILS AND BASELINES.

At a high level, PFNN and other baselines are implemented in **Pytorch** (Paszke et al., 2019). Both the training and evaluations are conducted on multiple A100s and Mac Studio with a 24-core Apple M2 Ultra CPU and 64-core Metal737 Performance Shaders (MPS) GPU. The evaluation is conducted on the CPU.

- LSTM: The implementation is based on the provided code of `https://github.com/pvlachas/RNN-RC-Chaos`.

- Koopman operator: The implementation is based on the provided code of `https://github.com/dynamicslab/pykoopman`.

- FNO: The implementation is based on the provided code of `https://pypi.org/project/fourier-neural-operator/`

- MNO: The implementation is based on the provided code of `https://github.com/neuraloperator/markov_neural_operator`.

