# OpenReview forum: "Learning Chaos In A Linear Way"
_ICLR.cc/2025/Conference — ICLR 2025 Poster_

### Official Review · Reviewer_GyXm · 2024-11-01

**Soundness:** 4
**Presentation:** 2
**Contribution:** 4
**Rating:** 8
**Confidence:** 4

**Summary:**

In this work, the authors address learning chaotic dynamical systems by introducing a model / learning paradigm that incorporates the physical properties of dissipative chaotic systems, namely that the volume of transient states contracts towards an attractive set on which measure is preserved by the dynamics. Across a variety of well-studied systems, the authors demonstrate the improvement from using their proposed PFNN method.

**Strengths:**

The problem and proposed methodology are well motivated. The exposition of why the various design decisions are made is rigorously stated and clear.

The experimental section is extensive and the improvements of PFNN over baselines is compelling. The authors provide a comprehensive appendix with all the relevant experimental details.

Figure 1 is very illustrative and a great way to convey the high level idea.

**Weaknesses:**

Overall, my main criticism is that while the methodology is well-motivated and rigorous, the actual implementation is somewhat difficult to follow. The phases and design decisions seem to be somewhat _ad hoc_, which is reflected in the number of hyperparameters (5 or 6 by my count), and I personally find the implementation difficult to follow. For example, I appreciate that the authors provided code, but looking through the code implementation, I struggle to see how the pseudocode in the algorithm maps to the implementation: in neither the Lorenz96 or NS script do I see the two-phase aspect of the training nor how equations 4, 6, or 8 get consistently mapped to code. I want to be clear, I am not taking the code implementation into account in my review/score, I only bring it up to illustrate that I think this method is less straightforward to implement.

Below I list more specific comments/concerns. The list roughly follows the order of appearance in the manuscript. In my opinion W1, W3, W5, W7/W9/W11, and W8  are the most important among these.

---

### W1: Is PFNN a parameterization or a training algorithm?
At a high level, I think the authors should clarify what PFNN refers to: i.e., is it the parameterization of the learned model (encoder + linear feature evolution) or does it refer to the 2 phase training paradigm with the various objectives proposed?

### W2: Lorenz63 is missing from the list of experimental setups in the abstract.
Not a big deal, but for completeness, I would recommend listing it.

### W3: Characterization of Schiff et al. (2024) is incorrect.
The authors place their work in contrast to Jiang et al. (2024) and Schiff et al. (2024), stating that unlike these works, the current proposal does not make any assumptions about the underlying PDE. While I agree that Jiang et al. (2024) requires knowledge of the system to build their set of invariant properties, this is not the case in Schiff et al. (2024). In fact, Schiff et al. (2024) clearly state that their proposed methodology does not make any assumptions about the underlying system, only that it supports an invariant measure, as in this work. In Schiff et al. (2024), the invariant measure is simply estimated as an empirical distribution from the trajectories in the training/test sets. Indeed, the same way the authors in this manuscript compute KL/MMD as evaluation metrics, Schiff et al. (2024) compute MMD as a regularizer.

I believe this characterization of previous work should be corrected. This does not detract from the fact that the current work, with its parameterization and two phase training, is novel.

**Additionally, it would be interesting to see how PFNN compares to baselines, say MNO (the strongest baseline), trained in conjunction with the regularized objective from Schiff et al. (2024). I highly recommend that this comparison be added to the experimental section.**

### W4: Introduction of the method is a bit confusing/misleading.
In the first paragraph the authors state that the algorithm / method has 3 steps. But if I understand the methodology correctly, then step 1 (i.e., learning the autoencode) is happening throughout both phase 1 (contractive) and phase 2 (measure preserving) of the training. This should be re-worded / clarified.

### W5: Consistency regularization seems to be dropped after Eq 4 is introduced.
The authors introduced the second term in Eq 4 as a way to enforce bijectivity. However there are a few things I find confusing:
1) After Eq 4, (e.g., in Eq 6 and Eq 8) this term is dropped. Why?
2) The hyperparameter $\gamma$ in Eq 4 is not discussed elsewhere in the manuscript and is not included in ablation studies.
3) The training algorithm in Algorithm 1 indicates that losses are computed by Eq 4 and Eq 6 (Line 1363) / Eq 4 and Eq 8 (Line 1366), but what does this mean in practice? Is the regularization term from Eq 4 included? The first term of Eq 4 is shared in both Eq 6 and Eq 8, is this term double counted? (presumably not)

### W6: Missing citation to “Evolve Smoothly, Fit Consistently: Learning Smooth Latent Dynamics For Advection-Dominated Systems” Wan et al. (2023).
In Wan et al. (2023), the learning dynamics are learned in a latent space and there is consistency regularization to ensure that the autoencoder is invertible. This shares several properties with the proposed methodology in the current manuscript. The authors should include Wan et al. (2023) in their related works and discuss how PFNN differs.

### W7: Isn’t the regularizer in Eq 6 quite expensive to compute?
To enforce the contractive property of $\hat{\mathcal{G}}$, the authors compute eigenvalues. Isn’t this quite expensive: $\mathcal{O}(L^3)$?

### W8: The choice of hyperparameter $k$ seems hard / requires knowledge of the system
The authors claim their method does not require any knowledge of underlying true dynamics, but the cutoff $k$ between contractive and measure preserving phases seems like it would highly depend on the system of interest.

Additionally, I would recommend adding more information (perhaps in the appendix) about how $c_{LSI}$ is derived/computed.

### W9: What is the complexity of computing objective in Eq 8?
$\mathcal{G}^*_m$ is described as the conjugate transpose, but in the provided code implementation it seems like it is computed as the Moore-Penrose pseudoinverse of the parameters of $\mathcal{G}_m$, unless I am misunderstanding something here. Is this not quite expensive to do and backprop through on each step/batch?

### W10: Algorithm should be provided in the main text
While I recognize that space is limited, I highly encourage the authors to find a way to include the training algorithm in the main body of the text. There are several components to keep track of in the methodology, and having them centralized in the main text in the form of the training Algorithm is really important and would aid clarity, as opposed to having to refer to the appendix.

### W11: Computation complexity of PFNN vs. baselines should be noted and commented on.
Related to my comment in W7/W9, it seems PFNN has several compute intensive overhead components. The complexity (both asymptotic and actual wall clock) of PFNN should be remarked on / included in the results sections.

### W12: Additionally, parameter counts should be included in Table 1
In addition to wall clock, the parameter counts of the various baselines and PFNN should be indicated to ensure methods are fairly compared.

### W13: Adding +/- for different random seeds on MMD/KLD would be very helpful
In my experience, ranking on metrics for the experimental setups explored here can vary significantly across different random seeds. If possible, adding this to Table 1, particularly for the MMD/KLD metrics, would make the results more compelling.

### W14: Why is MMD/KLD omitted for the NS experiment?
In particular, in the appendix, TKE is indicated to be a short-term accuracy metric, and so NS experiments seem to not have any reported “long-term” metrics.

### W15: In the ablation table, why is performance worse for $L = 512$?
I’d expect the trend (better results) to continue as $L$ grows.

### W16: Figure 5a (and other similar ones) is missing legends/labels
I am assuming the top row indicates trajectories and the bottom row shows error, but this needs to be indicated.

### W17: Minor typos / Formatting Issues & Suggestions:
- Defining $\nu$ as the data distribution should be more clear and included in the notation table in the appendix
- $m$ as the system dimension should be defined in paragraph 2 of Section 4. I would recommend also recalling what $m$ is in the caption of Table 1.
- Table 1 seems to have some bold formatting typos, e.g., the standard deviation is bolded for some rows (e.g. Line 401).
- Line 212: $\mathcal{G}_{ij}$ is missing the $L$ subscript.
- Line 221: In equation 4, the expectation should be over $\nu$ not $\mu$?
- Line 259/260: The footnote spacing is off. Should be attached to the previous word; looks like there is an added space.

---

**References:**

Ruoxi Jiang, Peter Y Lu, Elena Orlova, and Rebecca Willett. Training neural operators to preserve
invariant measures of chaotic attractors. Advances in Neural Information Processing Systems, 36,
2024.

Yair Schiff, Zhong Yi Wan, Jeffrey B Parker, Stephan Hoyer, Volodymyr Kuleshov, Fei Sha, and
Leonardo Zepeda-Nunez. Dyslim: Dynamics stable learning by invariant measure for chaotic systems. arXiv preprint arXiv:2402.04467, 2024.

Wan, Zhong Yi, Leonardo Zepeda-Nunez, Anudhyan Boral, and Fei Sha. "Evolve smoothly, fit consistently: Learning smooth latent dynamics for advection-dominated systems." arXiv preprint arXiv:2301.10391 (2023).

**Questions:**

### Q1: Is Equation 2 obvious?
As someone less familiar with operator theory, is the existence of $\mathcal{G}$ in Eq 2 obvious? The authors provide strong citations for the other theoretical aspects of their work, but I think citations are missing / would be helpful for the section where $\mathcal{G}$ is introduced.


Additionally, does the existence of $\mathcal{G}$ require any assumptions on the features $\phi$?

### Q2: In Lines 192-193, why do the authors state that the invariant measures “varies over time”?
Isn’t the invariant measure consistent over time?

### Q3: Why does $\gamma_2$ need to be between 0 and 1?
Is this a strict requirement or simply what the authors explored in their experiments?

### Q4: In the ablations, did the authors explore training only on the attractor?
In other words, what if we ignore $k$ in the training and assume all initial states are already on the attractor (e.g., by simply time evolving trajectories in the training/test sets and “throwing out the first $k$ steps). How does this affect performance of PFNN and baselines?

### In Figure 5b/c and other similar figures, where do the error bars / shaded regions come from?
How did the +/- get computed here?

---

> ### Author Response · Authors · 2024-11-22
> **Responses to Weaknesses**
>
> We, the authors, sincerely thank the reviewer for so many constructive suggestions and the in-depth understanding of theoretical and experimental parts, as well as its application to algorithm and loss function design. We sincerely appreciate the reviewer's patience in reading our work and offering thoughtful critiques, which we will acknowledge in the camera-ready version.
>
> General response to the weaknesses:
>
> The confusion of the code implementation in the general weakness statement is also linked to W10. We hope this work can meaningfully contribute to the open-source community of dynamics learning. During the anonymized period, we prefer to share the anonymized repository as a short version showcasing the essential implementation and key results. Upon conclusion of this period, we plan to release the full PFNN code to help reproduce and extend. We appreciate the reviewer's understanding, and the ongoing attention to this project is very welcome.

---

> ### Author Response · Authors · 2024-11-22
> **Responses to the prioritised weaknesses**
>
> **Answer to W1**:
>
> PFNN is a training algorithm for generic operator-based models that carefully incorporates the physical principle of chaotic systems as a constraint during training. We hope this answer addresses the reviewer's confusion and helps better position our work.
>
>
> **Answer to W3**:
>
> We appreciate the reviewer's elaboration on the difference between [1] and [2]. We have updated the characterization of [1] in the revised manuscript.
> The work [1] does excellent work in predictive chaos dynamics. We greatly value the reviewer's suggestion to include a comparison with the regularized objective from [1]. However, we do have difficulty in reporducing the results of [1] as the demo in the mentioned repository is empty. We appreciate the reviewer's understanding. (https://github.com/google-research/swirl-dynamics/blob/main/swirl_dynamics/projects/ergodic/colabs/demo.ipynb). (It seems the repo is under new rounds of development these months...).
>
> [1] Yair Schiff, Zhong Yi Wan, Jeffrey B Parker, Stephan Hoyer, Volodymyr Kuleshov, Fei Sha, and Leonardo Zepeda-Nunez. Dyslim: Dynamics stable learning by invariant measure for chaotic systems. arXiv preprint arXiv:2402.04467, 2024.
>
> [2] Ruoxi Jiang, Peter Y Lu, Elena Orlova, and Rebecca Willett. Training neural operators to preserve invariant measures of chaotic attractors. Advances in Neural Information Processing Systems, 36, 2024.
>
>
> **Answer to W5**:
>
> Thanks for your question. We do not drop the reconstruction loss term during the two-phase training. To clarify, we have added the reconstruction term to the loss functions in Equations 6 and 8 in the latest manuscript.
>
> **Answer to W7**:
>
> Thank you for your question. We agree that the computational complexity of the regularizer in Eq. 6 is $O(L^3)$. However, based on our numerical experiments, we found that using $L = 512$ is sufficient to handle complex 2D turbulence problems, achieving satisfactory performance while remaining computationally feasible with modern matrix computation. Additionally, we observed that the learned contraction operators tend to have many values close to zero, indicating the potential for learning operators with a sparse structure. This sparsity could further mitigate concerns about computational complexity in future extensions.
>
>
> **Answer to W8**:
>
> Our primary claim is that knowledge of the invariant distribution is neither required nor necessary to estimate it. In experiments, we actually did not estimate or check the exact invariant distributions before and during training.
>
> The calculation of the log-sobolev constant $c_{LSI}$ involves extensive mathematical details that span several pages. We value this question and try to interpret it concisely and easy-to-understand. Please refer to the **General Response For the Calculation Of Relaxation Time** at the top of the rebuttal page. We hope this interpretation helps understanding.
>
> **Answer to W9**:
>
> In our implementation, instead of directly computing the pseudo-inverse of $\mathcal{G}$, we explicitly parameterize $\mathcal{G}_m^*$ as a trainable matrix. During training, we enforce the desired relations between $\mathcal{G}_m^*$ and $\mathcal{G}_m$ using the loss function defined in Equation 8. Specifically, the consistency loss ensures they are conjugated and transposed to each other is maintained throughout the training process.
> We thank the reviewer for raising this question, as it helps clarify our algorithm.
>
> **Answer to W11**:
>
> Thanks for the question. We update (1) a table of computational complexity of PFNN and baselines; and (2) a table of wall clock time and visualization for running 100-step forecasting on a medium dimensional task. Please refer to the link (https://anonymous.4open.science/r/PFNN-F461/README.md)

---

> ### Author Response · Authors · 2024-11-22
> **Responses to other weakness**
>
> **Answers to W2, W4, W16, W17**:
>
> Thanks for spotting these points! We have addressed the questions in the updated paper.
>
> **Answer to W6**:
>
> Thanks for your advice. We cite this paper in the second paragraph of our introduction in the latest version. This paper is fundamentally different from our work:
>
>   1. Scope: the cited paper focuses on developing a data-driven, space-time continuous framework for learning surrogate models of advection-dominated systems described by PDEs. In contrast, our paper aims to provide a new perspective on understanding dissipative chaotic systems and enhancing long-term prediction outcomes.
>
>   2. Method: the cited work employs a hypernetwork-based approach to create efficient latent dynamical models that yield smooth, low-dimensional representations for more accurate and computationally efficient multi-step rollout predictions. Our approach, on the other hand, projects the state space into an infinite-dimensional feature space and embeds physical knowledge to enforce the properties of an infinite-dimensional operator. This embedding ultimately improves the statistical accuracy of long-term predictions of chaotic systems.
>
> **Answer to W10**:
>
> Yes, we are trying to spare some space from the main body and we plan to finalise the improved content arrangement in the camera-ready version.
>
> **Answer to W12**:
>
> We'd like to kindly **query** the reviewer whether the presentation of wall clock time and trainable parameters now shown in the link (https://anonymous.4open.science/r/PFNN-F461/README.md) is fine now? If so, we would add them into the camera-ready version.
>
> **Answer to W14**:
>
> Thanks for the question. Actually the TKE is a long-term statistics of turbulent flow [1]. We believe there might be some misunderstaning, i.e., short term and long term figures are adjacent in the same page that leads some rough galance on the caption.
>
> Compared with KLD and MMD, Turbulent Kinetic Energy (TKE) is a more applicable and meaningful metric in the NS experiment [2,3], particularly because Kolmogorov flow generates a statistically stationary turbulent flow.
>
> [1] Frisch U. Turbulence: the legacy of AN Kolmogorov[M]. Cambridge university press, 1995.
>
> [2] Wang, Rui, et al. "Towards physics-informed deep learning for turbulent flow prediction." Proceedings of the 26th ACM SIGKDD international conference on knowledge discovery & data mining. 2020.
>
> [3] Kohl, Georg, Liwei Chen, and Nils Thuerey. "Benchmarking autoregressive conditional diffusion models for turbulent flow simulation." ICML 2024 AI for Science Workshop. 2024.
>
>
> **Answer to W15**:
>
> We appreciate the reviewer's concern regarding the sensitivity of the latent dimension $L$. Given the fixed dataset, constantly increasing $L$ might lead to worse results because of the risk of aggressive overparameterization, which makes the model overfit or hard to optimize/compute. The result of the ablation study of $L$ also echoes the empirical analysis in [1] and [2], which discuss the benefits and challenges of overparameterization in related models.
>
> [1] Karniadakis, G. E. (2023). On the influence of over-parameterization in manifold based surrogates and deep neural operators. *Journal of Computational Physics* , 479 , 112008
>
> [2] Buhai, R. D., Halpern, Y., Kim, Y., Risteski, A., & Sontag, D. (2020, November). Empirical study of the benefits of overparameterization in learning latent variable models. In *International Conference on Machine Learning* (pp. 1211-1219). PMLR.

---

> ### Author Response · Authors · 2024-11-22
> **Responses to Questions**
>
> **Answer to Question 1**:
>
> Thanks for your advice. We have added the citation in Equation 2 to the latest version.
>
> Generally, there are some assumptions on the choice of features we'd like to address:
>   * Boundedness: features should be bounded to ensure that the operator can be well-defined on $L^2$ space.
>   * Linear Embedding: dynamics can be represented by a linear operator on the feature functions see Equation 2.
>   * Richness: features must be rich enough to capture the essential dynamics of the original nonlinear system with sufficiently small errors.
>
> We hope this helps the reviewer to understand it.
>
>
> **Answer to Question 2**:
>
> Thanks for your question. It is a typo, we have corrected it in the updated manuscript.
>
> **Answer to Question 3**:
>
> Thanks for your question. It is a standard setting in deep learning.
>
> **Answer to Question 4**:
>
> Thank you for your questions. We did not specifically train only on the attractor because our PFNN is designed to target general dissipative chaotic systems. We assume no prior knowledge about the locations of the system's attractors, making our approach more broadly applicable to a wide range of scenarios.
>
> **Answer to Question 5**:
>
> The error bars or shaded areas in all the figures are generated by repeatedly evaluating our PFNN and other baselines using randomly sampled initial states from the test set, which covers the entire state space, rather than limiting evaluations to points on the attractors.
>
> ---
> Thank you for your thoughtful review and valuable questions. We sincerely appreciate the time and effort you have taken to provide feedback to enhance our work. We hope our responses effectively address your questions and concerns, and we look forward to any further comments or insights you may have.

---

> ### Author Response · Authors · 2024-11-24
>
> May I ask the reviewer whether our response/improvement made has addressed your concerns? There is not much time left for discussion. We would be pleased to answer your further questions. Many thanks.

---

> ### Author Response · Authors · 2024-11-25
>
> This is a gentle reminder to inquire for reviewer GyXm if our responses and additional experiment results have addressed your concerns. Please note that the deadline is approaching, and we would greatly appreciate it if you could share any further feedback or increase score accordingly.
>
> Thank you for your time and consideration.

---

> > ### Comment · Reviewer_GyXm · 2024-11-25
> > **Response to rebuttal**
> >
> > Thank you very much for the detailed, thorough, and thoughtful responses to my concerns and questions. Overall, I think this is a good paper and the empirical improvements over baselines is compelling. I think my main hesitation is that I was not sure how the complexity of the method would reflect in param count and wall clock time. That concern has been addressed by the tables in the anonymized README. I highly recommend that the authors include these / similar tables in their camera ready version. I have adjusted my score to 8 to reflect the fact that I think this should definitely be published and shared with the community.
> >
> > One minor point, for W14: I don't  understand the author's response about this being a layout confusion. Specifically, looking at lines 2123 - 2133, the TKE is defined under a paragraph header **"Short-term prediction accuracy"**. Unless I am misunderstanding something this is confusing, because the authors are titling this a short-term prediction metric but in the main paper it's treated as a "long-term" metric. Let me know if I am missing something here. If not, this should be fixed for the camera ready version.

---

> ### Author Response · Authors · 2024-11-25
> **Thanks reviewer GyXm for your efforts**
>
> We greatly appreciate your efforts in refining our paper and suggesting potential directions for further enhancement. We have enjoyed our discussions with Reviewer GyXm.
>
> >I highly recommend that the authors include these / similar tables in their camera ready version
>
> We will include the above mentioned tables in our camera-ready version.
>
> > One minor point, for W14: I don't understand the author's response about this being a layout confusion. Specifically, looking at lines 2123 - 2133, the TKE is defined under a paragraph header "Short-term prediction accuracy". Unless I am misunderstanding something this is confusing, because the authors are titling this a short-term prediction metric but in the main paper it's treated as a "long-term" metric. Let me know if I am missing something here. If not, this should be fixed for the camera ready version.
>
> Thank you very much for pointing this out. We will separate the turbulence kinetic energy from the "short-term prediction accuracy" session and ensure this is updated.
>
> Best regards

---

### Official Review · Reviewer_pS8g · 2024-11-03

**Soundness:** 2
**Presentation:** 3
**Contribution:** 3
**Rating:** 6
**Confidence:** 3

**Summary:**

This paper proposes the Poincare Flow Neural Network (PFNN), an architecture for learning in dissipative chaotic systems. PFNN maps the input to a latent space and proposes to learn linear evolution operators in this latent space. PFNN is applied on several systems, including Lorenz 96, Kuramoto Sivashinsky, and Navier-Stokes, and it demonstrates improvements over prior baselines.

**Strengths:**

The paper is well-written and includes a good exposition of key concepts (e.g., ergodicity, global attractor, etc.) for audiences that may not be as familiar with the mathematical background. The experiments are in-depth, particularly in the appendix. The proposed method also outperforms prior works across a variety of settings and metrics. The detailed proofs in the appendix are also helpful.

**Weaknesses:**

The authors should explain in more detail the distinction between their framework and prior works. In particular, some details on the distinction between neural operators and the Koopman operator approach are not clear. To my understanding, the proposed approach is based on Koopman operator theory and is learning, in part, both the discretized operator matrix $\mathcal{G}_{ij}$ and the encoder and decoder feature functions. However, the chaotic systems in question are assumed to be defined over a subset of $\mathbb{R}^d$. In contrast, the neural operator (e.g., FNO, DeepONet) setting is one of mapping between function spaces. As such, methods like MNO are most interested in learning the solution operator to a time-evolving function (PDE). To my understanding, the proposed approach cannot do this directly in function space as designed, as it is fixed to a specific data resolution. Is this accurate? To me Koopman and neural operators seem to be different paradigms, but in my opinion the authors do not make sufficient distinction between them. However, my interpretation may be mistaken. Please let me know if I am missing a connection in this case.

I also have some questions regarding the experiments and baselines. These can be found in the “Questions” section below.

**Questions:**

1. At inference-time for a system, how do you choose $k$? How robust is the proposed model to variations in $k$?
2. Does this $k$ depend on the input trajectory? One could imagine a trajectory initialized very close or very far from the attractor. How is $k$ determined in that case?
3. How are the operator learning architectures (FNO and MNO) parameterized for the finite-dimensional problems (L63, L96)? Is this different from how they are parameterized for KS and NS?
4. Does constraining the spectrum of the operator $\mathcal{G}$ give contraction in the physical space as well? I suggest adding more details about this in the paper to make it clearer.
5. Is there potential for this method to be applied on function spaces, e.g., to forecast a Navier-Stokes fluid flow in a discretization-agnostic manner?
6. Since the model is dependent on the input discretization, how does the proposed architecture scale with input resolution? For instance, how does convergence and performance compare on Navier-Stokes at 256x256 resolution vs. on 64x64 resolution?
7. Does $\hat G$ in equation 4 correspond to the approximation finite-rank operator?

---

> ### Author Response · Authors · 2024-11-22
> **Responses to Weaknesses**
>
> **Answer to Weaknesses**:
>
> Thank you for your detailed feedback. We have revised the Introduction to better situate our work within the Koopman learning framework. We appreciate the reviewer's attention to these updates and would value a second reading to evaluate the changes made. Below, we address the key concerns:
>
> **1. Scope of PFNN**:
> Our primary focus is on modeling chaotic systems using physical insights, not necessarily designing a grid-agnostic framework. PFNN aims to design linear operators specifically for capturing dissipative chaos with embedded physical knowledge. This approach involves the PFNN model within discretized spatial domains, ensuring a fair comparison with other baselines. We appreciate the reviewer's understanding.
>
> **2. Applicable Beyond $\mathbb{R}^d$**:
> The PFNN is also applied in 2D tasks using a $\mathbb{R}^d \times \mathbb{R}^d$ structure, demonstrating its potential for spatial complexity. Meanwhile, PFNN has been successfully tested on both ODEs (e.g., Lorenz 63 and Lorenz 96) and PDEs (e.g., Kuramoto-Sivashinsky and Navier–Stokes equations), capturing dynamics across a variety of time-evolving systems.
>
> **3. Relationship between Koopman operators and neural operators**:
> Firstly, the Neural operator and the Koopman operator have natural connections with each other. The Koopman operator was originally conceived as an infinite-dimensional operator mapping between function spaces $L^2$, not merely a discretized matrix (see [1] and Section 3.1 in our paper). This operator theory fundamentally connects to a mapping between function spaces, similar to neural operators such as FNO or DeepONet.
> Building on this foundation, the Koopman operator can be endowed with grid-independent properties through a well-designed auto-encoder structure and thus satisfy all the neural operator properties, as demonstrated in recent work [2].
> Thus, the connection between Koopman operators and neural operators is natural. We hope this explanation clarifies the connection and encourages a deeper exploration if the reviewer is interested.
>
> In short, we understand the reviewer's concern regarding grid independence. At present, our method operates with fixed data resolution, but the theoretical properties of the Koopman operator [1, 2, 3] offer the potential for future extensions to grid-independent frameworks for learning chaotic systems. We view this as a meaningful avenue for further research.
>
> We thank the reviewer for raising these insightful points, which will help guide the future evolution of our work.
>
> [1] Koopman, Bernard O. "Hamiltonian systems and transformation in Hilbert space." Proceedings of the National Academy of Sciences 17.5 (1931): 315-318.
>
> [2] Xiong, Wei, et al. "Koopman neural operator as a mesh-free solver of non-linear partial differential equations." Journal of Computational Physics (2024): 113194.
>
> [3] Brunton, Steven L., et al. "Modern Koopman theory for dynamical systems." arXiv preprint arXiv:2102.12086 (2021).

---

> ### Author Response · Authors · 2024-11-22
> **Answer to Questions 1 and 2:**
>
> We thank the reviewer for their curiosity in choosing/determining the data split time and the robustness.
>
> The practical calculation of relaxation time has been provided in the General Response on the Calculation of Relaxation Time.
>
> **Robustness.** Our results show that the algorithm becomes quite stable once the hyperparameter, $\frac{1}{ c_{LSI} }$, passes a certain value. Specifically, as seen in our ablation study, when $\frac{1}{c_{LSI}}$ is greater than 5, the algorithm consistently performs well. However, $\frac{1}{c_{LSI}}$ should not be set too high, as this would affect the balance between the data used in the two training phases. It’s crucial to maintain this balance to ensure optimal performance.

---

> ### Author Response · Authors · 2024-11-22
> **Answer to Question 3**
>
> **Answer to Question 3**:
> Thanks for your question. We would like to clarify, for L63, we adhered to the default settings outlined in the baseline paper [1] for FNO and MNO. Specifically for the L63 task in [1], DenseNet (feedforward network) was employed as the operator, with dissipative post-processing applied to DenseNet to implement MNO. For the 1D high-dimensional tasks (L96 and KS), we used 1D FNO blocks for both FNO and MNO models respectively. For the 2D NS tasks, we used 2D FNO blocks for FNO and MNO models.
>
> Hope the answer addresses your concerns.
>
> [1] Li Z, Liu-Schiaffini M, Kovachki N, et al. Learning chaotic dynamics in dissipative systems[J]. Advances in Neural Information Processing Systems, 2022, 35: 16768-16781.

---

> ### Author Response · Authors · 2024-11-22
> **Answer to Question 4, 5, 6, 7**
>
> **Answer to Question 4**:
> Thank you for your question. To provide a comprehensive understanding, we have visualized the spectrum of both the contraction operator and the unitary operator. We also add a physical vector field of PFNN's predictions on Lorenz 63, which may help a physical understanding of contraction. These details have been added to the updated link for better clarity: https://anonymous.4open.science/r/PFNN-F461/README.md.
>
>
>
> **Answer to Question 5**: Thank you for your question. The answer is YES, the method has great potential to do discretization-agnostic applications. The grid invariance property relies on the structure of the encoder (see reference [1]). While our framework is proposed using a generic autoencoder, it can be extended to incorporate the grid invariance property. We include it as future research direction.
>
> [1] Xiong, Wei, et al. "Koopman neural operator as a mesh-free solver of non-linear partial differential equations." Journal of Computational Physics (2024): 113194.
>
> **Answer to Question 6 (New results added)**: Thanks for your question. PFNN definitely can be extended to a larger scale. We are excited to share new experiment results of stable forecasting the $256 \times 256$ Navier-Stokes equations to demonstrate the scalability of this approach, see: https://anonymous.4open.science/r/PFNN-F461/README.md.
>
> **Answer to Question 7**: Yes, thanks for the question. We have updated one sentence to explain it in the latest manuscript.
>
> ---
> Thank you for your thoughtful review and valuable questions. We appreciate the time and effort you invested in providing feedback to improve our work. We hope our responses adequately address your concerns and look forward to any further insights you may have.

---

> ### Author Response · Authors · 2024-11-24
>
> May I ask the reviewer whether our response/improvement made has addressed your concerns? There is not much time left for discussion. We would be pleased to answer your further questions. Many thanks.

---

> > ### Comment · Reviewer_pS8g · 2024-11-24
> >
> > I thank the authors for their improvements and for addressing my concerns. I have raised my score accordingly.

---

> ### Author Response · Authors · 2024-11-25
> **Official Comment by Authors**
>
> We would like to thank Reviewer pS8g for carefully reviewing our response and additional experiments. Your feedback helped us greatly improve the paper, and we greatly appreciate your re-evaluation and updated score.

---

### Official Review · Reviewer_Fbbj · 2024-11-04

**Soundness:** 3
**Presentation:** 2
**Contribution:** 3
**Rating:** 6
**Confidence:** 4

**Summary:**

In this paper, the authors proposed a Poincare Flow Neural Network (PFNN) to model dissipative chaotic dynamical systems that improves matching of invariant statistics in forecasting over long horizons. The proposed method uses an autoencoder to reformulate prediction on the state space into prediction on a finite-dimensional learned feature space, on which a linear operator might be able to represent the forward evolution of the system in the feature space theoretically. By estimating the transient time the system needs to converge to the attractor with "relaxation time", the trajectory data sets are separated into a contraction phase and a measure-invariant phase. Using soft constraints formulated as regularization loss of the learned linear operator, the physical properties of the system, namely (1) dissipativity, in other words, volume contraction during the contraction phase (2) measure invariance due to a unitary operator construction, are incorporated into the training loss. Overall, the method leverages a Koopman-like structure to encourage the model to match intrinsic physical behaviors of a dissipative chaotic system by adding soft constraints on the linear operator. The improvement in preserving invariant statistics in the true system of the proposed method is validated in a few numerical experiments compared to recent benchmarks in the literature.

**Strengths:**

Originality: The proposed method provides a new perspective on incorporating dissipative chaos physical behaviors into a neural network model through simple to compute yet powerful linear operator properties.
Quality: The proposed method has a strong mathematical foundation, which is explained well in the paper. Additionally, the reviewer finds the numerical experiments to be extensive and the ablation study helpful.
Clarity: Overall the paper is written clearly, with minor editorial comments on presentation listed in other sections of the review.
Significance: The paper addresses an important problem in learning chaos, which has extensive practical applications including weather forecast, turbulent flow modeling etc.

**Weaknesses:**

Overall, the reviewer finds the method innovative and well articulated. However, there are a few technical details and writing concerns that need to be addressed to make sure the paper can be accepted in a form that clarifies its own contributions.

Technical issue:
1. It is very crucial in this approach to obtain a good separation of trajectory data into the contraction and measure-invariance phase, since these two phases use different operators. However, it seems like all the approximation techniques introduced in Section 3.2 can only reduce the estimation to the same order of magnitude, which might be problematic if the transient period is relatively short.
More importantly, the steps for separation chosen in the paper $\frac{1}{c_{LSI} log(||\phi||_2/\epsilon)$ depend explicitly on $\phi$, which is the learned feature function. It is confusing how the steps can be chosen using this formula before the network is trained, however, this is a necessary step to preprocess the trajectory dataset.

2. The auto-encoder does not seem to have a structure to help learn the feature functions that are orthonormal or help construct a good finite-rank operator approximation. The reviewer would appreciate clarification of the link between Theorem 3.1, or more generally the design of autoencoder structure, and the Koopman theory.

Writing issues:
1. To the reviewer's best knowledge, the proposed structure is very similar to the auto-encoder structure inspired by Koopman operator theory proposed by Steven Brunton (which includes a series of papers). Although the authors have cited one of these works, the reviewer suggests that the author to clarify that the auto-encoder structure and Koopman-inspired network design is perhaps not a novel design.
2. It seems like quite a few theorems in the paper e.g., Proposition 3.2 and Theorem 3.3 are standard results either in functional analysis or Koopman literature. The reviewer fails to see the points of rederiving such results, instead, one might be able to cite these well-known results and directly use them.
3. It seems like the theory in Theorem 3.1 and functional analysis in relation to constructing the finite-rank operator which approximates the designed linear operator suggests the feature functions need to be orthonomal or at least somewhat independent/orthogonal. Since the feature functions are learned using auto-encoders, it would be very helpful to provide some insights on what features are learned. It's a bit concerning to the reviewer that the authors make the point that FNO/MNO relies on periodic Fourier modes contradictory to the chaos nature without the presence of their own learned feature map results.
4. This paper clearly has its own contributions in leveraging linear operator theory and incorporating certain properties. In order to keep the literature review fair, the reviewer suggests the authors to consider revising introduction especially comparison to neural operators. To make matters clear, the reviewer is not at any point a co-author of any neural operator work but is familiar with this line of work. It seems like more general forms of neural opeartors (1) do not necessarily use Fourier modes (2) might use nonlinear evolution such as kernel structure. It might not be fair to group this entire line of work into the linear evolution box and claims strict improvement over FNO/MNO (Neural operators with Fourier modes), which is a fraction of them.
5. It seems like the framework assumes a finite-dimensional ODE structure of the dynamical system. Given a discretization grid, this might be true for a broad range of systems, the reviewer suggests the authors to make this limitation clear and restricts the scope to given spatial discretization. There are other papers in the domain that can deal with more general discretization setup such as DeepOnet or Neural operators. Clearly, this paper would not be able to address discretization issues like the other papers.

**Questions:**

1. When the finite-dimensional features spaces are constructed (L different learned features), wouldn't the "operators" G_c, G_m simply collapse to matrices? If so, please make it clear as matrix transpose notion has already been used in the paper.
2. What is the main difference between the proposed architecture from the Koopman learning literature? e.g. Lusch, Bethany, J. Nathan Kutz, and Steven L. Brunton. "Deep learning for universal linear embeddings of nonlinear dynamics." Nature communications 9.1 (2018): 4950.
3. In Theorem 3.1, it seems that the result does not depend on the choice of the autoencoder. Is this true for every set of feature functions learned by the encoder?
4. Just out of curiosity, does such a linear operator always exist? If so, is there any intrinsic difficulty in approximating such infinite-dimensional operator using finite dimensional feature space for certain systems?

---

> ### Author Response · Authors · 2024-11-21
> **Responses to Technical Issues**
>
> **Answer to technical issue 1**:
>
> Thank you for raising this important point. The parameter $ k $ is an intrinsic property of chaotic dynamics, and it varies with the initial state $ z $. We have revised the manuscript to reduce some confusion. In the latest manuscript, the length of the contraction phase is defined as
> $
> \frac{1}{c_{LSI}} \log \left(\frac{\lVert z \lVert_2}{\epsilon}\right)
> $ based on the initial state $z$ and is used to estimate the relaxation time. It does not involve the learned feature function $\phi$.
>
> For dissipative chaotic systems,
> $
> \log \left(\frac{\lVert z \lVert_2}{\epsilon}\right)
> $
> can be interpreted as an **energy-like quantity** (see Lines 286–294 of the manuscript). If the initial state $ z $ is already within the attractor, the norm $\lVert z \lVert_2$ will be small, leading to a very short relaxation time. In such cases, the system quickly transits from the contraction phase to the measure-invariance phase. For detailed description of relaxation time referred to general response of Calculation Of Relaxation Time,
>
> We hope this explanation clarifies our approach and can resolve the confusion regarding the preprocessing of the trajectory dataset.
>
> **Answer to technical issue 2**:
>
> The primary purpose of Theorem 3.1 is NOT to guide the design of the autoencoder structure. Instead, it serves to prove that a finite-rank operator can achieve a sufficiently small error when the dimension is sufficiently large (see Lines 247–254). Actually, using other basis such as polynomials can also prove the finite-rank approximation [1, 2]. Utilizing an orthonormal basis is a common strategy in functional analysis for demonstrating the existence of finite-rank approximations [2]. As our paper clearly states:
>
> "The theorem reflects that the finite-rank operator, derived from the infinite-dimensional operator, operates in a function space spanned by learned latent feature functions. By selecting a sufficiently large $L$ as the dimension of the latent feature space, stepwise prediction can be achieved."
>
> For practical implementation in deep learning, we adopt the generic autoencoder structure (see Table 8 in Appendix G). Here, we learn the features by jointly training the autoencoder and enforcing the operator properties outlined in Equations 6 and 8 to satisfy the requirements.
>
> [1] Brunton, Steven L., et al. "Modern Koopman theory for dynamical systems." arXiv preprint arXiv:2102.12086 (2021).
>
> [2] Schmüdgen, Konrad. Unbounded self-adjoint operators on Hilbert space. Vol. 265. Springer Science & Business Media, 2012.

---

> ### Author Response · Authors · 2024-11-21
> **Responses to Writing Issues 1-3**
>
> **Answer to writing issue 1**:
>
> Thanks for your question. We review the previous work related to the Koopman learning framework. Hence, the design of the linear structure is NOT our aimed contribution. Our work focuses on how to embed the physical knowledge in the linear structure to learn dissipative chaos (see the revised version in Lines 97-107 and 121-125).
>
> **Answer to writing issue 2**:
>
> Proposition 3.2 and Theorem 3.3 are pivotal in linking spectral theory with ergodic theory. We have designed our paper to be self-contained, ensuring that a general audience can follow our main ideas. Additionally, we provide a comprehensive analysis in the Appendix that elaborates on the theoretical inspirations and explains how these theorems from functional analysis are interconnected with ergodic theory.
>
> **Answer to writing issue 3**:
>
> This answer is constructed in 3 parts.
>
> Part-1: The primary purpose of Theorem 3.1 is NOT to guide the design of a different autoencoder network structure. Instead, it serves to prove that a finite-rank operator can achieve a sufficiently small error when the dimension is sufficiently large (see Lines 247–254). Actually, using another basis, such as polynomials, can also prove the finite-rank approximation [1, 2]. Utilizing an orthonormal basis is a common strategy in functional analysis for demonstrating the existence of finite-rank approximations.
>
> Part-2: The feature is actually learned by constraining the spectral properties of operators. We jointly train the auto-encoder and operators following Equations 6 and 8. The visualization of the feature function is equivalent to visualizing the spectrum of learned operators. Figure 2 illustrates the factorized eigenvalues of the operator $(\mathcal{G} - \lambda_i I )\phi_{i} = 0$,  where $\lambda_i$ is the $i$-th eigenvalue of feature $\phi_{i}$. The factorized eigenvalues of the unitary operator demonstrate how the learned feature functions evolve linearly under this operator (see reference [1]). To comprehensively visualize the features of both the contraction operator and the unitary operator, we have also included them in the updated repository: https://anonymous.4open.science/r/PFNN-F461/README.md. We hope it helps understanding.
>
> Part-3 **Integer Fourier**: Based on the ergodic theory, operators of chaotic systems with ergodic behaviors have mixed spectrum (discrete and continuous spectrum) on the unit complex circle [2, 3]. When using the operator $\mathcal{G}$ to describe the evolution of dynamics, each feature updates as $\phi_i (z_{k+1}) = \lambda_i \phi_i(z_k)$. Thus, the spectrum of the operator $\mathcal{G}$, i.e. eigenvalues, reflected the learned features. The learned spectrum of PFNN is distributed densely on the unit complex circle (see Figure 2), aligning with the Koopman-von Neumann (KvN) ergodic theorem.
> In contrast, FNO and MNO [4,5] rely on Fast Fourier Transforms (FFT), which extract low-mode and integer Fourier components. Mathematically, the integer Fourier spectrum corresponds to periodic behavior, which contradicts the properties of ergodicity (see Theorem 1.1 in [6], Page 3). As a result, this approach is insufficient to capture the continuous spectrum.
>
> [1] Stone, Marshall H. "On one-parameter unitary groups in Hilbert space." Annals of Mathematics 33.3 (1932): 643-648.
>
> [2] Koopman, Bernard O., and J. V. Neumann. "Dynamical systems of continuous spectra." Proceedings of the National Academy of Sciences 18.3 (1932): 255-263.
>
> [3] Neumann, J. V. "Proof of the quasi-ergodic hypothesis." Proceedings of the National Academy of Sciences 18.1 (1932): 70-82.
>
> [4] Li, Zongyi, et al. "Learning chaotic dynamics in dissipative systems." Advances in Neural Information Processing Systems 35 (2022): 16768-16781.
>
> [5] Li, Zongyi, et al. "Fourier neural operator for parametric partial differential equations." arXiv preprint arXiv:2010.08895 (2020).
>
> [6] C. McMullen. "Ergodic theory, geometry and dynamics." https://people.math.harvard.edu/~ctm/papers/home/text/class/notes/ergodic/course.pdf

---

> ### Author Response · Authors · 2024-11-21
> **Responses to Writing Issues 4 and 5**
>
> **Answer to Writing issue 4**:
>
> In the updated manuscript, we refined the literature review related to MNO and FNO. (See lines 111-113)
>
> **Answer to Writing issue 5**:
>
> Thanks for the question. We don't assume an ODE structure in our paper. Our operator $\mathcal{G}: L^2 \to L^2$ is a map between function spaces (see the paper Section 3.1). We conduct experiments on two Partial Differential Equations (PDEs), e.g., the Kuramoto-Sivashinsky (KS) equation and the Navier-Stokes (NS) equations.
>
> Since we focus primarily on designing algorithms to model chaotic systems, adopting a general grid-agnostic structure is NOT our primary target. For the grid invariance property, it is more related to designing the new autoencoder structure, as referred to in [1].  Exploring grid-agnostic frameworks remains an **interesting** direction for future research, and we have included it in the limitation and future research.
>
> [1] Xiong, Wei, et al. "Koopman neural operator as a mesh-free solver of non-linear partial differential equations." Journal of Computational Physics (2024): 113194.

---

> ### Author Response · Authors · 2024-11-21
> **Responses to Questions**
>
> **Answer to Question 1**:
>
> Thanks for your question. Even in the finite-dimensional feature space, the "operators" $G_c, G_m: L^2 \to L^2$ are not simple matrices since, theoretically, they are mapping between function spaces $L^2$ as we described in Section 3.1. In operator theory, the transpose symbol can also be used for both matrices and finite-dimensional real operators.
>
> **Answer to Question 2**:
>
> Thanks for your question. The referenced paper [1] proposes an architecture for learning ODEs and PDEs, but its target and methodology are fundamentally different from ours, as outlined below:
> 	1.	We introduce a novel perspective based on dynamical systems theory to learn dissipative chaotic systems. Specifically, we focus on two distinct phases: the contraction phase and the measure-invariant phase. Using this framework, we employ relaxation time as a key metric to separate these two phases;
> 	2.	Within each phase, we propose new loss functions designed to capture the intrinsic behaviors of dynamical systems effectively, which are clearly different from [1]. We appreciate the reviewer's suggestion and have cited this paper in the updated manuscript.
>
> [1] Lusch, Bethany, J. Nathan Kutz, and Steven L. Brunton. "Deep learning for universal linear embeddings of nonlinear dynamics." Nature Communications 9.1 (2018): 4950
>
> **Answer to Question 3**:
>
> Thanks for your meaningful question.
> * As we mentioned earlier, the primary purpose of Theorem 3.1 is to prove that a finite-rank operator can achieve a sufficiently small error when the dimension is sufficiently large (see Lines 247–254). The applicable sets of feature functions should meet the following conditions:
>   * **Boundedness**: Features should be bounded to ensure the operator is well-defined on the $ L^2 $ space.
>   * **Linear Embedding**: The dynamics must be representable by a linear operator acting on the feature functions (see Equation 2).
>   * **Richness**: The features must be rich enough to capture the essential dynamics of the original nonlinear system with sufficiently small errors.
>
> The detailed proof in Theorem 3.1 reflects the properties. We hope this answer helps your understanding.
>
>
> **Answer to Question 4**:
>
> We appreciate the reviewer's curiosity in the in-depth understanding of PFNN.
>
> Such linear operators do not always exist. For an operator to exist in this context, it needs to be a bounded operator that is well-defined on a complete metric space, such as a Hilbert or Banach space, equipped with a strong topology (the conditions are clearly stated in Lines 219-221, 1113-1133).
>
> Approximating the linear operator for chaotic systems should capture the continuous spectrum. It is still an open problem in this community. We appreciate the reviewer's attention in this community.
>
> ---
> Thank you for your thoughtful review and valuable questions. We sincerely appreciate the time and effort you have taken to provide feedback to enhance our work. We hope our responses effectively address your questions and concerns, and we look forward to any further comments or insights you may have.

---

> ### Author Response · Authors · 2024-11-24
>
> May I ask the reviewer whether our response/improvement made has addressed your concerns? There is not much time left for discussion. We would be pleased to answer your further questions. Many thanks.

---

> ### Author Response · Authors · 2024-11-25
> **Official Comment by Authors**
>
> This is a gentle reminder to inquire for reviewer Fbbj if our responses and additional experiment results have addressed your concerns. Please note that the deadline is approaching, and we would greatly appreciate it if you could share any further feedback or increase score accordingly.
>
> Thank you for your time and consideration.

---

> ### Author Response · Authors · 2024-11-27
>
> We kindly inquire whether our responses and the improvements we have made have adequately addressed your concerns. With only 12 hours remaining to update the manuscript, we would be pleased to answer any further questions you may have. This matter is of utmost importance to us, and we are diligently working to polish the paper to the highest standard. Thank you very much for your time and consideration.

---

> ### Author Response · Authors · 2024-11-28
>
> Dear Reviewer Fbbj,
>
> I am writing to kindly inquire whether our responses and the additional experimental results we provided have adequately addressed your concerns regarding our manuscript. With the discussion deadline approaching, we would greatly appreciate any further feedback you might have. If you are satisfied with the revisions, we would also be grateful if you could consider updating your review score accordingly.
>
> Thank you very much for your time and consideration.

---

> > ### Comment · Reviewer_Fbbj · 2024-11-29
> > **Thanks for your detailed responses and comments; One more clarifying question**
> >
> > I appreciate the author's detailed responses to every point in my review comments and the clarification their responses have provided. I also acknowledge the author's effort in revising the paper with re-evaluation and updating numerical results in such a short time frame. Most of my own comments have been sufficiently addressed, except I still have some concerns about clarity on the scope of this paper.
> >
> > I agree that a grid-agnostic architecture is an orthogonal research direction. Under the fixed grid setting, the paper still addresses an interesting problem and makes solid contribution.
> >
> > However, it seems to me that the framework is still effectively learning an ODE model rather than modeling PDE in the function space. As clearly stated in problem setup and Eq (1), $z_{k+1} = T(z_k), z \in \mathbb{R}^m$, which means that the systems under consideration are strictly ordinary differential equations with finite dimensional states.
> >
> > Although I agree that numerically using 128 spatial sample points for KS and 64x64 resolution grid for 2D Kolmogorov flow are common ways to study PDEs, I think modeling a system that evolves from one state $z_k \in \mathbb{R}^m$ to the next state $z_{k+1} \in \mathbb{R}^m$ is still an ODE modeling problem, again as the states are finite-dimensional. More importantly, since the proposed framework maps between finite dimensional vectors, the dissipative properties are only enforced over such a vector to vector mapping but not enforced on any point in between the sample point grids. If my understanding is correct, then the learned network serves as a function that maps from $\mathbb{R}^m$ to $\mathbb{R}^m$, instead of $L^2$ to $L^2$, simply because we are at the end of the day dealing with sampled finite states rather than a continuous PDE solution function.
> >
> > I would much appreciate the author's clarification on this question and modify the paper (in the camera-ready version) if they agree with my assessment. Again, I would like to clarify that I understand and acknowledge of the important contribution even in an ODE case, however, I would be more comfortable to read a paper that carefully states its scope. If the goal is to learn $\mathbb{R}^m$ to $\mathbb{R}^m$ map, that is completely respectful, and I don't think there's any additional value in stating it as operator learning.
> >
> > If my assessment is incorrect, I welcome the author's feedback and clarification.

---

> ### Author Response · Authors · 2024-11-29
>
> Thank you for your feedback and for recognizing our contributions.
>
> We fully agree with your remarks regarding the scope of our research. Our work focuses on discretized PDEs modeled as high-dimensional ODE systems, and we will make a clear distinction between our approach and operator learning in the revised manuscript (camera-ready). Specifically, we will elaborate on how our framework handles the finite-dimensional state mappings and the implications for modeling dissipative chaotic properties. We have identified grid-agnostic properties as a limitation and propose addressing them in future research.
>
> In the camera-ready version, we will:
> - Clearly define the scope of our research, emphasizing modeling on discretized PDEs.
> - Differentiate our approach from neural operators (like FNO, DeepOnet) by highlighting the discrete state mappings.
>
> We are committed to thoroughly refining our paper to enhance its clarity and rigor. Your constructive feedback has been invaluable, and we have diligently incorporated your suggestions to ensure the highest quality of our work. We hope it can properly address your remaining concerns.
>
> Thank you once again for your valuable feedback.

---

> > ### Comment · Reviewer_Fbbj · 2024-11-29
> > **Thank you, I have adjusted my score**
> >
> > Thank you for your prompt response! I have adjusted soundness score from 2 to 3 and overall score from 5 to 6.
> >
> > Please make the adjustments listed in the 2 bullet points in the camera-ready version. In my opinion, a paper with clearly stated scope and contribution on a meaningful (if not in the most general form) problem provides better readability.

---

> ### Author Response · Authors · 2024-11-29
>
> We would like to thank Reviewer Fbbj for carefully reviewing our response. Your feedback helped us greatly improve the paper, and we greatly appreciate your re-evaluation and updated score :).

---

### Official Review · Reviewer_CMhA · 2024-11-04

**Soundness:** 2
**Presentation:** 2
**Contribution:** 2
**Rating:** 5
**Confidence:** 4

**Summary:**

This paper proposes Poincaré Flow Neural Network (PFNN), which maps chaotic dynamical systems to a latent space where their evolution is approximated by one of two linear operators: a dissipative operator for the initial contraction phase and a unitary operator for the measure-invariant phase on the attractor. PFNN enforces these properties of the learned operators using carefully designed loss functions. The switching time between the two phases / operators is a hyperparameter of the method that must be tuned. The proposed method is evaluated on several chaotic systems including Lorenz 63/96, Kuramoto-Sivashinsky, and Navier-Stokes equations.

**Strengths:**

1. The proposed method is rigorously grounded in dynamical systems theory, and the design of the loss function provides a nice example of how a theoretical or mathematical understanding of a problem can be incorporated into the learning process.
2. The paper is generally clear and very well presented, although the overall score here is let down by how the proposed method is contextualized with respect to the literature, as discussed below.

**Weaknesses:**

### Major
1. A major issue with the presentation of the paper is how it deals with Koopman operator learning. In particular, the core of the proposed method consists of a learned embedding followed by learned linear evolution in the latent space. This is Koopman operator learning [1,2,3,4], which is a vast and active area of research. Despite forming the foundation of the present work, Koopman learning only receives a few passing mentions in the text.

   For example, with regard to the proposed learning setup, the authors state "This concept is fundamentally aligned with neural operators, such as the Koopman operator (Bevanda et al., 2021)..." However, rather than simply being "aligned" with Koopman operator learning, the method proposed here _is_ Koopman operator learning. For someone not already familiar with Koopman learning, it would be easy to read this paper and think that the core learning strategy -- linear evolution in a learned latent space -- is a novel contribution.

   The actual contribution of this paper is in fact to split the time domain into separate contractive and measure invariant phases, with a carefully designed loss function for each. In other words, what we are doing here is a kind of "physics-informed Koopman learning" for dissipative chaotic systems. This is fine, albeit not particularly groundbreaking. Either way, it should be made very clear how the proposed method relates to prior work on Koopman learning.

    Taking all of this into account, I think it is essential that the proposed method is properly and rigorously placed within the context of existing literature on Koopman learning, both theoretically and experimentally. This requires a detailed discussion of key papers on the foundations as well as the state-of-the-art of Koopman learning. It is also necessary to compare empirically to the state-of-the-art in Koopman learning (the authors do include a Koopman operator baseline, but they do not include any details of the exact setup.) Finally, please also include a paragraph clearly stating the marginal contribution of this paper.

    Ultimately, it is for this reason that I have given this paper a 1 for presentation, even though it is generally quite well written.

2. The results presented in Table 1 are inconclusive and could be presented better. Firstly, only the NRMSE is provided with a measure of uncertainty (although I think the standard error would be more useful than standard deviation for judging the significance of the results). For almost every system, there is no significant difference between FNO, MNO, and PFNN for short-term predictions, as measured by NRMSE. For longer-term predictions, as measured by KLD/MMD/TKE, PFNN performs comparably with MNO for every system except KS, where it does appear better, but the lack of uncertainty estimates makes it hard to draw conclusions. Ultimately, the results as they are presented suggest that PFNN offers relatively little advantage over MNO.

3. The authors do not compare to echo state networks, which are extremely effective networks for learning chaotic dynamical systems. In my opinion, they are an essential comparison in a paper like this. See, for example, [5,6], as well as [7] and citations therein.

### Minor
1. The authors are a bit loose with the term “neural operator”, which I understand to mean a family of models for learning on function spaces. These models possess specific properties, such as grid invariance, that are not shared by the proposed method. See, e.g., [8,9].

2. As a shortcoming of MNO, the authors note that “choosing the right size for this absorbing ball is tricky”. However, the proposed method requires choosing the relaxation time $k$ as a hyperparameter, essentially replacing one hard-to-estimate hyperparameter with another. More generally, I have concerns about the dependence of the proposed method on the choice of $k$, since it requires that the dataset at hand can be neatly split in this manner (what if certain initial conditions are already on the attractor?). In addition, I find the ablation study in Table 2 unconvincing, since there is little meaningful difference between $k = 7$ and $k = 9$; I think the range of $k$ needs to be extended to larger values.

### References
[1] Lusch, B., Kutz, J.N. & Brunton, S.L. Deep learning for universal linear embeddings of nonlinear dynamics. Nat Commun 9, 4950 (2018). https://doi.org/10.1038/s41467-018-07210-0

[2] Qianxiao Li, Felix Dietrich, Erik M. Bollt, Ioannis G. Kevrekidis; Extended dynamic mode decomposition with dictionary learning: A data-driven adaptive spectral decomposition of the Koopman operator. Chaos 1 October 2017; 27 (10): 103111. https://doi.org/10.1063/1.4993854

[3] Otto, S. E., & Rowley, C. W. (2019). Linearly Recurrent Autoencoder Networks for Learning Dynamics. SIAM Journal on Applied Dynamical Systems, 18(1), 558-593. https://doi.org/10.1137/18M1177846

[4] E. Yeung, S. Kundu and N. Hodas, "Learning Deep Neural Network Representations for Koopman Operators of Nonlinear Dynamical Systems," 2019 American Control Conference (ACC), Philadelphia, PA, USA, 2019, pp. 4832-4839, doi: 10.23919/ACC.2019.8815339.

[5] Jaideep Pathak, Zhixin Lu, Brian R. Hunt, Michelle Girvan, Edward Ott; Using machine learning to replicate chaotic attractors and calculate Lyapunov exponents from data. Chaos 1 December 2017; 27 (12): 121102. https://doi.org/10.1063/1.5010300

[6] Pathak, J., Hunt, B., Girvan, M., Lu, Z., & Ott, E. (2018). Model-Free Prediction of Large Spatiotemporally Chaotic Systems from Data: A Reservoir Computing Approach. Physical Review Letters, 120(2), 024102.

[7] Yan, M., Huang, C., Bienstman, P. et al. Emerging opportunities and challenges for the future of reservoir computing. Nat Commun 15, 2056 (2024). https://doi.org/10.1038/s41467-024-45187-1

[8] Kovachki, N., Li, Z., Liu, B., Azizzadenesheli, K., Bhattacharya, K., Stuart, A., & Anandkumar, A. (2023). Neural Operator: Learning Maps Between Function Spaces With Applications to PDEs. Journal of Machine Learning Research, 24(89), 1-97.

[9] Azizzadenesheli, K., Kovachki, N., Li, Z. et al. Neural operators for accelerating scientific simulations and design. Nat Rev Phys 6, 320–328 (2024). https://doi.org/10.1038/s42254-024-00712-5

**Questions:**

1. How do you understand the relation of the proposed method to Koopman learning?

1. For the Koopman operator baseline, what is the exact network setup? Are there any other Koopman learning architectures you think are relevant to the current problem?

2. Why is the model called Poincare Flow Neural Network? The name seems to be an homage to Poincare rather than a description of what the network actually does.

3. In Table 1, it’s not clear what is meant by TKE. Is this an error in TKE with respect to ground truth?

4. Could you elaborate on this statement? "Notably, while FNO uses FFT to extract features and provides relatively accurate short-term predictions, its reliance on integer Fourier modes tends to produce periodic behavior."

5. Could you elaborate on this statement? "Models like LSTM and Koopman, which do not account for transient behaviors for states lying outside attractors..."

6. Is there a reason the longer-term experimental results are not reported with uncertainty estimates?

7. Is there a reason you haven't compared with echo state networks?

---

> ### Author Response · Authors · 2024-11-21
> **Responses to Major Weakness 1 and Question 2**
>
> **Answer to Major Weakness 1 and Question 2**:
>
> Thank you for recognizing our work. Our method is fundamentally a type of Koopman operator learning specifically tailored for dissipative chaotic systems. To clarify, we have revised the introduction (see Lines 98-106) better to position our approach within the Koopman learning literature.
>
> **Relationship to Koopman operator learning**:
> Our method fundamentally applies Koopman operator learning with an emphasis on dissipative chaotic systems, and this is not addressed by prior Koopman frameworks. Unlike the generic approaches that apply a finite-rank operator for ODE/PDE modeling, our method incorporates two distinct phases of contraction and measure-invariant with specific data processing method and training constraints into the deep autoencoder based koopman operator learning. By incorporating the physical characteristics of dissipative chaos system, PFNN achieved more accurate and robust performance than the baselines.
>
> **The novel contribution of PFNN**:
> Our core contribution lies in the novel separation of the time domain into contractive and measure-invariant phases, allowing the Koopman operator to evolve in alignment with the underlying system's physical properties. This enables us to handle dissipative chaotic systems by employing a dissipative operator during the contraction phase and a unitary operator during the ergodic phase, guided by the Lumer-Phillips and Von Neumann theorems. This dual-phase approach is what distinguishes our work as 'physics-informed Koopman learning' for chaotic systems.
>
> **Positioning PFNN in Koopman learning literature**
> The reviewed literature establishes two primary directions of data-driven koopman operators: (1) traditional methods, including kernel approaches and ynamic mode decomposition (DMD)[1,2]; and (2) deep Koopman learning techniques using autoencoder networks [3,4, 5, 6]. However, none of these approaches specifically target dissipative chaos. By combining operator theory with physical dynamics, our method fills this gap, providing an effective solution for chaotic systems that have both dissipative and measure-invariant phases.
>
> **Empirical comparison with modern Koopman methods (Answer relates to Question 2 as well)**
> We evaluated the Koopman baseline using the default configuration of the pyKoopman toolbox (https://pykoopman.readthedocs.io/en/master/), which provides a strong baseline for comparison. This baseline is also deployed with deep autoencoder based Koopman operator architecture and uses the same scale of parameters to ensure fair comparison with PFNN. We provide (1) a table of trainable parameters used by each model for low-dimensional and high-dimensional chaotic dynamics; (2) a picture of a Koopman baseline model architecture implemented with the pyKoopman toolbox on the page (https://anonymous.4open.science/r/PFNN-F461/README.md). We hope this illustration helps the reviewer to understand PFNN better.
>
> [1] Ikeda, Masahiro, Isao Ishikawa, and Corbinian Schlosser. "Koopman and Perron–Frobenius operators on reproducing kernel Banach spaces." Chaos: An Interdisciplinary Journal of Nonlinear Science 32.12 (2022).
>
> [2] Kostic, Vladimir, et al. "Learning dynamical systems via Koopman operator regression in reproducing kernel Hilbert spaces." Advances in Neural Information Processing Systems 35 (2022): 4017-4031.
>
> [3] Williams, Matthew O., Ioannis G. Kevrekidis, and Clarence W. Rowley. "A data–driven approximation of the koopman operator: Extending dynamic mode decomposition." Journal of Nonlinear Science 25 (2015): 1307-1346.
>
> [4] Takeishi, Naoya, Yoshinobu Kawahara, and Takehisa Yairi. "Learning Koopman invariant subspaces for dynamic mode decomposition." Advances in neural information processing systems 30 (2017).
>
> [5] Lusch, Bethany, J. Nathan Kutz, and Steven L. Brunton. "Deep learning for universal linear embeddings of nonlinear dynamics." Nature communications 9.1 (2018): 4950.
>
> [6] Nathan Kutz, J., Joshua L. Proctor, and Steven L. Brunton. "Applied Koopman Theory for Partial Differential Equations and Data‐Driven Modeling of Spatio‐Temporal Systems." Complexity 2018.1 (2018): 6010634.

---

> ### Author Response · Authors · 2024-11-21
> **Responses to Major Weakness 2 and Question 7**
>
> **Answer to Major Weakness 2 and Question 7**:
> We thank the reviewer for their question. Our method shows a clear improvement over MNO in terms of long-term statistics (Table 1) and the energy spectrum (Figure 2). Indeed, as noted, uncertainty quantification has often been overlooked in similar experiments in the literature and baseline papers, e.g. [1] and [2] which we agree is a significant limitation. Aligning with the reviewer’s perspective, we made considerable efforts to address this issue in our work. Please refer to the updated Table 1 in the revised manuscript for details.
>
> The overall evaluation data in Table 1 remain stable due to our robust methodology. Specifically, we repeatedly generated long-term roll-outs in an auto-regressive manner for the entire test set across 50 different random seeds. Notably, the uncertainty of the Maximum Mean Discrepancy (MMD) metric is smaller than that of the Kullback-Leibler Divergence (KLD). This is expected because MMD, which evaluates the mean discrepancy between distributions, consistently exhibited small standard deviations across all methods. In contrast, KLD, which measures differences between probability distributions, showed relatively larger standard deviations across all baselines.
>
> These observations are reasonable and align with our expectations. Importantly, our proposed method (PFNN) demonstrated consistent performance under both metrics, underscoring its robustness and reinforcing the claims made in our manuscript. We hope this analysis provides clarity and adds value to the reviewer’s feedback.
>
> [1] Li Z, Liu-Schiaffini M, Kovachki N, et al. Learning chaotic dynamics in dissipative systems[J]. Advances in Neural Information Processing Systems, 2022, 35: 16768-16781.
>
> [2] Schiff, Yair, et al. "DySLIM: Dynamics Stable Learning by Invariant Measure for Chaotic Systems." arXiv preprint arXiv:2402.04467 (2024).

---

> ### Author Response · Authors · 2024-11-21
> **Responses to Major Weakness 3 and Question 8**
>
> **Answer to Major Weakness 3 and Question 8**:
> This is not a very standard baseline in learning dissipative chaotic systems, as indicated by the previous work [1, 2]. To provide additional context, we conducted experiments using an echo state network (ESN) [3, 4] on the Lorenz 63 system and the Kuramoto–Sivashinsky equation. The new results can be found at (https://anonymous.4open.science/r/PFNN-F461/README.md) linked to a Jupyter Notebook of detailed implementation. We hope it adds value for the reviewer.  The performance of ESN was significantly lower compared to some other baselines and PFNN. As it is not the official code of the reviewer's references (we cannot find an open-source official code referenced by the reviewer), we remain conservative in adding this as a baseline to ensure fairness. We would appreciate the reviewer's understanding. Hope the new update can provide quick insights about how ESN works on learning chaos dynamics.
>
> [1] Li, Zongyi, et al. "Learning dissipative dynamics in chaotic systems." arXiv preprint arXiv:2106.06898 (2021).
>
> [2] Schiff, Yair, et al. "DySLIM: Dynamics Stable Learning by Invariant Measure for Chaotic Systems." arXiv preprint arXiv:2402.04467 (2024).
>
> [3] Jaideep Pathak, Zhixin Lu, Brian R. Hunt, Michelle Girvan, Edward Ott; Using machine learning to replicate chaotic attractors and calculate Lyapunov exponents from data. Chaos 1 December 2017; 27 (12): 121102. https://doi.org/10.1063/1.5010300
>
> [4] Pathak, J., Hunt, B., Girvan, M., Lu, Z., & Ott, E. (2018). Model-Free Prediction of Large Spatiotemporally Chaotic Systems from Data: A Reservoir Computing Approach. Physical Review Letters, 120(2), 024102.

---

> ### Author Response · Authors · 2024-11-21
> **Responses to Minor Weakness 1 and Minor Weakness 2**
>
> **Answer Minor Weakness 1**:
> Thank you for the question. We have revised the manuscript in Lines 171-175.
>
> **Answer to Minor Weakness 2**:
> We appreciate the reviewer’s thoughtful exploration of this question. If we have interpreted the inquiry correctly, it pertains to the log-Sobolev constant $c_{\text{LSI}}$. We believe the misunderstanding may stem from the labeling in Table 2, specifically Column 3, titled “relaxation time.” In our paper, we define the relaxation time as
> $k = \frac{1}{c_{LSI}} \log (\frac{\lVert z \lVert_2}{\epsilon})$
> as described in Lines 284–294. Here, $k$ is governed by the hyperparameter $c_{\text{LSI}}$. If the initial state is already within the attractor, the norm of the state $\lVert z \lVert_2$ will be small, resulting in a short relaxation time. Under these conditions, the contraction phase quickly transitions to the measure-invariant phase. Our data split accounts for such scenarios, and these situations have no significant impact on the training results, as sufficient data remains available for training the measure-invariant model.
>
> 1. **Estimating $c_{\text{LSI}}$:**
>    $c_{\text{LSI}}$ is not particularly challenging to estimate (the calculation referred to general response Calculation Of Relaxation Time), as the relaxation time is an intrinsic property of dissipative systems. For partial differential equations (PDEs) and ordinary differential equations (ODEs), the relaxation time can often be calculated explicitly due to their well-defined spectral properties [1, 2]. For example:
>    - In Lorenz 63 and Lorenz 96 systems, $c_{\text{LSI}}$ can be estimated based on the damping terms.
>    - For the Kuramoto–Sivashinsky (KS) equation, the biharmonic operator provides a basis for bounding $c_{\text{LSI}}$ [3].
>    - Similarly, for the Navier–Stokes (NS) equations, the diffusion term is key to estimating $c_{\text{LSI}}$ [4].
>
> 2. We understand the review's concern in the larger values of $k$, or $\frac{1}{c_{LSI}}$, which we've corrected the misunderstood term from "relaxation time" to "Log-Sobolev Constant" (in Table 2). The time steps before $\frac{1}{c_{LSI}} \log (\frac{\lVert z \lVert_2}{\epsilon})$ correspond to the contraction phase, and the time steps after correspond to the measure-invariant phase. In fact, we find that when the hyperparameter $\frac{1}{c_{LSI}}$ exceeds $5$, the relaxation time is sufficiently long to allow the state to evolve toward the attractor. Thus, there's no need to further test the constant above 9 for the contraction phase. This is consistent with the relaxation time typically proven in [1], which is usually logarithmic. From the measure-invariant aspect, continuously increasing the parameter beyond this range does not have a positive impact on learning long-term statistics. Practically, if the hyperparameter $\frac{1}{c_{LSI}}$  is set too high, the split data volume may become insufficient for training the measure-invariant phase under Equation 8 in our paper.
>
>
> [1] Guionnet, Alice, and Bogusław Zegarlinksi. "Lectures on logarithmic Sobolev inequalities." Séminaire de probabilités XXXVI (2003): 1-134.
>
> [2] Sobolev, Sergeĭ Lʹvovich. Partial differential equations of mathematical physics: International series of monographs in pure and applied mathematics. Elsevier, 2016.
>
> [3] Tadmor, Eitan. "The well-posedness of the Kuramoto–Sivashinsky equation." SIAM Journal on Mathematical analysis 17.4 (1986): 884-893.
>
> [4] Kupiainen, Antti. "Ergodicity of two dimensional turbulence." arXiv preprint arXiv:1005.0587 (2010).

---

> ### Author Response · Authors · 2024-11-21
> **Responses to Questions 1 and Question 3**
>
> **Answer to Question 1**:
>
> We appreciate the opportunity to clarify our work and be enlightened by your expertise. Our framework is firmly grounded in the theory of the Koopman operator, as outlined in the introduction (Lines 47–49, 98–106) and supported by references [1, 2, 3]. The motivation for this paper stems from the mean ergodic theorem, specifically its second measure-invariant phase. Our primary contribution is the development of a method for learning dissipative chaotic systems evolution in function space by embedding physical knowledge, which we classify within Koopman learning theory. To enhance the rigor of our approach, we have added a citation to Equation (2) in the revised manuscript and included a comparison to classical Koopman theory in Appendix D. We have been careful to delineate our contributions accurately, avoiding overstatement. As noted, our work focuses on offering a novel perspective on leveraging Poincaré flow theories to better understand and learn dissipative chaotic systems. This is achieved by integrating modern Koopman operator theory into the inspired deep learning method, which represents the core innovation of our approach.
>
> [1] Koopman, Bernard O., and J. V. Neumann. "Dynamical systems of continuous spectra." Proceedings of the National Academy of Sciences 18.3 (1932): 255-263.
>
> [2] Birkhoff, George D., and Bernard O. Koopman. "Recent contributions to the ergodic theory." Proceedings of the National Academy of Sciences 18.3 (1932): 279-282.
>
> [3] Koopman, Bernard O. "Hamiltonian systems and transformation in Hilbert space." Proceedings of the National Academy of Sciences 17.5 (1931): 315-318.
>
>
>
> **Answer to Question 3**:
>
> Thanks for this question. Following the response to Question 1, we'd like to elaborate on the following:
> Firstly, the fundamentals of ergodic theory were pioneered by Henri Poincaré [1]. Poincaré observed that for Hamiltonian systems with infinitely many particles, solving for long-term behavior analytically is not theoretically feasible. However, he discovered the important recurrence properties of such systems, now formalized as Poincaré’s recurrence theorem. These recurrence properties, along with conserved measures, characterize many chaotic systems whose long-term behavior is ergodic and exhibits an invariant measure.
>
> Secondly, we named our model the "Poincaré Flow Neural Network", not just as an homage but also to highlight these foundational principles in the inspired method. The network is specifically designed to learn and analyze the evolution of chaotic dynamical systems by leveraging principles similar to recurrence and invariant measures that Poincaré explored.
>
> Hence, our work builds upon the advancements made by Birkhoff, Koopman, and John von Neumann [2, 3], integrating these ergodic principles into a modern neural network framework. This integration enables the network to effectively capture the underlying structure and dynamics of chaotic systems. The loss function in Equation 8 closely connects to Poincaré contribution.
> Therefore, the name "Poincaré Flow Neural Network" exactly implies the network's functionality and theoretical foundations.
>
> [1] Poincaré, Henri. Œuvres de Henri Poincaré: publiées sous les auspices de l'Académie des sciences... Vol. 1. Gauthier-Villars et cie, 1928.
>
> [2] Neumann, J. V. "Proof of the quasi-ergodic hypothesis." Proceedings of the National Academy of Sciences 18.1 (1932): 70-82.
>
> [3] Koopman, Bernard O., and J. V. Neumann. "Dynamical systems of continuous spectra." Proceedings of the National Academy of Sciences 18.3 (1932): 255-263.

---

> ### Author Response · Authors · 2024-11-21
> **Responses to Question 4 - 6**
>
> **Answer to Question 4**:
>
>  Thanks for spotting this issue. Yes, it is the absolute error between the ground truth and predicted TKE. It has been revised in the latest updated version.
>
> **Answer to Question 5**:
>
> Thank you for this question. This statement relates to ergodic theory. Operators associated with chaotic systems exhibiting ergodic behaviour possess a mixing of the discrete and continuous spectrum on the unit complex circle [1, 2]. Consequently, the spectrum of the operator $\mathcal{G}$, i.e. eigenvalues, indicates the learned features of the system. PFNN learnt this intrinsic property, validated from the learned spectrum of $\mathcal{G}$. The spectrum is densely distributed on the unit complex circle (see Figure 2 in the manuscript), aligning with the Koopman-von Neumann (KvN) ergodic theorem.
>
> In contrast, FNO and MNO [3, 4] rely on Fast Fourier Transforms (FFT), which extract low-mode and integer Fourier components. Mathematically, the integer Fourier spectrum represents periodic behavior, which contradicts ergodicity (see Theorem 1.1 in [6], Page 3). This approach, therefore, is insufficient for capturing the continuous spectrum essential to ergodic behaviors.
>
>
> [1] Koopman, Bernard O., and J. V. Neumann. "Dynamical systems of continuous spectra." Proceedings of the National Academy of Sciences 18.3 (1932): 255-263.
>
> [2] Neumann, J. V. "Proof of the quasi-ergodic hypothesis." Proceedings of the National Academy of Sciences 18.1 (1932): 70-82.
>
> [3] Li, Zongyi, et al. "Learning chaotic dynamics in dissipative systems." Advances in Neural Information Processing Systems 35 (2022): 16768-16781.
>
> [4] Li, Zongyi, et al. "Fourier neural operator for parametric partial differential equations." arXiv preprint arXiv:2010.08895 (2020).
>
> [6] C. McMullen. "Ergodic theory, geometry and dynamics." https://people.math.harvard.edu/~ctm/papers/home/text/class/notes/ergodic/course.pdf
>
>
> **Answer to Question 6**:
>
> Models such as LSTM and Koopman struggle to capture transient behaviours because trajectory data is dominated by ergodic behaviour. In dissipative chaotic systems, where the relaxation time is logarithmic, most of the data is concentrated on attractors (short contraction phase but long measure-invariant phase). As a result, training a model on the entire trajectory makes it difficult for the model to learn the transient dynamics effectively.
>
> ---
> Thank you for your thoughtful review and valuable questions. We greatly appreciate your time and effort in providing feedback to improve our work. We hope our answers address your questions and concerns effectively. We look forward to your further comments and insights.

---

> ### Author Response · Authors · 2024-11-24
>
> May I ask the reviewer whether our response/improvement made has addressed your concerns? There is not much time left for discussion. We would be pleased to answer your further questions. Many thanks.

---

> > ### Comment · Reviewer_CMhA · 2024-11-24
> > **Response**
> >
> > Thank you for updating the manuscript to properly explain how the proposed method relates to Koopman operator learning - it is a significant improvement. I still think the title of the paper is too vague to be useful, and would perhaps suggest adding the word "dissipative", but that's up to you.
> >
> > Thank you for adding the uncertainty estimates in Table 1. I didn't expect them to be so small. Since the table now looks very busy, I would be happy for you to include these in the appendix only, and otherwise leave the table as it was before.
> >
> > Thank you for adding the ESN baseline, but I find the results you've plotted in the notebook extremely unconvincing. I have personally trained ESNs on the Lorenz 63 system using significantly less data and easily achieved accurate predictions well in excess of 5 Lyapunov times with minimal tuning. You can also see this clearly in the Pathak et al. paper.
> >
> > > We evaluated the Koopman baseline using the default configuration of the pyKoopman toolbox (https://pykoopman.readthedocs.io/en/master/), which provides a strong baseline for comparison.
> >
> > So you just used the default settings and didn't do any hyperparameter optimization? Did you optimize the hyperparameters of any of your baselines? I don't see any mention of this in the paper, and I don't think it can be taken for granted that the default hyperparameters of the baselines will also be the optimal ones for your particular experiments. I'm afraid this casts doubt on the robustness of the baselines.
> >
> > > In contrast, FNO and MNO [4, 5] rely on Fast Fourier Transforms (FFT), which extract low-mode and integer Fourier components. Mathematically, the integer Fourier spectrum represents periodic behavior, which contradicts ergodicity (see Theorem 1.1 in [6], Page 3). This approach, therefore, is insufficient for capturing the continuous spectrum essential to ergodic behaviors.
> >
> > This statement appears to contradict the universal approximation properties of neural operators, see e.g. [1].
> >
> > > Models such as LSTM and Koopman struggle to capture transient behaviours because trajectory data is dominated by ergodic behaviour. In dissipative chaotic systems, where the relaxation time is logarithmic, most of the data is concentrated on attractors (short contraction phase but long measure-invariant phase). As a result, training a model on the entire trajectory makes it difficult for the model to learn the transient dynamics effectively.
> >
> > The key insight of your approach appears to be: dissipative chaotic systems exhibit distinct contractive and measure-invariant dynamics and therefore you should train two models instead of one (you also derive regularization terms based on the spectral properties of the Koopman operator in each phase). Couldn't this "two-phase" approach be used straightforwardly with any of your baselines, minus the spectral regularization?
> >
> > This led me to question how much of the supposed benefit of your method is due to the two-phase learning approach and how much is due to the regularization. However, there are a number of inconsistencies in the detailed ablations in Table 10. Firstly, for the base hyperparameters ($\gamma_1 = 0.3$, $\gamma_2 = 0.5$, $L = 256$, $\frac{1}{C_{LSI}} = 5$), the results should be identical across each row, but they're not. Secondly, identical or nearly identical numbers appear in a number of places where they should not. For example, compare the long-term metrics for $\gamma_1 = 0.3$ and $\gamma_1 = 0.5$. In addition, compare the numbers at the following pairs of (row,column) coordinates:
> >
> > 1. (4,1) vs (9,1)
> > 2. (4,2) vs (13,2)
> > 3. (5,3) vs (20,3)
> >
> > All of these numbers differ by exactly one digit, which is unlikely when there are about $10^{11}$ such possible numbers. As a result, I have little confidence in the ablations.
> >
> > [1] Lanthaler, Samuel, Zongyi Li, and Andrew M. Stuart. "The nonlocal neural operator: Universal approximation." arXiv preprint arXiv:2304.13221 (2023).

---

> ### Author Response · Authors · 2024-11-25
> **Response to Reviewer CMhA**
>
> We would like to thank the reviewer for carefully revising our updated MS and comments. Below is our response:
>
> >  I didn't expect them to be so small. Since the table now looks very busy, I would be happy for you to include these in the appendix only, and otherwise leave the table as it was before.
>
> **Uncertainty not in expectation.** As stated in our previous response, the uncertainty in the Maximum Mean Discrepancy (MMD) metric is expected to be small, as MMD measures the mean discrepancy between distributions. Additionally, while the KL divergence is relatively larger compared to MMD (as seen in similar applications, such as Dyslim), it converges well with our sampling strategy given a sufficiently long horizon. Could you please clarify what level of uncertainty you were expecting? We would greatly appreciate if the reviewer could provide specific reasons as to why the uncertainty should not be this small for such long-time averaged statistics, rather than simply stating 'I am not expected'. We are disappointed by the suggestion to move our table to the Appendix, given that it was highlighted as a major weakness (see Weakness 3). We have treated your comments with care, but at this point, it appears that this change may not substantially benefit the overall presentation.
>
> > Thank you for adding the ESN baseline, but I find the results you've plotted in the notebook extremely unconvincing. I have personally trained ESNs on the Lorenz 63 system using significantly less data and easily achieved accurate predictions well in excess of 5 Lyapunov times with minimal tuning. You can also see this clearly in the Pathak et al. paper.
>
> **ESN implementation.** As mentioned in our previous response, we have not re-implemented the official implementations of ESN model due to the lack of an available well-documented open-source implementation. To avoid an unfair comparison, we did not initially include ESN as a baseline. We perform such investigation only upon the reviewer's request. **We would appreciate it if the reviewer could share the implemented code as soon as possible, since deadline is approaching.**
>
> > So you just used the default settings and didn't do any hyperparameter optimization? Did you optimize the hyperparameters of any of your baselines? I don't see any mention of this in the paper, and I don't think it can be taken for granted that the default hyperparameters of the baselines will also be the optimal ones for your particular experiments. I'm afraid this casts doubt on the robustness of the baselines.
>
> **Koopman baseline.** We maintained the model structure of the PyKoopman implementation while increasing the number of parameters to match the level indicated in \url{https://anonymous.4open.science/r/PFNN-F461/README.md}. However, we are unclear about which specific hyperparameters the reviewer is addressing. If the concern pertains to hyperparameters such as the learning rate or batch size, please note that PyKoopman already provides default settings for these parameters. Arbitrarily adjusting them could significantly impact the results. Our findings indicate that the Koopman-based baseline performs robustly, suggesting that the current hyperparameter settings are effective. Additionally, extensive fine-tuning of baseline implementations is generally not the primary focus in PFNN and other machine learning studies. Therefore, we believe that the criticism regarding hyperparameter tuning is vague and does not convincingly undermine our work.
>
> > This statement appears to contradict the universal approximation properties of neural operators, see e.g. [1].
>
> **Claim on FFT contradicts with ergodicity.** Mathematically, the integer Fourier spectrum represents periodic behavior, which contradicts ergodicity [1, 2, 3]. This point is verfied by many Fields-medal mathematicians (Curtis T. McMullen, Omri Sarig, Terence Tao) instead of one learning paper.
>
>     [1] C. McMullen. "Ergodic theory, geometry and dynamics." https://people.math.harvard.edu/~ctm/papers/home/text/class/notes/ergodic/course.pdf
>
>     [2] Sarig, Omri. "Lecture notes on ergodic theory." Lecture Notes, Penn. State University (2009).
>
>     [3] https://terrytao.wordpress.com/2008/02/04/254a-lecture-9-ergodicity/

---

> ### Author Response · Authors · 2024-11-25
> **Response to Reviewer CMhA (Continuous)**
>
> > The key insight of your approach appears to be: dissipative chaotic systems exhibit distinct contractive and measure-invariant dynamics and therefore you should train two models instead of one (you also derive regularization terms based on the spectral properties of the Koopman operator in each phase). Couldn't this "two-phase" approach be used straightforwardly with any of your baselines, minus the spectral regularization?
>
> **Two phase training on other baseline.** The core insight of our two-phase training approach lies in the respective spectral regularization. The default implementation of these baselines does not include such two-phase training. We are unclear on why we should apply two-phase training to these baselines, as without incorporating such regularization deviates significantly from our primary research focus and their intended designs of baselines. If the reviewer is suggesting that we modify existing baselines to include our method, we find this much beyond the scope of our work.
>
>
> > This led me to question how much of the supposed benefit of your method is due to the two-phase learning approach and how much is due to the regularization. However, there are a number of inconsistencies in the detailed ablations in Table 10.
>
> **The inconsistency in Table 10.** Regarding the first question of reviewer, some rows are expected to be identical. For short-term results, we conducted cross-validation and evaluated outcomes from random initial states multiple times to assess uncertainty quantification (UQ). For long-term results (without UQ), the entries are identical because they represent long-term average results, which are inherently robust. The minimal variation observed not only confirms the correctness of our implementation but also highlights the robustness of our approach. Furthermore, we are unsure why the reviewer believes these entries should not be similar.
>
> > All of these numbers differ by exactly one digit, which is unlikely when there are about $10^{11}$ such possible numbers. As a result, I have little confidence in the ablations.
>
> We would appreciate more convincing reasons questioning the validity of our results, rather than casually suggesting "should not" or "not as expected." Such statements could influence the perception of our research integrity, especially if the justification is unclear, which we find concerning.

---

> ### Author Response · Authors · 2024-11-25
> **Follow-up response**
>
> Dear Reviewer CMhA,
>
> There are only 43 hours left. We're currently working with high dedication on polishing the paper to the best we can. Therefore, an answer to our questions would be really helpful.
>
> Thanks again!

---

> ### Comment · Reviewer_CMhA · 2024-11-26
> **Main Issues**
>
> ## Presentation
> > We are disappointed by the suggestion to move our table to the Appendix, given that it was highlighted as a major weakness (see Weakness 3). We have treated your comments with care, but at this point, it appears that this change may not substantially benefit the overall presentation.
>
> There seems to have been a misunderstanding. The addition of uncertainty estimates is an important improvement. I was merely pointing out that the new table, with uncertainty estimates over many different orders of magnitude, is now harder to read; it feels crowded. Therefore, as a compromise, I suggested that you could keep your original table in the main body of the paper, and place the complete table in the appendix, thereby achieving the “best of both worlds”. Ultimately it’s up to you, I was just trying to be helpful.
>
> In terms of presentation generally, the addition of uncertainty estimates and the proper citations of the Koopman learning literature are both important improvements and would justify an increase in my score. However, additional concerns about the robustness of the baselines and the correctness of the reported results have since come to light, which I outlined in my previous response and which the authors have not addressed. I will attempt to outline them again, very clearly.
>
> ## Baselines
> > However, we are unclear about which specific hyperparameters the reviewer is addressing. If the concern pertains to hyperparameters such as the learning rate or batch size, please note that PyKoopman already provides default settings for these parameters.
>
> Learning rate and batch size are indeed two hyperparameters that can affect the performance of your baselines. Others may relate to the expressiveness of the particular architecture and may directly impact the performance on your experiments. However, it is clear from your answers that you have not attempted to optimize any of these for your baselines.
>
> > Arbitrarily adjusting them could significantly impact the results.
>
> Hyperparameter optimization consists of **systematically** adjusting the hyperparameters. That doing so could significantly impact the results is of course the whole point, that is, it could improve the results.
>
> > Additionally, extensive fine-tuning of baseline implementations is generally not the primary focus in PFNN and other machine learning studies.
>
> You have performed **zero fine-tuning** of the baselines. To suggest that adequate tuning of your baselines is not important in machine learning studies, particularly when you claim to beat the baselines, is bizarre.
>
> ## Ablations
> > Regarding the first question of reviewer, some rows are expected to be identical.
>
> Please read what I said carefully. You are correct that some rows are expected to be identical; one of the issues is that **some of the rows that should be identical are not identical**. This is clearly wrong and it is frustrating to have to repeat myself.
>
> In addition, allow me to spell out some of the unlikely coincidences that occur in Table 10. The following pairs of numbers appear in the first, second, and third columns, respectively, of Table 10:
>
> - $12.2610 \pm 1.5682$ and $11.2610 \pm 1.5682$
> - $23.2001 \pm 0.7418$ and $21.2001 \pm 0.7418$
> - $25.8763 \pm 0.4796$ and $24.8763 \pm 0.4796$
>
> The following pairs of numbers appear in rows 3 and 4:
>
> - $7.1571$ and $17.1571$
> - $5.9686$ and $7.9686$
>
> I can think of no scientific reason for these numbers to differ by an integer and have identical standard deviations.
>
> > Such statements could influence the perception of our research integrity, especially if the justification is unclear, which we find concerning.
>
> To say that these coincidences are surprising in no way influences “the perception of [your] research integrity”. However, how you choose to respond certainly does.
>
> ## Summary
> The main issues that remain with this paper, from my perspective, are:
>
> 1. The authors have performed zero fine-tuning of the baseline methods, instead using the "default parameters" in each case. As a result, **I have low confidence in the robustness of the experimental results and the claimed improvement over existing methods**, which in many of the experiments is marginal.
> 2. The paper contains almost no details about the training procedure (e.g. learning rate schedule, epochs) for any method (the one exception is the Kuramoto-Sivashinsky system). The provided code contains only a couple of scripts and there is no trace of the baselines or data generation, or indeed any code for systematically reproducing the results in the paper. Taking paper and code together, **I have zero confidence in the reproducibility of the results**.
> 3. **There are irregularities in the ablations reported in Table 10**, some of which are provably wrong (different results for the same hyperparameters) and some of which appear to be highly unlikely coincidences. I tried to bring this up with the authors, but they refused to engage on the substance of the issue.

---

> ### Author Response · Authors · 2024-11-27
> **Response to Reviewer**
>
> Thank you very much for your insightful comments regarding the inconsistencies in Table 10 of our manuscript. We admit there are inconsistencies presented arose due to copy misalignment during the preparation process. Specifically as you pointed out in Table 10:
> 1. (4,1) vs (9,1)
> 2. (4,2) vs (13,2)
> 3. (5,3) vs (20,3).
>
> Upon carefully revisiting your feedback, we realized that Table 10 was prepared during the different stages of our research, which led to inconsistencies. In order to address this, we will systematically re-evaluate Table 10 and we will provide implementation details for the updated Table 10. We commit to provide an updated and corrected version of Table 10 in the camera-ready version, as this is a very time-consuming task and AC also do not request such additional work/experiments at this stage. We do hope the reviewer can understand.
>
> **Actions**
>
> 1. Systematic Re-evaluation and Table Update:
>     Table 2 has been updated to accurately and transparently reflect our findings.
>
> 2.  Update Table 10 in camera-ready version:
>     Due to time constraints during the rebuttal stage, we were unable to fully re-evaluate the ablation study presented in Table 10 within 24 hours. To prevent any potential misunderstandings, we have removed Table 10 from the revised version of our manuscript. However, we commit to provide an updated and corrected version of Table 10 in the camera-ready version.
>
> 3. Provision of Supplementary Resources:
>     We have provided a comprehensive Notebook that enables interested reviewers and readers to independently evaluate the metrics. **We believe this approach will maintain the integrity of our results while offering full transparency and flexibility for further analysis** (see the Link: https://anonymous.4open.science/r/PFNN-F461/ablation_notebooks/Readme.md). Correspondingly, the KS data generation code is provided at https://anonymous.4open.science/r/PFNN-F461/lake/data_generation/KS/ks.m.
>
> **Assurance of Conclusions**
>
> These changes to the ablation studies do not affect our overall conclusions and contributions. We believe this approach maintains the integrity of our results while offering full transparency and flexibility for further analysis. We greatly appreciate your attention to detail and your valuable suggestions, which have significantly contributed to the refinement of our work. Your careful review has been instrumental in improving the quality of our manuscript.
>
>
> **Response to No details about training procedures**
>
> We would like to remind the reviewer that we have provided comprehensive training files, which include all necessary details for training, available at https://anonymous.4open.science/r/PFNN-F461/scripts/L96_train_contract.py, https://anonymous.4open.science/r/PFNN-F461/scripts/NS_train_invariant.py. We believe this demonstrates our commitment to transparency during the anonymous submission stage. Additionally, we hope that the provided hyperparameter table addresses your concerns regarding baseline implementations.

---

> ### Author Response · Authors · 2024-11-27
> **Response to Reviewer (continue)**
>
> **The Hyperparameter Table of Baselines**
>
> We ensured **a fair comparison** by aligning the parameters to match those of our models. To further clarify the settings for all baseline methods, we have provided a detailed table below, outlining both the model structures and training settings. Our details can be found in Table 9 in manuscript.
>
> **L63**
>
> |  **L63 Example**     | **LSTM** | **Koopman** | **FNO (MLP)** | **MNO (MLP)** | **PFNN** |
> |----------------|---------|---------|---------|---------|---------|
> | Batch size     |   256    |   256   |    256  |   256   | 256      |
> | Layer          | FC+LSTM Block |FC | FC | FC | FC | FC|
> | Number of Layers |  3   |   6  |  3   |  3    | 6 |
> | Hidden dimension |   80    |   150   |   150   |  150  | 150    |
> | number of parameters |  18078     |  182253   |   114303   |   114303   |114153 |
> | Activation     | ReLU       | ReLU   |  ReLU    | ReLU     | ReLU      | ReLU |
> | Optimizer      | Adam      | Adam     | Adam    | Adam     | Adam      |
> | Learning rate  | 0.0001      | 0.0001      | 0.0001      | 0.0001      | 0.0001      |
> | Epochs         | 100 | 100 | 100 | 100 | 100 |
> | Loss function  |  MSE    |  MSE + Reconstruction  |  $L_2$   | $L_2$ + disspasive loss + post processing | MSE + Contraction/Consistent Loss + Reconstruction |
>
> **KS Equation**
>
> |  **KS-128 Example**     | **LSTM** | **Koopman** | **FNO** | **MNO** | **PFNN** |
> |----------------|---------|---------|---------|---------|---------|
> | Batch size     |    256   |  256     |    256   |   256   | 256   |
> | Layer          | FC+LSTM Block | FC | SpectralCovn-1D| SpectralCovn-1D | FC |
> | Hidden dimension |   100   |    512   |   (64,128)     |   (64,128)     |      512   |
> | # of layers | 3 | 6 | 3 | 3 | 6 |
> | # of parameters |   20903    |     1560320     |    41873     |  41873      |    1444480    |
> | Activation     | ReLU      |  ReLU  |  SELU    |  SELU   | ReLU   |
> | Optimizer      | Adam       | Adam      | Adam     | Adam       | Adam       |
> | Learning rate  | 0.0005      | 0.0005      | 0.0005      | 0.0005      | 0.0005      |
> | Epochs         | 100 | 100 | 100 | 100 | 100 |
> | Loss function  |  MSE    |  MSE + Reconstruction  |  $L_2$   | $L_2$ + disspasive loss + post processing | MSE + Contraction/Consistent Loss + Reconstruction |
>
>
> Please note that the SpectralConv-1D block in FNO and MNO based on FFT is parameter efficient compared with FC.
>
>
> **NS Equation**
>
> |  **NS Example**     | **LSTM** | **Koopman** | **FNO** | **MNO** | **PFNN** |
> |----------------|---------|---------|---------|---------|---------|
> | Batch size     |    50   | 50      | 50      | 50      | 50      |
> | Layer          | Conv-LSTM | Conv-2D | SpectralConv-2D | SpectralConv-2D | Attention |
> | Hidden dimension |   (32,16,16)    |    1024    |   (64,64,64)     |   (64,64,64)     |      512   |
> | # of layers | 2 blocks | 8 | 4 blocks | 4 blocks | 8 |
> | # of parameters |    7873444    |    7653630     |     4507489   |   4877289      |    4425826    |
> | Activation     |    ReLU   |  GELU   |   SELU    |    SELU   | GELU      |
> | Optimizer      | Adam      | Adam     | Adam    | Adam     | Adam      |
> | Learning rate  | 0.0001      | 0.0001      | 0.0001     | 0.0001      | 0.0001      |
> | Epochs | 50 | 50 | 50 | 50 | 50 | 50|
> | Loss function  |  MSE    |  MSE + Reconstruction  |  $L_2$   | Sobolev loss + disspasive loss + post processing | MSE + Contraction/Consistent Loss + Reconstruction |

---

> ### Author Response · Authors · 2024-11-27
> **Response to Reviewer (continue)**
>
> **The contribution of method**
>
> 1. **New Perspective on Dissipative Chaos** We introduce a novel framework that distinguishes dissipative chaos into two distinct phases: contraction and measure-invariant. This perspective offers a deeper understanding of the underlying dynamics of dissipative systems.
>
> 2. **Tailored Loss Functions** For each identified phase, we developed corresponding loss functions that are specifically designed to address the unique characteristics of contraction and measure-invariant phases.
>
> 3. **Pioneering Approach** To the best of our knowledge, this dual-phase approach and the associated loss functions have not been previously explored, marking a significant advancement in the study of dissipative chaos.
>
> **Improvement of method**
>
> 1. **Energy Spectra** Our method enhances energy spectra, as illustrated in Figure 2 of the manuscript.
>
> 2. **Lorenz 63** Achieves a **22%** improvement in long-term statistical accuracy.
>
> 3. **Kuramoto-Sivashinsky (KS) Equation** Realizes a **threefold (300%)** improvement in long-term statistical performance.
>
> 4. **80 dimensional - Lorenz 93 System** Attains a **33%** improvement in KLD, indicating a closer alignment with the true distribution.
>
> 5. **Turbulent Kinetic Energy (TKE):** Navier-Stokes (NS) Equations: Demonstrates a **14%** improvement in TKE, reflecting enhanced modeling of turbulent flows.
>
> We respectfully show the improvements to response the reviewer's concerns in marginal improvements. The substantial improvements across multiple key metrics and systems underscore the effectiveness and impact of our proposed method.
>
> Thank you once again for your detailed, thoughtful and thorough review.

---

> ### Comment · Reviewer_CMhA · 2024-11-29
>
> Given significant improvements in the presentation of the paper, I have increased my score from 3 to 5. The tables of hyperparameters shared by the authors in their most recent response should be included in the Appendix.
>
> **My main reason for not giving a higher score is that I remain concerned about the robustness of the baselines and, in turn, the correctness of the claims made in the paper.** In particular, the authors have been somewhat vague about their approach to training the baselines. In their most recent response, they claim to have attempted a “fair comparison” by aligning the number of parameters across models. While this is one possible approach to making a principled comparison between models, **the claim made in the paper is not “our model is better when using the same number of parameters” but rather an unqualified “our model is better”**. This is not a trivial distinction, especially when considering that all of the models in the paper are really quite small, meaning that practitioners may be more concerned about other factors such as accuracy or runtime.
>
> To evaluate the “our model is better” claim in its own right, independent of the number of parameters, we might then try to compare the hyperparameters used by the authors with those used by the original authors of the baselines. When doing so, significant differences emerge that could impact performance. For example, the original MNO paper used 6 hidden layers for L63 and 4 for KS, while this paper only uses 3 in both cases. The original MNO paper used significantly higher learning rates than those indicated by the authors here. MNO used Sobolev losses, which are not mentioned in the table here (but may have been used in practice, we don’t know). This is not an exhaustive list, but rather the result of a quick comparison between the two papers.
>
> Taking all of this together, I repeat my opinion that the claimed (absolute) improvement over the existing literature is not robust in its current form. I would encourage the authors to put as much care into optimizing the baselines as they have put into their own method, in order to ensure a properly fair comparison.

---

> ### Author Response · Authors · 2024-11-29
> **Thanks and acknowledging the feedback.**
>
> We would like to thank reviewer CMhA for the thoughtful feedback and for acknowledging the significant improvements in our paper's presentation. We appreciate the decision to increase your score. Our training procedure follows the MNO paper, it practically uses $L^2$ loss during training in Lorenz and Sobolev loss (1-order) for NS. To address your concerns, we will carefully revise our claims according to your suggestions to avoid confusion. Specifically, in the camera-ready version, we will reframe our claims to emphasize our model's performance within the context of comparable parameters and other relevant factors, such as hyperparameters. The run time (shown in Readme of anonymous GitHub) and listed parameters count will be included into our appendix.  Thank you again for your valuable feedback. Your insights have been important in helping us improve the quality of our work, which we would like to acknowledge in the final version.

---

### Author Response · Authors · 2024-11-21
**General Response**

Dear all reviewers,

We sincerely thank all reviewers for recognizing the novelty, presentation quality, theoretical strengths, and depth of our experiments. Your insightful comments have greatly contributed to improving our work, and we will acknowledge all of your efforts in the camera-ready version.

We have provided an updated version of our manuscript, with revisions highlighted in red for ease of reference. All reviewer comments have been addressed in the revised version.

Additionally, we have conducted further uncertainty quantification of our model, as presented in Table 1. We believe these new results provide additional insights into the robustness of our approach.

We have also updated the implementation link, which now includes parameter counts, Koopman model architecture, run time and a comparison with the Echo State Network (ESN) on the Lorenz 63 system/Kuramoto–Sivashinsky equation, as suggested by **Reviewer CMHA**. This can be found at: https://anonymous.4open.science/r/PFNN-F461/README.md. Additionally, we have extended our experiments to include the $256 \times 256$ Navier-Stokes equations to demonstrate the scalability of our approach and contraction in the physical space (suggested by Reviewer pS8g). The updated link also contains a wall-clock (run) time comparison with other baseline methods (suggested by **Reviewer GyXm**). We also update the detailed information of ablation study with transparent scripts.

We are grateful for your constructive feedback, and we look forward to further enhancing our work based on your suggestions.

---

> ### Author Response · Authors · 2024-11-21
> **General Response For the Calulation Of Relaxation Time**
>
> We provide a general response for the calculation of relaxation time.
>
> Relaxation time $k$ is an intrinsic property of chaotic dynamics. When the initial state $z$ varies, so does $k$. Actually, it is practically calculated as $\frac{1}{c_{LSI}} \log (\frac{\lVert z \lVert_2}{\epsilon})$, we don't involve the learned feature function for the estimation of relaxation time (see Line 290-293). For the attractor in dissipative chaos, the $\log (\frac{\lVert z \lVert_2}{\epsilon})$ is parametrized by $\lVert z \lVert_2$, which can be regarded as the **energy-like quantity**. When the initial state is close to the attractor (i.e., $\lVert z \lVert_2$ is small in dissipative chaos), the relaxation time will be short.
>
> The calculation of $k$ involves log-Sobolev constant $c_{LSI}$ and practical threshold $\epsilon$. We explain it step by step. The **log-Sobolev constant** $c_{LSI}$ is highly complex and involves extensive mathematical details that span several pages. Here, we provide a concise sketch of the calculation. Comprehensive and detailed proof will be included in the camera-ready version appendix.
>
> To understand the dissipation mechanisms in underlying dynamical systems, we first examine what induces relaxation in these systems. Specifically, in dissipative Ordinary Differential Equations (ODEs) and Partial Differential Equations (PDEs), relaxation is generally driven by diffusion terms [3]. Below are examples illustrating this phenomenon:
> * **Lorenz 63** A simplified ODE model of convection-diffusion dynamics where relaxation is induced by damping terms, which serve as a simplified form of diffusion.
> * **Lorenz 96** A simplified advection-diffusion model featuring cyclically coupled variables. In this system, damping terms facilitate relaxation through a simplified diffusion mechanism.
> * **Kuramoto–Sivashinsky Equation**  Incorporates hyperdiffusion (biharmonic operator) term to induce relaxation.
> * **Kolmogorov Flow**: Describes fluid motion with viscous diffusion acting on the velocity field. Within the Navier–Stokes framework, viscous diffusion leads to the relaxation of the system.
>
> In the theory of log-Sobolev inequality [1,2], the log-Sobolev constant quantifies the rate at which entropy dissipates in a system. For example, in the Kuramoto-Sivashinsky (KS) equation $u_t + uu_x + u_{xx} + u_{xxxx} = 0$ with periodic boundary conditions $u(t,0) = u(t,L)$ ($L$ is the domain length), the convection term $uu_x$ does not contribute to dissipation. Instead, the dissipation of energy is entirely governed by the hyperdiffusion term $u_{xxxx}$. Applying a Fourier transform to the solution $u(t,x)$ of the KS equation reveals that the dissipation rate induced by the hyperdiffusion term scales as $k^4$ for each wave number $k$. This means that higher-frequency (larger $k$) modes experience stronger dissipation, effectively damping out small-scale structures more rapidly. Here, we can roughly estimate the constant as $c_{LSI} = k^4_{min}$ and $k_{min} = \frac{2\pi}{L}$ is the minimal wave number (Please note this estimation is corresponds to the Poincaré constant, serving as a lower bound for the log-Sobolev constant. That is why we refer to it as a "rough" estimate.). This example can be extended to other systems where the dissipation rate is determined by the spectral properties of the underlying differential operators. Specifically, the smallest non-zero eigenvalue of the Laplace operator (or the biharmonic operator) (or the relevant diffusion operator) often serves as a critical parameter in determining the log-Sobolev constant. This eigenvalue, related to the spectral gap [4], quantifies the rate at which the system dissipates energy and converges to attractors. Similarly, the associated constant caused by diffusion terms for other ODEs and PDEs can also be derived.
>
> After determining the constant $c_{LSI}$, the $\epsilon$ is obtained as $0.01$ since this value has a weak effect due to the logarithmic relationship. Plug the current state $z$, $\epsilon$ and $c_{LSI}$ we can calculate the step $k$ naturally.
>
>
> [1] Guionnet, Alice, and Bogusław Zegarlinksi. "Lectures on logarithmic Sobolev inequalities." Séminaire de probabilités XXXVI (2003): 1-134.
>
> [2] Sobolev, Sergeĭ Lʹvovich. Partial differential equations of mathematical physics: International series of monographs in pure and applied mathematics. Elsevier, 2016.
>
> [3] Kupiainen, Antti. "Ergodicity of two-dimensional turbulence." arXiv preprint arXiv:1005.0587 (2010).
>
> [4] Bakry, Dominique, et al. "Poincaré Inequalities." Analysis and Geometry of Markov Diffusion Operators (2014): 177-233.

---

### Author Response · Authors · 2024-11-27
**General Response**

Dear reviewers and AC:

We hope this message finds you well. As the submission deadline approaches, we have carefully considered and appreciated Reviewer CMhA feedback regarding inconsistencies in Table 10 of our ablation study.

Upon receiving the reviewer's comments, we revisited our code and conducted a systematic re-evaluation of our work. As a result, we have updated Table 2 to more accurately reflect our findings. However, due to the time constraints associated with the impending deadline, we were unable to fully re-evaluate the ablation study. To prevent any potential misunderstandings, we have decided to remove Table 10 from the current version of our paper. We commit to provide an updated and corrected version of Table 10 in the camera-ready version, making everything transparent.

**Major Changes in the revised version:**

Table 2 Correction: during our re-evaluation, we identified a misalignment in the middle column of Table 2, which should correctly display the values with feature dimension $128, 256, 512$, and $1024$.
Feature Space: we determined that a feature dimension of $L=512$ is optimal, whereas dimension $L=1024$ leads to over-parameterization issues. This finding has been incorporated into both Table 2 and the main text of our manuscript.
To maintain transparency and allow interested readers to further explore our metrics, we will be providing a comprehensive notebook alongside our submission (code available at https://anonymous.4open.science/r/PFNN-F461/ablation_notebooks/Readme.md). This resource will enable others to evaluate the metrics independently.

Please be assured that these modifications to the ablation studies do not alter our overall conclusions. We sincerely appreciate your attention to detail and your valuable suggestions, which have significantly contributed to the refinement of our work.

Thank you once again for your constructive feedback and understanding.

---

> ### Author Response · Authors · 2024-11-27
> **General Response (continue)**
>
> The training parameters are available: https://filebin.net/u7b41dqpn0jvz4sc. Instructions for using these parameters can be found in the:  https://anonymous.4open.science/r/PFNN-F461/ablation_notebooks/Readme.md.
>
>
> Thank you!

---

### Meta-Review · Area_Chair_aAde · 2024-12-20

**Metareview:**

The work proposes a method for capturing the invariant measure of dissipative dynamical systems based on Koopman theory. The basis of the approach is to learn an autoencoder which linearizes the dynamics in latent space. The method is well-founded theoretically and supported by many numerical experiments.

**Additional Comments On Reviewer Discussion:**

The authors have been able to address reviewer concerns about positioning the work within the Koopman operator learning literature and added larger-scale numerical experiments. They have also clarified their experimental design and improved the presentation.

---

### Decision · Program_Chairs · 2025-01-22

Accept (Poster)